# Carbon Dioxide Removal via Macroalgae Open-ocean Mariculture and Sinking: An Earth System Modeling Study

Jiajun Wu[1], David P. Keller[1], and Andreas Oschlies[1,2]

[1]GEOMAR Helmholtz Centre for Ocean Research Kiel, Kiel, Germany
[2]Kiel University, Germany

**Correspondence:** Jiajun Wu (jwu@geomar.de)

**Abstract.** In this study we investigate the maximum physical / biogeochemical potential of macroalgae open-ocean mariculture and sinking (MOS) as ocean-based carbon dioxide removal (CDR) method. Embedding a macroalgae model into an Earth system model, we simulate macroalgae mariculture in the open-ocean surface layer followed by fast sinking of the carbon-rich macroalgal biomass to the deep seafloor (depth >3,000m), which assumes no remineralization of the harvested biomass during the quick sinking. We also test the combination of MOS with artificial upwelling (AU), which fertilizes the macroalgae by pumping nutrient-rich deeper water to the surface. The simulations are done under RCP4.5, a moderate emission pathway. When deployed globally between years 2020 and 2100, the carbon captured and exported by MOS is 270 PgC, which is further boosted by AU to 447 PgC. Because of feedbacks in the Earth system, the oceanic carbon inventory only increases by 171.8 PgC (283.9 PgC with AU) in the idealized simulations. More than half of this carbon remains in the ocean after cessation at year 2100 until year 3000.

The major side effect of MOS on pelagic ecosystems is the reduction of phytoplankton net primary production (PNPP) due to the competition for nutrients by macroalgae and canopy shading. MOS shrinks the mid layer oxygen minimum zones (OMZs) by reducing the organic matter export to, and remineralization in, subsurface and intermediate waters, while it creates new OMZs on the seafloor by oxygen consumption from remineralization of sunk biomass. MOS also impacts the global carbon cycle, reduces the atmospheric and terrestrial carbon reservoir when enhancing the ocean carbon reservoir. MOS also enriches dissolved inorganic carbon in the deep ocean. Effects are mostly reversible after cessation of MOS, though recovery is not complete by year 3000. In a sensitivity experiment without remineralization of sunk MOS biomass, the entire MOS-captured carbon is permanently stored in the ocean, but the lack of remineralized nutrients causes a long-term nutrient decline in the surface layers and thus reduces PNPP.

Our results suggest that MOS has, theoretically, a considerable CDR potential as an ocean-based CDR method. However, our simulations also suggest that such large-scale deployment of MOS would have substantial side effects on marine ecosystems and biogeochemistry up to a reorganisation of food webs over large parts of the ocean.

# 1 Introduction

Anthropogenic emissions are rapidly increasing the global atmospheric $CO_2$ concentration. In the last decade (2011 to 2020), global fossil $CO_2$ emissions averaged $\sim$9.49 PgC yr$^{-1}$ (equivalent $\sim$34.8 PgCO2 yr$^{-1}$) with a growth rate of 0.4 % yr$^{-1}$ (Friedlingstein et al., 2021). In 2019, $CO_2$ emissions reached a record high of 9.71 $\pm$0.49 PgC yr$^{-1}$ (equivalent 35.6$\pm$1.8 Pg $CO_2$ yr$^{-1}$), and there is no sign of a peak (Edo et al., 2019; Friedlingstein et al., 2021). The slow speed of emission reductions until now makes it difficult to reach the promised climate goals to keep global warming within the guardrail of 2°C (Peters et al., 2013), much less the recent agreement to seriously consider an even more ambitious 1.5°C goal (UNFCCC, 2015).

In addition to mitigation efforts to reduce greenhouse gas (GHG) emissions, it is increasingly realized that Carbon Dioxide Removal (CDR), sometimes also called Negative Emissions Technologies (NETs), will likely be a necessary step to achieve the targets of the Paris Agreement (Minx et al., 2017; Rogelj et al., 2018). CDR aims to remove $CO_2$ from the atmosphere and store them, ideally permanently, in either the terrestrial, marine or geological carbon reservoirs, thereby mitigating global warming (Glaser, 2010). Due to the limited remaining emission budget (650 $\pm$ 130 Pg $CO_2$ to 1.5 °C and 1300 $\pm$ 130 Pg $CO_2$ to 2 °C), deployment of CDR is required in most pathways studied in the scientific literature to achieve these ambitious targets (Lawrence et al., 2018; IPCC, 2018).

As the second-largest inorganic carbon reservoir on the planet, the ocean plays a pivotal role in naturally regulating the atmospheric $CO_2$ concentration. Since the beginning of the industrial era, the ocean has taken up more than 560 $PgCO_2$, about 25% of the anthropogenic $CO_2$ emissions ($\sim$2030 $PgCO_2$, Gruber et al. (2019); Ciais et al. (2013); Heinze et al. (2015)). Its high carbon storage capacity could theoretically match or exceed fossil fuel resources (Scott et al., 2015). Thus, a variety of ocean-based CDR methods have been proposed to take advantage of this potential storage capacity. The proposed ocean-based CDR approaches aim at increasing the rate of oceanic $CO_2$ uptake and storage by either enhancing abiotic processes (i.e., chemical or physical, e.g. ocean alkalinization (Keller et al., 2014; Taylor et al., 2016; Köhler et al., 2013; Albright et al., 2016)) or biotic processes (e.g. ocean fertilization (Keller et al., 2014; Smetacek et al., 2012; Oschlies et al., 2010b; Matear and Elliott, 2004; Robinson et al., 2014)). Some technologies also seek to remove $CO_2$ directly from seawater and store it in some other reservoir, e.g., a geological one (Eisaman et al., 2012).

Macroalgae species (also known as 'seaweed' or 'kelp') are highly efficient carbon fixers with a high C:N ratio (Atkinson and Smith, 1983; Fernand et al., 2017) and observed net primary production (NPP) rates of 91–522 gC $m^2yr^{-1}$. In the 1970's, the concept of ocean farming using macroalgae for marine carbon sink and bioenergy production was studied with an actual small test farm established off the coast of southern California. These research activities were abandoned due to the damage of the test farm by winter storms and for several technical and economic reasons (Ritschard, 1992). Utilizing macroalgae for biological ocean-based CDR has recently received renewed interest (Duarte et al., 2017; Chung et al., 2011; Gao et al., 2020; Fernand et al., 2017; Raven, 2018). The macroalgae aquaculture industry is well established globally with an annual harvest of over 30 million tonnes wet weight (WW, FAO (2018)). Thus, some proposals have focused on using harvested macroalgae for producing biochar (Roberts et al., 2015; Bird et al., 2011) or bio-energy combined with carbon capture and storage (BECCS, Chung et al. (2011); Buschmann et al. (2017); Gao and McKinley (1994); Chen et al. (2015); Fernand et al. (2017)). However,

as current macroalgae aquaculture facilities are mainly located in coastal regions, the scope to expand macroalgae aquaculture is limited by the shortage of suitable coastal areas due to nutrient availability and shifting temperature regimes (Duarte et al., 2017; Oyinlola et al., 2020). To address these issues, several offshore macroalgae aquaculture facilities have been designed and evaluated (e.g., the SeaweedPaddock by Sherman et al. (2019), the offshore ring by Buck and Buchholz (2004), and the depth-cycling strategy by Navarrete et al. (2021) in which macroalgae are physically towed into the deep nutrient-rich water at night). Moreover, the Advanced Research Projects Agency-Energy (ARPA-E) of the U.S. Department of Energy (DOE) has committed more than 60 million dollars on the Macroalgae Research Inspiring Novel Energy Resources (MARINER) program to develop the technologies for macroalgal biomass production, including integrated ocean cultivation and harvesting systems (APAR-e, 2021). Thus, the ideas of expanding macroalgae cultivation to the open oceans (mariculture) are ambitious but no longer fictional, and they provide a theoretical possibility to expand macroalgae aquaculture to the open-ocean for CDR.

In this study we evaluate 'Macroalgae Open-ocean mariculture and Sinking (MOS)' as ocean-based CDR method that is designed to artificially enhance the macroalgae-based carbon dioxide removal. The aim of this study is to investigate 1) the maximum physical / biogeochemical CDR potential of MOS; 2) the side effects of such large scale deployment, and 3) to understand where offshore macroalgae farming would be viable if done at a large scale. This information is needed to help prioritize further research into CDR, to understand if there are potential MOS side effects that become evident only at large scale, and to provide information on the viability of large-scale offshore macroalgae farming in different regions over time by accounting for the implications of nutrient utilization and climate change.

To do this, simulated macroalgae are seeded and cultivated on offshore floating platforms that are moored to the seabed (e.g., see platform designs in Buck and Buchholz (2004)). The platforms are also assumed to float below the open ocean surface (at 5m depth) to avoid storm damages. At the end of an annual cycle, platforms with matured macroalgae are rapidly sunk to the seafloor and unload the biomass there. This can be thought of as a short circuiting of the biological pump by bringing marine biomass directly to the seafloor without having it remineralized along the way. Afterwards, the sunk biomass is assumed to continue remineralization at the seafloor, consuming oxygen and releasing dissolved inorganic carbon (C), nitrogen (N) and phosphate (P) into the deep ocean where it ideally remains for centuries to millennia (Fig.1). The macroalgae used here is an idealized genus. The assumed constant C:N:P ratio is 400:20:1, which is higher than the stoichiometric ratio of the general phytoplankton in the UVic ESCM (C:N:P=106:16:1, the Redfield ratio). In practice, some of the biomass may also be permanently buried in sediments (Luo et al., 2019; Sichert et al., 2020), and we will explore the extreme case of zero remineralization in the water column in a sensitivity experiment. In another sensitivity experiment we investigate combining MOS with artificial upwelling (AU) to alleviate nutrient limitation in the open ocean surface (Duarte et al., 2017; Kim et al., 2019; Laurens et al., 2020).

To investigate the biogeochemical and climatic implications of MOS we use an Earth system model of intermediate complexity. Though the idea of massive macroalgae cultivation and biomass offsetting for CDR has been assessed in some earlier publications (Orr and Sarmiento, 1992; Gao and McKinley, 1994; Froehlich et al., 2019; Lehahn et al., 2016), as far as we are aware it has not been evaluated using an Earth system model (ESM). ESM-based assessments are required for studying the response of the global carbon cycle to such perturbations and for estimating the efficacy in a global carbon cycle context

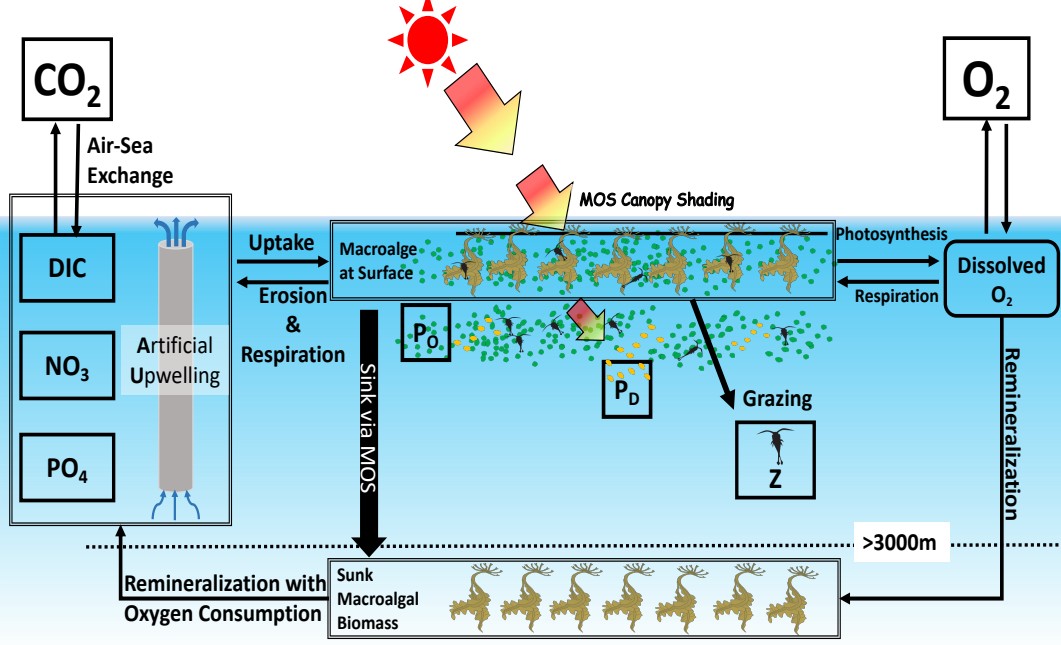

**Figure 1.** Schematic illustrating the biogeochemical fluxes and physical impacts of MOS on nutrients ($NO_3$ & $PO_4$), oxygen, dissolved inorganic carbon (DIC), ordinary phytoplankton ($P_O$ in green), diazotrophs ($P_D$ in pale brown) & zooplankton (Z).

(with regards to atmospheric $CO_2$ removal) of such methods. Furthermore, such models can dynamically simulate macroalgae growth, the permanence of carbon storage (i.e., the fate of sunk biomass on the seafloor), as well as their interactions with global marine biogeochemistry. It is essential to clarify these issues before any decisions about eventual implementation of

MOS can be made.

## 2 Methods

### 2.1 Model description

In this study we employ the University of Victoria Earth System Climate Model (UVic ESCM) version 2.9 (Weaver et al. (2001); Eby et al. (2009); Keller et al. (2012)), which consists of three dynamically coupled components: a three-dimensional

ocean circulation model (Pacanowski, 1996) including a dynamic–thermodynamic sea-ice model (Bitz and Lipscomb, 1999), a terrestrial model (Meissner et al., 2003; Weaver et al., 2001) and a simple one-layer atmospheric energy–moisture balance model (Fanning and Weaver, 1996). The model has a fully coupled carbon cycle including dynamic terrestrial, atmospheric and oceanic carbon inventories. The horizontal resolution of all components is 3.6° longitude × 1.8° latitude, and the ocean component has 19 vertical layers. The descriptions of air–sea gas exchange and seawater carbonate chemistry are based on the

Ocean Carbon Cycle Model Intercomparison Project (OCMIP) abiotic protocol (Orr et al., 1999). The ocean biogeochemistry

is presented with a nutrients-phytoplankton-zooplankton-detritus (NPZD) model that includes one general phytoplankton, diazotrophs, and one zooplankton type (Keller et al., 2012; Eby et al., 2013). The UVic ESCM has been evaluated in several recent studies (e.g. Keller et al. (2014); Mengis et al. (2016); Reith et al. (2016); Kvale et al. (2021)).

## 2.2 Modelling MOS in the UVic ESCM

In this study, the modelling of macroalgae is done with a macroalgae growth model coupled into the UVic ESCM. The aim of the macroalgae model is to investigate the carbon sequestration capacity of MOS as well as the potential impacts on marine biogeochemistry. In the macroalgae model, the net growth rate is affected by several limiting factors, including nutrients, temperature, and solar radiation intensity. The cellular C:N:P ratio of macroalgae is fixed. The loss of macroalgal biomass includes erosion and grazing by zooplankton. The deployment of MOS is done with an algorithm considering spatial and
temporal conditions.

The macroalgae model is also connected to global marine biogeochemical processes, including the inorganic carbon and nutrient pools. In the surface layers, it impacts on phytoplankton via nutrients competition and canopy shading. The single, aggregated zooplankton compartment of the biogeochemical model,which represents higher trophic levels, is also designed to graze on macroalgae. In the bottom layers, the remineralization of sunk macroalgal biomass will consume the dissolved
oxygen, which in turn limits the rate of remineralization.

### 2.2.1 Macroalgae model

The macroalgae model is an idealized generic model of genus *Laminaria* and *Saccharina*, mainly based on Martins and Marques (2002) and Zhang et al. (2016). The rate of biomass change is governed by Eq. 1 as the imbalance of **NGR** (net growth rate, $d^{-1}$) and **LR** (loss rate, fraction of daily biomass loss due to mortality, erosion and grazing by zooplankton, $d^{-1}$).
Modelled macroalgae is seeded 5 meters underwater, considering the light requirement and reduction of damaging risks (Eq. 11). The deployment of macroalgae considers ambient nutrients availability and avoidance of winter periods (Sect. 3.1).

$$\frac{dBiomass}{dt} = (NGR - LR) \times \text{Biomass} \tag{1}$$

The NGR is regulated by:

$$NGR = R_{growth} - R_{resp} \tag{2}$$

where $\mathbf{R}_{growth}$ is the gross growth rate ($d^{-1}$) and $\mathbf{R}_{resp}$ is the respiration rate ($d^{-1}$). The growth rate of macroalgae ($\mathbf{R}_{growth}$) is given by Eq.3, regulated by water temperature (**T**), solar irradiance (**I**) and dissolved nutrient concentrations ($NO_3$ and $PO_4$, **NP**).

$$R_{growth} = \mu_{max} \times f(T_w) \times f(NP) \times f(I_{ma}) \tag{3}$$

In the current model the macroalgal growth rates are controlled by external concentrations of available nutrients via assumed Michaelis-Menten kinetics with half-saturation constants $K_N$ and $K_P$ for $NO_3$ and $PO_4$, respectively:

$$f(N) = \frac{NO_3}{K_N + NO_3} \tag{4}$$

$$f(P) = \frac{PO_4}{K_P + PO_4} \tag{5}$$

$$f(NP) = Min\{f(N), f(P)\} \tag{6}$$

The need for iron is not considered in our macroalgae growth model. Although iron is utilized during macroalgae growth (e.g., Suzuki et al. (1995); Kuffner and Paul (2001)), iron limitation on macroalgae is not widely discussed, especially for the genus *Saccharina*. Besides, as iron is a micronutrient needed in low quantities, the MOS platform could be designed with an iron supply for the macroalgae, in which case MOS could be considered to include a targeted variant of the ocean iron fertilization concept.

The temperature limiting factor used here is an optimum curve following Bowie et al. (1985). $\mathbf{T}_{opt}$ is the species-specific optimum temperature at which the growth rate is maximized. $\mathbf{T}_{max}/\mathbf{T}_{min}$ define the upper/lower temperature limit above/below which macroalgae growth ceases. The temperature optimum curve of the macroalgae is shown in Fig A3.

$$f(T_w) = e^{-2.3 \times X_T^2} \tag{7}$$

$$X_T = \frac{T_w - T_{opt}}{T_x - T_{opt}} \tag{8}$$

$$T_x = \begin{cases} T_{min} & if & T_w \leq T_{opt} \\ T_{max} & if & T_w > T_{opt} \end{cases} \tag{9}$$

Respiration is described by an Arrhenius function considering water temperature $\mathbf{T}_w$ in degrees Celsius (Duarte and Ferreira, 1997; Martins and Marques, 2002):

$$R_{resp} = R_{max20} \times r^{(T_w - 20)} \tag{10}$$

where **R_max20** is the maximum respiration rate at 20 °C of the simulated macroalgae species, **r** stands for the empirical coefficient for macroalgae respiration (Tab. 1).

The limiting factor of solar irradiance density for macroalgae photosynthesis ($f(I_{ma})$) is given in Eq.11 (Steele's photo-inhibition relationship, Kirk (1994)).

$$f(I_{ma}) = \frac{I_{ma}}{I_{opt}} \times e^{(1 - \frac{I_{ma}}{I_{opt}})} \tag{11}$$

where $I_{ma}$ stands for the shortwave radiation intensity reaching the depth $Z$ (given by Eq.12), $I_{opt}$ for the optimum light intensity for macroalgae growth (constant, Tab.1).

$$I_{ma} = I_s \times e^{(-k_w Z_m - \int_0^{Z_m}(P_o + P_D)k_c \times dZ_m)} \times df \tag{12}$$

Eq.12 calculates the shortwave radiation ($I_{ma}$) reaching the depth $Z_m$. This is modified from Keller et al. (2012) and Schmittner et al. (2005), with $Z_m$ denoting the depth of MOS macroalgae platforms beneath the water surface. $Z_m$ is assumed as 5 meters, compromising the empirical depth with sufficient light for macroalgae photosynthesis (1m to 2m for cultivation (Buck and Buchholz, 2004), 0m to 10m for wild communities (Eriksson and Bergström, 2005)) and the depth to reduce the risks of damaging by stressful turbulence or severe weather events (e.g. hurricanes). $df$ denotes the day length as a fraction of 24 hours. $I_s$ stands for the shortwave radiation density at the top of the layer. $P_O$ and $P_D$ are biomass of ordinary phytoplankton and diazotrophs, respectively, in the layers above the macroalgae. $k_w$ is the light attenuation coefficient for water. $I_{opt}$ is the optimum light intensity for macroalgae growth. $k_c$ is the light attenuation coefficient of phytoplankton and also accounts for co-varying particulate and dissolved inorganic and organic materials (Kvale and Meissner, 2017). As described in Sect.2.2.1, the morphology of the frond will not be considered, the self-shading effects by fronds are not considered here (Duarte and Ferreira, 1997; Brush and Nixon, 2010).

The loss rate $LR$ is regulated by:

$$LR = ER + Graze_{ma} \tag{13}$$

$$ER = Biomass \times R_{erosion} \tag{14}$$

$$Graze_{ma} = \mu_Z^{max} Z \times \psi_{ma} \times Biomass \tag{15}$$

where the erosion of biomass (ER) is controlled by the individual erosion rate $R_{erosion}$. As the frond morphology of macroalgae is not modelled here, we set the $R_{erosion}$ as a constant independent of physical impacts (Trancoso et al., 2005; Zhang et al., 2016). The eroded macroalgal biomass will be directly converted back to nutrients and DIC (dissolved inorganic carbon) according to the macroalgae stoichiometry ratios without remineralization or further degradation by zooplankton. This parameterization of erosion, a small biomass loss of 0.01% per day, pragmatically set as instantaneous remineralization rather than introducing another finite remineralization and finite sinking parameterization with difficult-to-constrain parameters. It was used to minimize the computational expense of the model and avoid having to add another state variable that is subjected to physical transport.

$Graze_{ma}$ is the biomass loss due to grazing by zooplankton. Z is the zooplankton biomass which is calculated by the NPZD model. $\mu^{max}_Z$ stands for the maximum potential growth rate of zooplankton defined in Keller et al. (2012, Eq.28). The zooplankton grazing preference on macroalgae ($\psi_{ma}$) is defined in Sect.2.2.3.

For simplicity and to limit the number of state variables, we made the following modifications to the macroalgae model:

1. We did not include a dynamic C:N:P ratio or a representation of luxury nutrient uptake and storage (Broch and Slagstad, 2012; Hadley et al., 2015). Instead, the C:N:P ratio of the macroalgae biomass was set as a constant (Tab.1), which is based

on seasonally averaged measurements of the biomass composition of these genus (Zhang et al., 2016; Martins and Marques, 2002).

2. The macroalgae life cycle processes (e.g. alternations of generations) are also not considered in our model (Brush and Nixon, 2010; Trancoso et al., 2005; Duarte and Ferreira, 1997). We thus assumed that the plantlet (e.g. sporophytes for *Saccharina*) will be reseeded annually on the MOS infrastructure. The assumed deployment strategy, i.e., timing of seeding and sinking, of MOS is latitude-dependent according to the seasonality of solar irradiance (see Sect.3.1). Whenever conditions are unfavorable for macroalgae and no growth occurs during an annual cycle, no re-seeding of macroalgae will occur in these regions.

The parameterization of DOC release by macroalgae could not be included because of the lack of enough information. Few studies exist and the uncertainties about the release of DOC are large. For example, Barrón et al. (2014) reported a release of DOC by macroalgae from a few species of $23.2 \pm 12.6$ mmol C m$^{-2}$ d$^{-1}$ with no information on bioavailability. Meanwhile, refractory DOC dynamics are difficult to include in a global Earth system model and beyond the scope of this study (Anderson et al., 2015; Mentges et al., 2019; Zakem et al., 2021). Thus, the DOC release from macroalgae is not included in this study.

### 2.2.2 Remineralization of sunk macroalgal biomass

Biomass sinking is simulated by instantly transferring the macroalgal biomass from the surface grid cell to the deepest grid cell at the respective location at the end of each cultivating period. This assumes that the harvested biomass could be engineered to sink to the seafloor in a rapid and efficient manner with no remineralization along the way. Afterwards the next macroalgae generation will start to grow in the surface layer. Eq.(16) calculates the temperature dependent remineralization rate of sunk macroalgal biomass ($\mu_{ma}$) following the function of remineralization of detritus in the UVic ESCM (described in Schmittner et al. (2008, Eq.A16)). Remineralization consumes oxygen and returns DIC, PO$_4$ and NO$_3$ from the sunk macroalgal biomass to the sea water, and is described as

$$\mu_{ma} = \mu_{ma_0} exp(T_w/T_b)[0.65 + 0.35 tanh(O_2 - 6)] \tag{16}$$

where $\mu_{ma0}$ is the remineralization rate of sunk macroalgal biomass at 0°C. $\mathbf{T}_w$ and $\mathbf{T}_b$ represent the sea water temperature and e-folding temperature of biological rates, O$_2$ is the dissolved oxygen concentration in mmol m$^{-3}$. When the dissolved oxygen is insufficient ($<5$ mmol m$^{-3}$), aerobic remineralization will be replaced by oxygen-equivalent, but slower, denitrification via reduction of NO$_3$ (Keller et al., 2012). Note that remineralization will cease when also NO$_3$ is completely consumed.

There are considerable uncertainties concerning the fate of sunk macroalgae (Sichert et al., 2020; Krause-Jensen and Duarte, 2016; Luo et al., 2019). A sensitivity simulation explores the situation where $\mu_{ma_0}$ is set to zero, which would assume permanent deposition of the sunk biomass on (or in) the seafloor without decaying.(Sect.3.2).

### 2.2.3 Interactions with pelagic microbial ecosystems

Besides the competition for nutrient resources, the macroalgae canopies may also reduce downward solar irradiance ('canopy shading') and thus limit phytoplankton photosynthesis beneath the macroalgae (Jiang et al., 2020)). Eq.(17, modified from

Eq.14, (Keller et al., 2012)) describes the shortwave radiation attenuation ($I_{phyt}$) through the macroalgae layer as well as phytoplankton and water (MOS is not deployed in areas covered by sea ice):

$$I_{phyt} = I_s \times e^{-k_w Z - \int_0^Z (P_O + P_D)k_c \times dZ - k_{ma} \times h_{ma} \times Biomass} \tag{17}$$

where $k_{ma}$, the macroalgae light extinction coefficient ($m^{-1}$), is calculated based on the biomass of macroalgae in carbon as:

$$k_{ma} = a_{ma} \times Biomass \times MR_{C:N} \tag{18}$$

Here $a_{ma}$ is the macroalgae carbon specific shading area ($m^2$ $kgC^{-1}$, Trancoso et al. (2005)), $h_{ma}$ is the thickness of macroalgae layer, $MR_{C:N}$ stands for the molar C:N ratio of macroalgal biomass.

The original NPZD model in Keller et al. (2012) is extended by allowing zooplankton to graze on macroalgae. Our as-
sumption that zooplankton can graze on macroalgae is based on the notion that the marine biogeochemical component of "zooplankton" in UVic ESCM represents all higher trophic levels, including known macroalgae grazers such as amphipods (Jacobucci et al., 2008), gastropods (Chikaraishi et al., 2007; Krumhansl and Scheibling, 2011), sea urchins (e.g. Yatsuya et al. (2020)) and fishes (e.g. Peteiro et al. (2014)). Thus, we included this food web pathway to assess the sensitivities of macroalgae to potential grazers in the ocean, assuming that with large macroalgae farms the pelagic larva of some grazing organisms like
fish or urchins, would settle within the farms.

The grazing preference for macroalgae ($\psi_{ma}$) is set to $1 \times 10^{-4}$ according to observational studies (Trancoso et al., 2005). Macroalgae thus provide a grazing option for zooplankton in addition to the traditional NPZD-type model food sources (phytoplankton, diazotrophs, detritus and zooplankton via self-grazing). Therefore, the four original grazing preferences (0.3 on phytoplankton, 0.1 on diazotrophs, 0.3 on detritus and 0.3 on zooplankton (Keller et al., 2012, Tab.1)) are reduced by $\frac{1}{4}$ $\psi_{ma}$
each. In the areas where MOS is absent (i.e, in the ice-covered ocean surface), the zooplankton grazing will follow the original description in Keller et al. (2012, Tab. 1) without the preference for macroalgae. No $CaCO_3$ formation by macroalgae is simulated here (Bach et al., 2021; Macreadie et al., 2017, 2019), as calcareous macroalgae species and epibiont calcifiers are not considered. Therefore, the only alkalinity impact of growing and remineralizing macroalgae comes via changes in nitrate and phosphate.

### 2.2.4    Mass conversions

In order to parameterize and validate the model, it is necessary to convert from commonly measured macroalgae variables (often in wet and dry weight units) to the model units. These conversions include: the calculation of carbon and $CO_2$ sequestered in macroalgal biomass ($C_{ma}$, gram carbon), as well as the conversions of dry weight (DW, gram) and wet weight (WW, gram)):

$$C_{ma} = Biomass \times MR_{C:N} \times 12.011 \tag{19}$$

$$\text{CO}_{2\text{ma}} = C_{ma} \times 3.67 \tag{20}$$

$$DW = C_{ma} \div MR_{C:DW} \tag{21}$$

$$WW = DW \times MR_{DW:WW} \tag{22}$$

where 3.67 is the ratio between the atomic mass of $CO_2$ (44 g/mol) to carbon (12g/mol), *Biomass* is in moles of nitrogen, 12.011 is the relative molecular weight of carbon (g/mol).

### 2.2.5 MOS carbon retained in the ocean and outgassing

The DIC from remineralization of sunk biomass will eventually be conveyed back to the ocean surface and may leak back to the atmosphere. Eq.23 calculates the ocean-retained fraction (FR, %) of MOS-captured carbon (MOS-C), where the $C_{captured}$ is carbon in cumulative sunk biomass, $C_{SunkBiomass}$ is the carbon in sunk macroalgal biomass that still remains on the seafloor.

$$FR = \frac{C_{retained}}{C_{captured}} = \frac{(\text{DIC}_{remineralized} + C_{SunkBiomass})}{C_{captured}} \tag{23}$$

In order to track the leakage of MOS-C after remineralization, a tracer of remineralized MOS-C (MOS_DIC) is added to the UVic ESCM aside of the original DIC tracer. MOS_DIC participates in the inorganic ocean carbon cycle Weaver et al. (2001, Section 3e). When reaching the surface, the outgassing of MOS_DIC will follow the air-sea gas exchange process in UVic ESCM, which is given in Weaver et al. (2001, Section 3e). The air-sea exchange flux of MOS-C is also calculated for analysing the location and quantity of outgassing. The results of MOS-C outgassing are shown in Sect. 4.6.

## 3 Experiment design

The UVic ESCM is spun up for > 10,000 years to an equilibrium state under pre-industrial (year 1765) atmospheric and astronomical boundary conditions, and is then integrated for another 250 years without prescribing atmospheric $CO_2$ concentrations to allow the carbon cycle to equilibrate. Afterwards, the model is run from 1765 until 2005 and forced with historical fossil-fuel emissions and land-use changes (crop and pastureland) (Keller et al., 2014). From year 2005 to 2100, simulations are forced with $CO_2$ emissions represented as a direct adjustment to radiative forcing, land use change by agriculture, volcanic radiative forcing and sulphate aerosols which are prescribed according to the Representative Concentration Pathway 4.5 (RCP4.5, Meinshausen et al. (2011); Thomas (2014); Keller et al. (2014); Partanen et al. (2016)). Solar insolation at the top of the atmosphere, wind stress, and wind fields are varied seasonally. After the year 2300, $CO_2$ emissions are assumed to decrease linearly until the end of year 3000 with other forcing held constant.

The full list of simulations is given in Tab. 3. To test the maximum potential as well as the global carbon cycle and biogeochemical responses, we simulate MOS for 1,000 years beginning in year 2020 (MOS_Conti). Additionally, termination

**Table 1.** Model parameters

| Symbol | Parameter | | Unit | Value | Reference |
|---|---|---|---|---|---|
| $a_{ma}$ | Macroalgae carbon specific shading area | | $m^2 kgC^{-1}$ | 11.1 | Trancoso et al. (2005) |
| d | Distance between the cultivating ropes | | m | 10 | Van Der Molen et al. (2017) |
| $R_{erosion}$ | Individual erosion rate | | $\% \, d^{-1}$ | 0.01 | Zhang et al. (2016) |
| $I_{opt}$ | Optimum light intensity for macroalgae growth | | $W \, m^{-2}$ | 180 | Zhang et al. (2016) |
| $NO_3$ | Nitrate Concentration | | $\mu mol \, l^{-1}$ | model calculation | Keller et al. (2012) |
| $PO_4$ | Phosphate Concentration | | $\mu mol \, l^{-1}$ | model calculation | Keller et al. (2012) |
| $K_N$ | Half-saturation constant for nitrogen uptake | | $\mu mol \, l^{-1}$ | 2 | Zhang et al. (2016) |
| $K_P$ | Half-saturation constant for phosphorus uptake | | $\mu mol \, l^{-1}$ | 0.1 | Zhang et al. (2016) |
| $k_w$ | Coefficient of light attenuation through water | | $m^{-1}$ | 0.04 | Keller et al. (2012) |
| $k_c$ | Coefficient of light attenuation through phytoplankton | | $m^{-1}(mmol \, m^{-3})$ | 0.047 | Keller et al. (2012) |
| $M_{ma}$ | Thickness of MOS macroalgae canopy | | m | 10 | Trevathan-Tackett et al. (2015) |
| $MR_{C:N}$ | Molar C:N ratio of macroalgal biomass | | - | 20 | Atkinson and Smith (1983) |
| $MR_{P:N}$ | Molar P:N ratio of macroalgal biomass | | - | 0.05 | Atkinson and Smith (1983) |
| $MR_{C:P}$ | Molar C:P ratio of macroalgal biomass | | - | 400 | calculated |
| $MR_{DW:WW}$ | Ratio of DW to WW of macroalgal biomass | | - | 0.1 (values reported:0.05~0.2) | Aldridge and Trimmer (2009); Conover et al. (2016); Van Der Molen et al. (2017) |
| $MR_{C:DW}$ | Carbon content of dried macroalgal biomass | | % | 30 | Chung et al. (2011) |
| $MR_{N:DW}$ | Nitrogen content of dried macroalgal biomass | | - | 0.16 | Duarte et al. (2003) |
| $R_{max20}$ | Maximum respiration rate at 20°C | | $\% \, d^{-1}$ | 1.5 | Martins and Marques (2002); Zhang et al. (2016) |
| r | Empirical coefficient for macroalgae respiration | | $d^{-1}$ | 1.047 | Martins and Marques (2002) |
| Seed | Initial macroalgal biomass | per kilometer cultivating line | $kgC \, km^{-1}$ | 2.5 | |
| | | concentration of N | $mmol \, N \, m^{-3}$ | 0.02 | Van Der Molen et al. (2017) calculated |
| $T_b$ | E-folding temperature of biological rates | | °C | 15.56 | Schmittner et al. (2008) |
| $T_{opt}$ | Optimum temperature for growth | | °C | 20(values reported: 13-30) | Zhang et al. (2016); Martins and Marques (2002) |
| $T_{max}$ | Upper temperature limit above which growth ceases | | °C | 35 | Breeman (1988) |
| $T_{min}$ | Bottom temperature limit below which growth ceases | | °C | 0 | Martins and Marques (2002) |
| $u_{max}$ | Maximum growth rate | | $d^{-1}$ | 0.2 | Zhang et al. (2016) |
| w | Areal mean artificial upwelling rate | | $cm \, d^{-1}$ | 1 | Oschlies et al. (2010b) |
| $Y_{max}$ | Maximum yield of macroalgal biomass on MOS | | $t \, DW \, km^{-2}$ | 3300 | |
| $\psi_{ma}$ | Zooplankton grazing preference on macroalgae | | - | $1 \times 10^{-4}$ | Trancoso et al. (2005) |
| $\mu_{ma0}$ | Remineralization rate of sunk macroalgal biomass at 0 °C | | $\% \, d^{-1}$ | 7 | Partanen et al. (2016) |

experiments (MOS_Stop) are performed to analyze the response of the ocean and climate to an abrupt termination of MOS at year 2100.

## 3.1 Deployment strategies of MOS

The current study focuses on estimating the maximum carbon sequestration potential of MOS, and assumes instantaneous seeding on floating infrastructure in the open ocean. The macroalgae is represented as a biogeochemical tracer (Eq 1) that is not subject to physical transports and remains fixed in the top (1st) ocean layer of the UVic ESCM, which is assumed to be well mixed. In our idealized experiments, MOS deployment must fulfill the following requirements:

- the water depth must be $\geq$ 3,000m: according to the assessment by Reith et al. (2016), leakage of dissolved inorganic carbon added to deep waters (in this case from remineralization of sunk macroalgal biomass) is small at such depths compared to shallower ones;

- the ambient surface $NO_3$ concentration is greater than *Seed* plus $K_N$ (Tab. 1); this ensures sufficient nutrients for initial growth as *Seed* is directly transferred from dissolved $NO_3$ and $K_N$ is the half saturation constant for $NO_3$ uptake. Note that in this calculation, *Seed* has been converted from the unit of kgC km$^{-1}$ in Tab. 1 the unit of concentration of Nitrogen (mmol N m$^{-3}$);

- spatially located between 57°N and 72°S to remain in sea ice free waters.

Note that the DIC, N and P components of the initial *Seed* are directly removed from the inorganic matter pool of the respective grid box in order to maintain model mass balance and avoid adding extra nutrients/carbon to the ocean at the time of seeding.

During the MOS simulations, seasonality of temperature as well as solar radiation are essential limiting factors of the primary productivity of MOS in various latitudinal regions. In order to avoid the unnecessary loss of macroalgal biomass during winter periods when solar radiation is insufficient and the ambient water temperature is low, we partitioned the global ocean surface into three belts (N, M and S) and pragmatically applied farming strategies according to Tab. 2. The period between the seeding and sinking of macroalgae is set as six months from May to October in belt N and from November to the next April in belt S. In belt M, the macroalgae is seeded at the beginning of the year and sinking occurs after 12 months. The geographical locations of the three belts are shown in Fig. 2.

**Table 2.** Latitudinal division of MOS deployment regions

| Belt | Latitudinal range | Date for | |
|:---:|:---:|:---:|:---:|
| | | Seeding | Sinking |
| N | 51.3°N to 17.1°N | 01.May | 31.Oct |
| M | 17.1°N to 18.9°S | 01.Jan | 31.Dec |
| S | 18.9°S to 56.7°S | 01.Nov | 30.Apr* |

*In the following year

The maximum yield of *Biomass* is set to a constant value of $Y_{max}$ (Tab.1). When the biomass reaches $Y_{max}$ in a grid cell, macroalgae will stop growing and wait for sinking. After an annual cultivation cycle, the macroalgal biomass is instantaneously delivered to the seafloor apart from a small fraction (equivalent to *Seed*) that remains at the surface for re-seeding. In some regions where conditions are unfavorable and no net macroalgae growth had occurred during the last cultivation period, the total

305 *Biomass* will be sunk once without any further re-seeding. In order to prevent MOS from removing too much atmospheric $CO_2$ in long-term simulations where emissions eventually reach zero, MOS deployment will be terminated once atmospheric $CO_2$ concentration hits 280ppm, assuming that there is no need for more CDR once pre-industrial $CO_2$ values have been reached.

## 3.2 Sensitivity Studies

As test simulations indicated that the CDR potential of MOS is in many ocean regions limited by the availability of nutrients

in the surface layer, sensitivity simulations were performed with MOS combined with artificial upwelling (AU) that pumps up nutrient-rich deeper waters to the surface and thereby relaxes nutrient stress and enhances the macroalgae growth.

The simulated MOS-AU system is based on Oschlies et al. (2010b) and Keller et al. (2014). We placed modelled 'pipes' that pump deeper water to the ocean surface in areas where MOS is deployed. The simulated upwelling works by transferring water adiabatically from the grid box at the lower end of the pipe to the surface grid box at a rate of 1 cm day$^{-1}$. These pipes

will function continuously until the termination of MOS (in year 2100 or 3000). However, because these earlier studies have revealed a dominant effect associated with low temperatures of the upwelled colder waters, we here concentrate on the nutrient aspect and simulate a hypothetical MOS-AU system that keeps temperatures at ambient levels (e.g. via heat exchangers).

In the simulated AU system, water together with dissolved tracers is transferred from the grid box at the lower end of the pipes to the surface grid box resulting in a model grid box-average upwelling rate (w, set to 1cm/day, Tab.1). The lower end

of the pipes is fixed at a depth of 1000m. Similar to the normal MOS simulations without AU, the MOS_AU simulations are deployed from year 2020 and then terminated at either year 2100 in discontinuous run or year 3000 in continuous one (Tab. 3).

The MOS-AU joint system is deployed using the following strategies: AU pipes will be deployed everywhere with depth $\geq$ 3000m and start upwelling immediately. If surface nutrient concentrations are raised to the initial seeding condition (Sect.3.1) in any grid box, MOS will be deployed, thereby expanding the range where MOS can grow.

Another model parameter selected for sensitivity studies is the remineralization rate of sunk macroalgal biomass ($\mu_{ma}$, Eq. 16). $\mu_{ma}$ is a critical factor impacting the residence time of MOS-captured carbon in the ocean and associated benthic oxygen consumption by remineralization. Macroalgal biomass has been reported to be recalcitrant to microbial degradation, however, the fate of macroalgal biomass in the deep sea is uncertain (Krause-Jensen and Duarte, 2016; Luo et al., 2019; Sichert et al., 2020). Thus, an extreme and idealized situation with $\mu_{ma}$ set to zero is tested in sensitivity simulations (MOS_NoRe). This

can be thought of as a case where all biomass is permanently buried upon reaching the seafloor. This sensitivity study can also simulate an extreme case of infinitely slow remineralization, which can help estimating the range of possible fates of remineralized organic matter. Meanwhile, this sensitivity study also represents a different macroalgae farming approach - that of harvesting the biomass to create bioenergy with carbon capture or storage (BECCS) or biochar (e.g., Kerrison et al. (2015); Laurens et al. (2020); Roberts et al. (2015)), with the assumption that all harvested biomass was permanently removed from the

ocean. While this is a very idealized case, it serves the useful purpose of providing information on how marine biogeochemistry is impacted by the permanent removal of fixed C, N, and P.

The stoichiometric C:N ratio of macroalgal biomass ($MR_{C:N}$) may also influence the CDR capacity of MOS. In the current study, the $MR_{C:N}$ (400:20, Tab. 1) is set as 20, nearly 2 times higher than the phytoplankton stoichiometric biomass C:N ratio in the UVic ESCM (C:N=106:16, the Redfield ratio). However, the difference between the macroalgae and phytoplankton sto- ichiometric C:N ratio may have strong influences on the CDR potential of MOS. For instance, Bach et al. (2021) has indicated that the CDR potential of floating macroalgae (*Sargassum*) belt may be reduced by 7% to 50% due to the nutrient reallocation caused by the variation of gaps of C:N between macroalge and phytoplankton. Thus, sensitivity experiments of macroalgal C:N ratios (MOS_Conti_CN$_{High}$ and MOS_Conti_CN$_{Low}$) have been performed to investigate the impacts of $MR_{C:N}$ on the MOS CDR capacity.

**Table 3.** Description of the Model Experiments. "Stop" represents the termination of the simulation in year 2100; "Conti" represents the continuous MOS deployment till year 3000; "AU" represents artificial upwelling; NoRe represents zero-remineralization of sunk macroalgal biomass; CN represents the molar C:N ratio of macroalgal biomass ($MR_{C:N}$ in Tab. 1).

| Category | Experiment | Description |
|---|---|---|
| Normal MOS simulations | **Control_RCP4.5** | Control simulation under RCP4.5 |
| | **MOS_Conti** | As Control_RCP4.5, but MOS deployed from year 2020 to year 3000 |
| | **MOS_Stop** | As Control_RCP4.5, but MOS implemented from year 2020 to year 2100 |
| Sensitivity simulations | **MOS_Conti_NoRe** | As MOS_Conti, but with zero remineralization rate |
| | **MOS_Stop_NoRe** | As MOS_Stop, but with zero remineralization rate |
| | **MOS_AU_Conti** | MOS synergy with AU, area-averaged upwelling velocity(w) is 1cm/day. |
| | **MOS_AU_Stop** | As MOS_AU_Conti, but MOS implemented until year 2100 |
| | **MOS_Conti_CN$_{High}$** | As MOS_Conti, but the $MR_{C:N}$ increases by 20% from 20 to 24. |
| | **MOS_Conti_CN$_{Low}$** | As MOS_Conti, but the $MR_{C:N}$ decreases by 20% from 20 to 16. |

## 4 Results

### 4.1 Evaluation of MOS

To evaluate if the simulated MOS systems have plausible macroalgae growth characteristics, we evaluate the seasonal dynamics of the simulated MOS system for a 30 days averaged time slice from 2020 to 2024 under the RCP4.5 emission scenario and without artificial upwelling.

### 4.1.1 Distribution of MOS

The red contours in Fig.2a delineate the occupied area that basically follows the pattern of the simulated $NO_3$-rich ocean surface (Keller et al. (2012, Fig 9), Garcia et al. (2010, WOA2009 Dataset)) in the Northern and the equatorial Eastern Pacific, as well the Southern Ocean. Except for the coastal regions and Arctic areas which are not considered for MOS here, the distribution pattern of MOS agrees with the other estimation of potential open-ocean macroalgae farming locations (e.g. Lehahn et al. (2016, Fig.2.A),Froehlich et al. (2019, Figure 1.)). Another powerful limiting factor is the ocean surface temperature (Garcia et al., 2010, WOA2009 Dataset) which is too warm in many places for our idealized species, i.e., temperatures are above the $T_{opt}(20°C)$ and nearly reach $T_{max}(35°C)$.

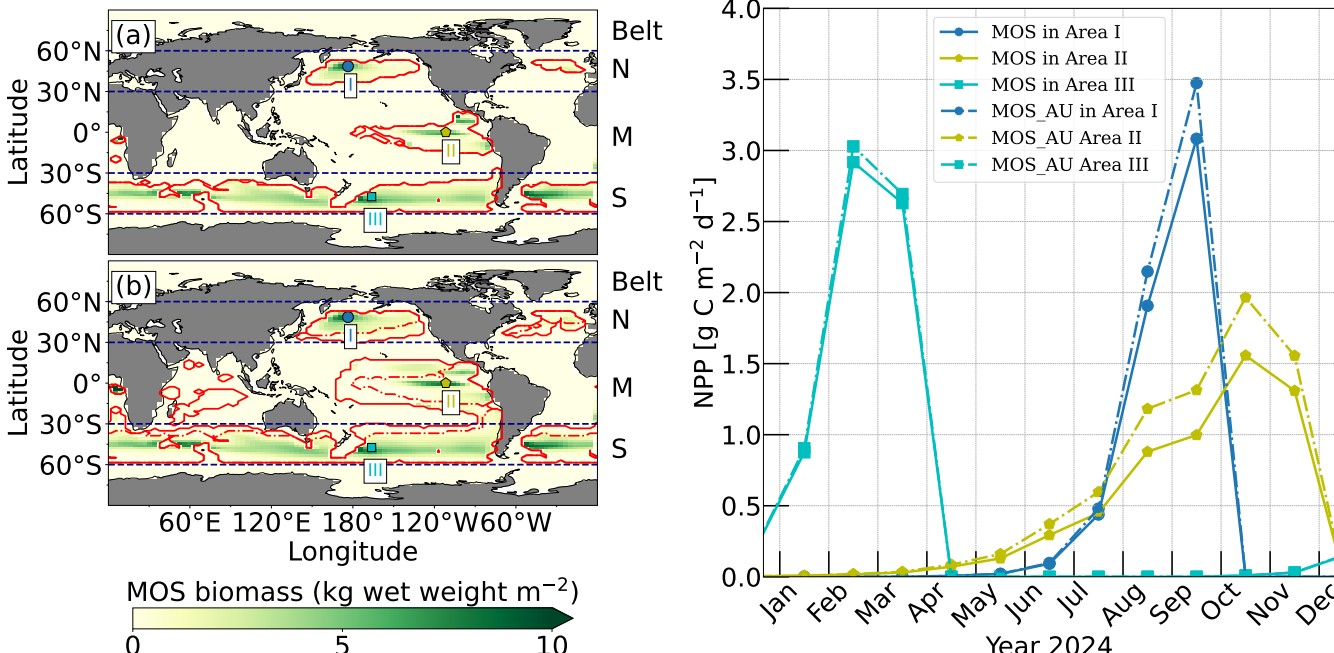

**Figure 2.** Annual vertically integrated macroalgae biomass of MOS (**a**) and MOS_AU (**b**) in year 2024. Red solid lines outline the MOS occupied area at year 2024 in both, while red dashed lines outline the initial MOS seeding area at year 2020 in (**a**). The simulated MOS area generally covers the $NO_3$-rich ocean surface (**a**) and can be expanded with nutrients supplemented by AU (**b**), larger than the estimated adequate area for macroalgae in previous studies (Lehahn et al., 2016; Froehlich et al., 2019)). Results for areas I (blued circle), II (yellowish pentagon) and III (cyan rectangle) are discussed in the text and displayed in Fig 3. Braces indicate the belts of N, M and S with various seeding strategies of macroalgae (Tab. 2), which are designed to avoid winter periods.

**Figure 3.** Vertically integrated macroalgae NPP simulated by experiment MOS (solid lines) for year 2024 and representative Areas I (dark blue circle), II (yellow pentagon) and III (cyan rectangle) highlighted by rectangles with corresponding colors in Fig.2. The NPP of macroalgae of experiment MOS_AU (dashed) shows an enhancement of NPP as expected.

At the beginning of year 2020, a surface area $S_{MOS}$ of $72 \times 10^6$ km$^2$ was selected by the MOS algorithm according to the requirements described in Sect.3. This is equivalent to a total cultivated rope length ($L_{MOS}$) of $7.2 \times 10^6$ km (Eq.A1, Sect.A1). When the macroalgae start to grow and to consume nutrients, regions with nutrient levels insufficient for further growth are gradually abandoned. By the end of year 2024, the MOS coverage has declined by about 3% to $69.6 \times 10^6$ km$^2$ (Tab.4).

Despite the similar distribution patterns, MOS occupied area ($69.6 \times 10^6$ km$^2$, $\sim$19.7% of the world ocean) is larger than the assessments of $\sim$10% of the world ocean by Lehahn et al. (2016) and $\sim$48 $\times 10^6$ km$^2$ by Froehlich et al. (2019). Compared to the static estimation based on historical nutrient levels and temperature suitability (Lehahn et al. (2016, Fig.2.A)), the dynamic processes redistributing nutrients as well as the explicit macroalgae growth module in our simulations contribute to

the simulated larger potential area for MOS, especially in the equatorial Eastern Pacific and the Southern Ocean (Tab. 3). Besides, the adequate area for macroalgae cultivation was limited to the Economic Zones (EEZs) by Froehlich et al. (2019) due to limitations of cost and political feasibility. This constraint has been ignored in the current study. Thus, our simulated MOS-adequate area is 45% larger than the estimation by Froehlich et al. (2019).

### 4.1.2 Macroalgae model validation

Validation of the macroalgae model is crucial, as the productivity and macroalgal biomass yield is vital for $CO_2$ sequestration. Here we examine the simulated seasonality, NPP rate, and biomass yield of MOS in comparison with available observations and assessments. Simulated NPP is high in the first year of deployment in many regions because nutrients are abundant, and then sharply declines in the following years as a new local biogeochemical state is reached. Thereafter, NPP gradually reaches a relatively steady state by 2024 (Fig.A2). To provide some validation of the macroalgae model we select three areas named Area I, II and III from Belt N, M and S and analyze their performance in year 2024 (Fig.3). Each area covers 4 grid boxes in the uppermost ocean layer of the UVic ESCM.

According to Sect.3.1 and Tab.2, the seeding date for MOS is 1st of May in Area I, 1st of January in Area II and 1st of November in Area III, while the sinking dates are 31st October, 31st December and 30th in the next April correspondingly. As a result shown in Fig.3, the macroalgae NPP in Area I peaks around September with the accumulation of macroalgae biomass in that area. In Area II, due to the nutrient limitation and nutrient competition with ambient phytoplankton, macroalgae biomass grows slower than in the other two areas, leading to a later peak of NPP around October. In Area III which locates in the Southern Ocean where the nutrients are rich, the macroalgae NPP peaks at February. These results indicates a plausible seasonality of our macroalgae model.

In our simulations, simulated macroalgae NPP rates are comparable to the observed ranges in the productive areas that we selected here. Observed wild macroalgae NPP varies widely, ranging from 91 to 750 gC $m^{-2}$ $yr^{-1}$ (Krause-Jensen and Duarte, 2016). Our model reproduces the macroalgae NPP of 159.2-199.3 gC $m^{-2}$ $yr^{-1}$ in the selected areas (Tab.4). Simulated biomass yields in these areas are in the previously reported range as well. Reports of the biomass yield of aquacultured *Laminaria saccharina* (now regarded as a synonym of *Saccharina latissima*) range from 40t DW $km^{-2}$ $yr^{-1}$ in an off-shore cultivation experiment by Buck and Buchholz (2004) to 456t DW $km^{-2}$ in a coastal cultivation experiment by Peteiro et al. (2014). In our simulations, the yield of selected areas ranges from 492.4 to 648.2t DW $km^{-2}$ $yr^{-1}$. The selected Area I yields 648.2 DW $km^{-2}$ $yr^{-1}$. In regions with similar latitudes as Area I, the biomass yield of aquacultured *Saccharina japonica* (formerly classified as *Laminaria japonica*) was ~300t DW $km^{-2}$ $yr^{-1}$ in China (Zhang et al., 2016) and reached 7,280t DW $km^{-2}$ $yr^{-1}$ in Japan (Yokoyama et al., 2007). Nevertheless, some simulated low values from the globally averaged and latitudinal belt-averaged results are not surprising considering that the open ocean tends to be more nutrient limited than coastal or near-shore regions where the aforesaid observed macroalgae NPP was measured. Our results provide some confidence that our idealized model can simulate macroalgae well enough with respect to typical biomass yield, seasonality and geographical distribution.

Table 4. Properties of globally implemented MOS. Selected areas are from data of year 2024, whereas Belt areas are values averaged from 2020 to 2024. Areal NPP rates and Biomass Yield refer to the respective MOS area. The observational data comes from the references.

| Property | Unit | Observations | Exp. | Selected area(10³ km²) | | | Belt(10⁶ km²) | | | |
|---|---|---|---|---|---|---|---|---|---|---|
| | | | | Area I | Area II | Area III | N | M | S | Global |
| MOS occupied area($S_{MOS}$) | km² | - | MOS | 218.8 | 320.3 | 204.2 | 9.1 | 15.7 | 44.8 | 69.6 |
| | | - | MOS_AU | | | | 17.4 | 44.3 | 64.6 | 126.3 |
| NPP | gC m⁻² yr⁻¹ | 91-750[1] | MOS | 159.2 | 176.9 | 199.3 | 50.8 | 52.0 | 67.5 | 61.8 |
| | | | MOS_AU | 202.2 | 231.1 | 217.7 | 45.5 | 32.2 | 56.9 | 46.7 |
| Biomass Yield | t DW km⁻² yr⁻¹ | 40-456[2] | MOS | 648.2 | 492.4 | 579.7 | 173.2 | 160.3 | 206 | 191.4 |
| | | Area I: 300-7,280[3] | MOS_AU | 715.3 | 615.4 | 597.0 | 142.2 | 85.4 | 169 | 136 |
| Total CO₂ captured in biomass | Pg CO₂ yr⁻¹ | - | MOS | 0.14 | 0.16 | 0.12 | 1.6 | 2.5 | 9.2 | 13.3 |
| | | - | MOS_AU | 0.15 | 0.20 | 0.12 | 2.5 | 3.8 | 10.9 | 17.2 |

[1]Krause-Jensen and Duarte (2016); [2]Buck and Buchholz (2004); Peteiro et al. (2014); [3]Zhang et al. (2016); Yokoyama et al. (2007)

## 4.2 Evaluation of MOS with Artificial Upwelling (AU)

As expected, AU increases the area occupied by MOS from $69.6 \times 10^6$ km² in the run without AU to $129.6 \times 10^6$ km² in the run with AU (Fig.2.2). Obvious expansions of areas with suitable growing conditions are found in the Eastern Tropical Pacific and the North Atlantic. AU also expands S_MOS to the Indian Ocean, which was almost abandoned in regular MOS simulations. In Area I, II and III, both NPP rate and biomass yield are enhanced due to the upwelled nutrients (column Belt, Tab.4 & Fig.3). A closer look into the Belt N, M and S areas shows that both the NPP rate and biomass yield per square-meter of the deployment area decrease in simulation MOS_AU when compared to the standard MOS simulation (column Belt, Tab.4). This is related to a 'dilution effect': in the new adequate areas made accessible for MOS by AU, the available nutrients are limited, thus the MOS NPP is relatively low compared to the original nutrient-rich NPP areas. Despite of this, the expanded MOS area in MOS_AU increases the total CO₂ captured by about ∼30% (Tab.4).

## 4.3 MOS deployment until year 2100

This section showcases the CDR and climate change mitigation capacities of MOS within the 21st century. Impacts of MOS on marine biogeochemistry (nutrients, dissolved oxygen and pelagic ecosystem) and global carbon cycles will also be examined.

### 4.3.1 CDR & climate change mitigation capacities

Over the 80 years between year 2020 and year 2100, MOS is mainly deployed in nutrient-rich regions such as the Southern Ocean and the northern and eastern equatorial Pacific (Fig.B1(a)), although some contraction of initially occupied areas occurred due to the removal of nutrients.

By the year 2100, MOS (MOS_Stop and MOS_Conti) has sequestered 270 PgC (990 $PgCO_2$, Tab.5), representing $\sim$37% of the cumulative $CO_2$ emissions in the RCP4.5 pathway. Essentially all of MOS-captured carbon is retained in the ocean over this period as either remineralized dissolved inorganic carbon, or organic carbon in the sunk biomass.

The CDR capacity of MOS is sensitive to the $MR_{C:N}$ (molar C:N ratio of macroalgal biomass, Tab. 1). Compared to MOS_Conti, the carbon captured by MOS (MOS-C) in MOS_Conti_$CN_{High}$ raises by 22% (by the year 2100) and 19% (by the year 3000) when the $MR_{C:N}$ is increases by 20% to 24. When the $MR_{C:N}$ is 20% lower than the original value, MOS-C decreases by 13% (by the year 2100) and 18% (by the year 3000). Our results agree with the range of CDR potential reduction by nutrient reallocation (7-50%) reported in Bach et al. (2021).

**Table 5.** Model Simulations under the RCP4.5 emission scenario. MOS-C represents the carbon sequestered via MOS. $C_{atm}$, $C_{oc}$, $C_{ter}$ stand for atmospheric, oceanic and terrestrial carbon reservoir respectively. $\Delta$SAT stands for surface averaged temperature relative to 13.18°C, the pre-industrial.

| Experiment | pCO2 (ppm) | | Cumulative CO2 Emission (PgC) | | MOS-C (PgC) | | FR (%) | | $C_{atm}$ (PgC) | | $C_{oc}$ (PgC) | | $C_{ter}$ (PgC) | | $\Delta$SAT (°C) | | Phyt NPP ($\frac{PgC}{yr}$) | |
|---|---|---|---|---|---|---|---|---|---|---|---|---|---|---|---|---|---|---|
| Year | 2100 | 3000 | 2020-2100 | 2020-3000 | 2020-2100 | 2020-3000 | 2100 | 3000 | 2100 | 3000 | 2100 | 3000 | 2100 | 3000 | 2100 | 3000 | 2100 | 3000 |
| Control_RCP4.5 | 573.1 | 615.5 | 718.1 | 1392 | / | / | / | / | 1217 | 1307 | 37611 | 38180 | 1854 | 1935 | 2.52 | 4.32 | 47.6 | 56.8 |
| *Normal MOS experiments minus Control_RCP4.5* | | | | | | | | | | | | | | | | | | |
| MOS_Conti | -67.2 | -297.0 | 718.1 | 1392 | 270.0 | 2533 | 100 | 75.3 | -142.6 | -630.5 | 171.8 | 901.9 | -29.7 | -278.8 | -0.38 | -2.87 | -9.5 | -22.2 |
| MOS_Stop | -67.2 | -28.5 | 718.1 | 1392 | 270.0 | 270.0 | 100 | 58.6 | -142.6 | -60.5 | 171.8 | 77.4 | -29.7 | -16.8 | -0.38 | -0.23 | -9.5 | -1.8 |
| *Sensitivity MOS experiments minus Control_RCP4.5* | | | | | | | | | | | | | | | | | | |
| MOS_AU_Conti | -108.7 | -225.3 | 718.1 | 1392 | 446.8 | 1970 | 99.9 | 72.9 | -230.7 | -452.3 | 283.9 | 665.3 | -53.2 | -186.8 | -0.63 | -2.49 | -5.1 | -13.0 |
| MOS_AU_Stop | -108.7 | -52.3 | 718.1 | 1392 | 446.8 | 446.8 | 99.9 | 64.4 | -230.7 | -111.1 | 283.9 | 143.5 | -53.2 | -32.4 | -0.63 | -0.43 | -5.1 | -3.2 |
| MOS_Conti_NoRe | -67.3 | -310.5 | 718.1 | 1392 | 269.9 | 2008 | 100 | 100 | -142.9 | -659.2 | 171.8 | 964.3 | -29.8 | -305.0 | -0.38 | -3.27 | -9.5 | -15.1 |
| MOS_Stop_NoRe | -67.3 | -54.34 | 718.1 | 1392 | 269.9 | 269.9 | 100 | 100 | -142.9 | -115.4 | 171.8 | 145.0 | -29.8 | -29.6 | -0.38 | -0.40 | -9.5 | -2.9 |
| MOS_Conti_$CN_{High}$ | -79.5 | -335.1 | 718.1 | 1392 | 329.5 | 3011 | 100 | 75.0 | -168.8 | -711.4 | 204.5 | 1059 | -36.3 | -347.4 | -0.46 | -3.61 | -9.2 | -19.1 |
| MOS_Conti_$CN_{Low}$ | -53.6 | -270.7 | 718.1 | 1392 | 235.1 | 2078 | 100 | 79.3 | -113.7 | -574.6 | 136.1 | 817.7 | -22.7 | -229.1 | -0.30 | -2.67 | -9.4 | -22.9 |

In the model MOS thus gradually reduces atmospheric $CO_2$, and thereby also limits global warming with respect to the pre-industrial period ($\Delta$SAT, Fig.4); i.e., the temperature increase of 2.14°C by the year 2100 is 0.38 °C lower than $\Delta$SAT of Control_RCP4.5, but still missing the 2°C target.

When AU is deployed in conjunction with MOS, the CDR capacity and mitigation effects of MOS are enhanced (Figs.4a&c, Fig.5). By the end of year 2100, 446.8 Pg carbon is sequestered by MOS_AU, an increase of 39.5% relative to normal MOS.
Correspondingly, MOS_AU successfully achieves the 2°C target of the Paris Agreement by maintaining a $\Delta$SAT at 1.89°C relative to pre-industrial (Fig.4c & Tab. 5). As in the run without AU, essentially all of the carbon captured via MOS is stored in the ocean until the end of the 21st century (**FR**, Tab. 5).

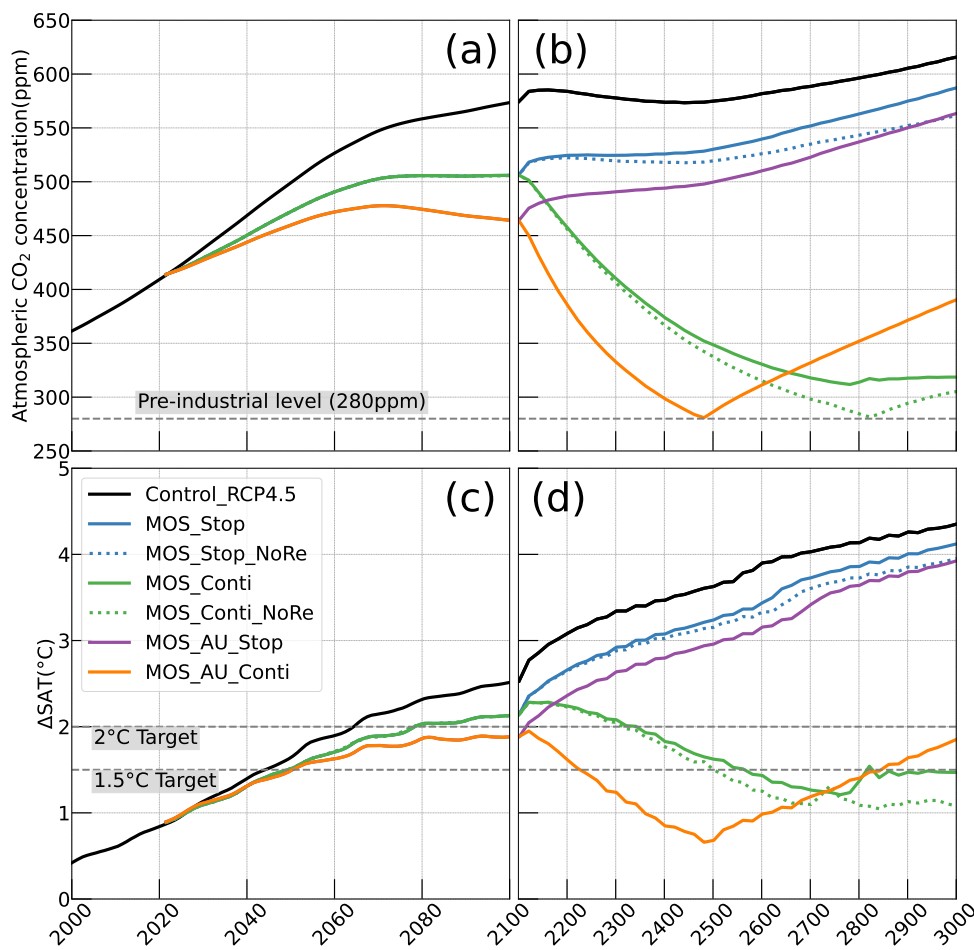

**Figure 4.** Simulations of: **a**, **b**:annual global mean atmospheric $CO_2$ concentration; **c**, **d**: surface averaged temperature relative to the pre-industrial (averaged of year 1850 to year 1900) level of 13.18°C ($\Delta$SAT). Under RCP4.5 scenario, MOS reaches the 2°C target in conjunction with AU, while the 1.5°C cannot be met in all MOS simulations. Note that MOS is terminated whenever pre-industrial concentrations of atmospheric $CO_2$ are reached, as seen for MOS_AU_Conti (orange solid) and MOS_Conti_NoRe (blue dotted) in (**b**&**d**). Both atmospheric $CO_2$ and $\Delta$SAT remain lower than control after MOS termination.

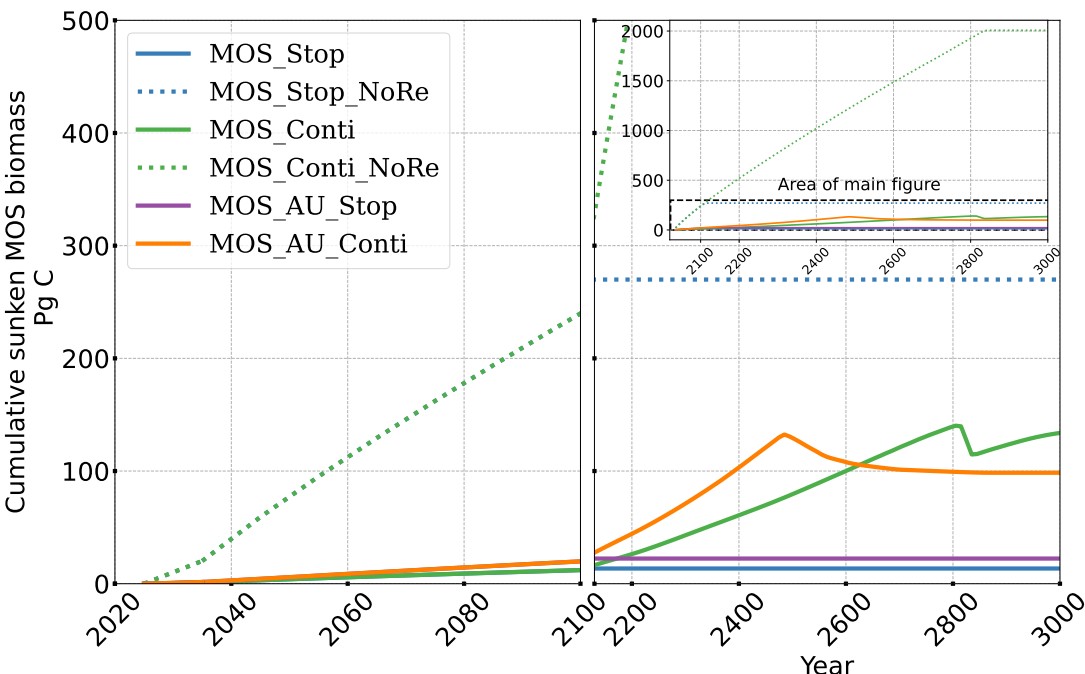

**Figure 5.** Temporal evolution of globally integrated sunk macroalgal biomass on the sea floor. Biomass generally increases with fertilization by AU. In the idealized zero-remineralization simulations, all sunk macroalgal biomass remains on the seafloor, and globally integrated sunk macroalgal biomass shows a monotonous increase.

### 4.3.2 Global carbon cycle impacts

The net effect of the MOS-induced climate-carbon cycle perturbation is an increase of the oceanic carbon reservoir ($C_{oc}$) and a decrease of the atmospheric and terrestrial carbon reservoirs ($C_{atm}$, $C_{ter}$). MOS enhances oceanic carbon uptake by increasing the atmosphere to ocean carbon flux (Fig.B11), which is driven by the DIC removal by MOS in the oceans' surface layer. However, the terrestrial carbon reservoir declines (relative to Control_RCP4.5) in all MOS simulations (Tab.5). The atmosphere to land carbon flux is reduced in MOS simulations (Fig.B10). One cause is the photosynthesis reduced by lower $CO_2$ fertilization of land biota (Keller et al., 2018). This result is in line with other studies showing that CDR can lead to a weakening and even reversal of natural carbon sinks (Keller et al., 2018). For instance by the year 2100 (Tab. 5), due to the declined terrestrial carbon pool ($C_{ter}$, -29.7 PgC), the reduction in atmospheric carbon pool ($C_{oc}$, -142.6 PgC) is less than the gain in the ocean ($C_{oc}$, 171.8 PgC).

Besides, it is also worth noting that the increment of $C_{oc}$ in MOS/MOS_AU is 171.8 PgC/283.9 PgC, which is less than the cumulative amount of carbon sunk out of the surface layer via MOS by year 2100 (Tab.5). One reason is that the reduced oceanic carbon uptake by declined PNPP (phytoplankton net primary production, Sect. 4.3.4) offsets the MOS-induced carbon sequestration. As shown in Fig.9, by the year of 2100, global PNPP is reduced by 20%, while POC export reduced by 30% in MOS_Stop.

MOS also impacts the distribution of DIC in the ocean. The DIC profiles in Fig.6 illustrate that the general effect of MOS is to move more DIC to greater depth ($z \geq 3000m$). By the end of year 2100, MOS simulations show an increased total DIC

concentration in the deeper oceans when compared to Control_RCP4.5 (except for the zero-remineralization sensitivity runs discussed below). For instance, in the deep Southern Ocean, the DIC concentration is on average nearly 80 $\mu$mol/kg higher than the Control_RCP4.5 in year 2100 (Fig.6). The conjunction of MOS with AU increases average deep ocean DIC even more. An example is the simulated increase of DIC in the deep Pacific Ocean and Atlantic Ocean basins by MOS_AU in year 2100 (orange line in Fig.6 DIC panel). In contrast, DIC concentrations are reduced in shallower waters (depth <1000m) as the

air-sea carbon flux is unable to fully compensate the carbon removal by MOS.

The current model results provide additional evidence that the CDR potential of MOS is partly offset by its negative impacts on the pelagic biological production and the biological carbon pump. In an additional model run (not shown) without MOS but with $CO_2$ emissions reduced by the annual equivalents of the MOS-induced carbon exports, yielding a total amount of 270 PgC by year 2100, the 270 PgC emissions removal yields a reduction of atmospheric $CO_2$ by 171.5 GtC by year 2100. This

reduction is 20% higher than the atmospheric $CO_2$ reduction of 142.6 PgC realized in the original MOS simulation where the MOS-induced shading and removal of nutrients from the surface layers reduces the biological carbon pump and the associated carbon storage in the ocean. When $CO_2$ emissions are instead cut by an amount corresponding to 80% of the MOS-induced carbon export, atmospheric $CO_2$ concentrations simulated by the MOS-free emission-cut runs agree closely with those of the respective MOS experiments. That is, each ton of $CO_2$ sequestered in the ocean by MOS is, in our model and on a 100 year

timescale, equivalent to an emission cut of about 0.8 tons of $CO_2$.

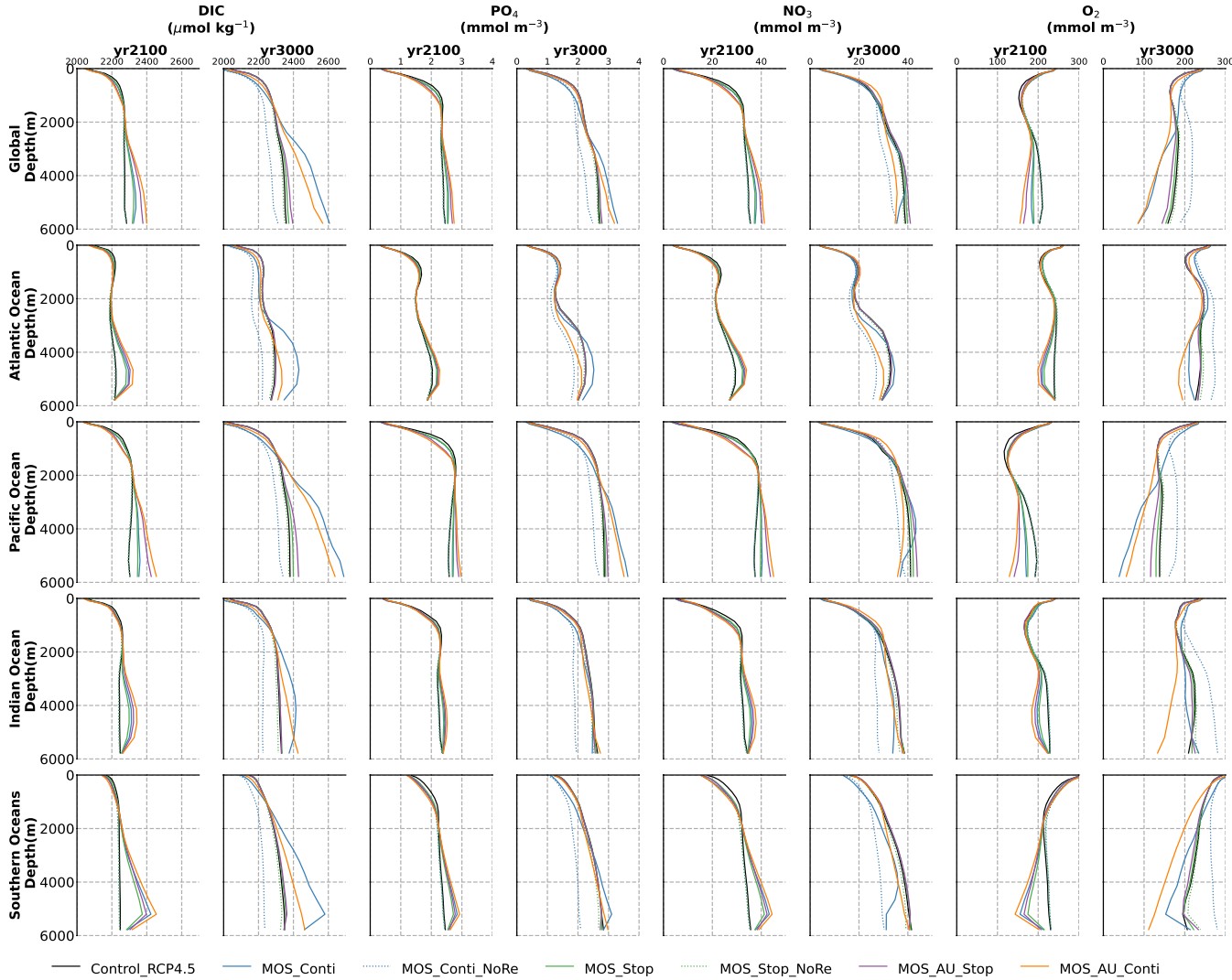

**Figure 6.** Global and basin-wide averaged vertical profiles of various model tracers in year 2100 and year 3000 under the RCP4.5 emission scenario. In general, MOS (except for the zero remineralization ones) transports DIC and nutrients in the surface layer to the deep ocean. The oxygen levels are increased in the mid layers due to the declined downward organic particle flux (Sect.4.3.4) but decreased in the deep ocean caused by the remineralization of sunk biomass. These impacts are strengthened when MOS is deployed continuously and/or in conjunction with AU.

### 4.3.3 Impacts on global nutrients distributions

By the year 2100, the deployment of MOS has changed the global patterns of $NO_3$ and $PO_4$. At the surface $NO_3$ and $PO_4$ concentrations decrease due to MOS nutrient consumption. In the deep ocean (depth $\geq$3000m), $PO_4$ and $NO_3$ increase due to the remineralization of sunk macroalgal biomass (except for the MOS_NoRe simulations). The largest increase in deep ocean

PO$_4$ appears in the Southern Ocean, while the smallest increase is found in the Indian Ocean (PO$_4$ yr2100 groups in Fig.6). This is caused by the distribution of MOS in the surface layer, which in our simulations occupies large areas in the Southern Ocean but only a relatively small region in the Indian Ocean (Fig.2.1).

The remineralization of sunk biomass consumes dissolved oxygen and releases NO$_3$ and PO$_4$. Low-oxygen environments and the associated switch from aerobic remineralization to denitrification, however, occupy relatively small areas, so that this
impact is not easily detectable in global nutrient profiles.

In addition to the localized depletion of nutrients by MOS, the MOS-induced Southern Ocean uptake and transport of N and P to the deep ocean, acts as a type of "nutrient trapping" (Fig.B2). These dynamics thereby reduce nutrients and productivity in mid- to low-latitudes because less N and P are available to be transported out of the Southern Ocean. A similar dynamic has been seen in modelling studies of ocean iron fertilization (Oschlies et al., 2010a; Keller et al., 2014).

### 4.3.4  Impacts on simulated pelagic ecosystems and the organic particle export

In our simulations, large scale deployment of MOS has an impact on pelagic ecosystems, mainly on phytoplankton NPP (PNPP) and biomass.

In the MOS simulations (MOS_Conti/Stop), globally integrated annual PNPP decreases by 20% (9.5 PgC/yr) by year 2100 (Tab. 5). One reason is the canopy shading effect of the floating macroalgae farms, which reduces downward solar radiation
available for the phytoplankton community below. In addition, there is nutrient competition between macroalgae and phytoplankton. As shown in Fig.7a, by the end of the 21st century, PNPP declines in MOS areas, e.g. northern and eastern equatorial Pacific and the Southern Ocean. Intriguingly, in a few regions outside the MOS deployment region, PNPP increases instead. For instance, a 'halo' of enhanced PNPP can be observed surrounding the eastern equatorial Pacific MOS region (Fig.7.a). Similar circumstances are simulated in the North Pacific, the Southern Ocean (60°E:120°E,30°S) and off the equatorial west
coast of Africa. This PNPP-enhancement is sustained by the outflow of residual nutrients from MOS deployment regions (see Fig.B9. One reason is that the macroalgae growth is constrained by the maximum biomass yield as described in Sect.2.2.1. Macroalgae nutrient uptake thus cannot compensate the loss of nutrient consumption by light-limited PNPP within the MOS region which results in enhanced surface nutrients compared to the simulation without MOS, especially when AU supplies additional nutrients to the surface.

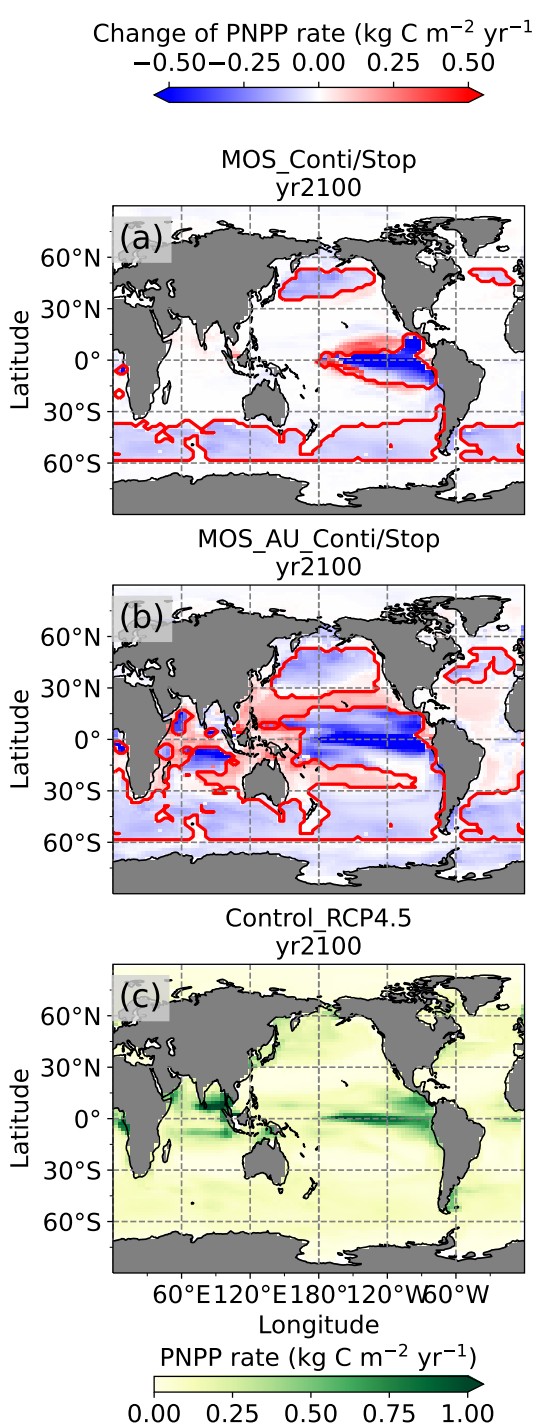

**Figure 7.** Vertically integrated annual PNPP in year 2100. **a**&**b:** MOS minus Control_RCP4.5 with red boundaries contouring the MOS occupied area; **c**: Control_RCP4.5. **a** illustrates a decline in PNPP in MOS occupied areas accompanied by a 'halo' of enhanced PNPP surrounding MOS areas, particularly in the ETP caused by the leakage of residual nutrients (Sect.4.3.4). These impacts on PNPP are amplified in MOS-AU (**b**).

In the MOS_AU simulations, the PNPP 'halo' can be seen in almost the entire MOS-free ocean surface (Fig.7.b). The AU fertilization effect enhances the nutrient leakage from the MOS area. This leads to a higher PNPP in MOS_AU than in the normal MOS simulations (Fig.7.c), but still lower than in the Control_RCP4.5 run (Fig.9.a1).

Changes in the global particulate organic carbon (POC) export flux generally follow the pattern of PNPP changes (Fig.9.b1). Thus, when MOS is present, the PNPP reduction results in a weakened POC flux.

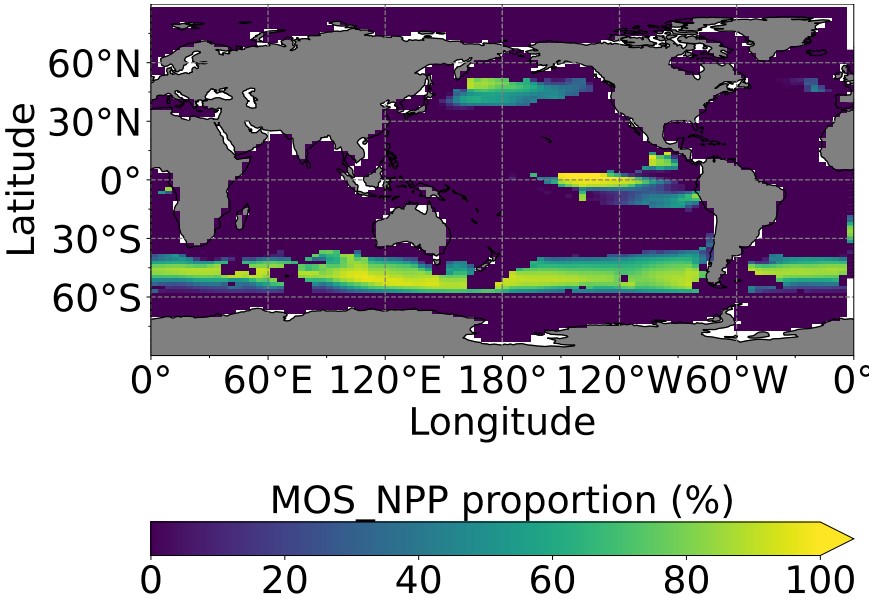

**Figure 8.** Proportion of MOS NPP in the global oceanic NPP by year 2100 (MOS_NPP/(MOS_NPP + PNPP) x 100). Note that the NPP values are converted to carbon using the respective C:N ratio. The MOS_NPP generally amounts to more than 70% of total oceanic NPP where MOS is deployed, indicating an obvious NPP shift from phytoplankton PNPP to MOS_NPP.

Fig.8 illustrates the shift of oceanic NPP from PNPP to MOS_NPP. In regions where MOS is deployed, 70% of the total NPP is macroalgae NPP. The macroalgal NPP is thus nearly twice as high as PNPP. This may lead to additional ecological and biogeochemical issues. One of them is the decline of zooplankton led by the reduced PNPP in this study. We performed an additional simulation, in which the zooplankton grazing on MOS is turned off, and the grazing preferences follow the original settings in Keller et al. (2012). As shown in Fig.B12, the grazing by zooplankton on MOS has no significant effect on neither the zooplankton biomass nor the MOS_NPP. As the zooplankton grazing preference for macroalgae is lower than for phytoplankton, the zooplankton community is still mainly fed by phytoplankton. Therefore, the decline in zooplankton biomass (Fig.B4) follows the declining phytoplankton biomass trend (Fig.B6).

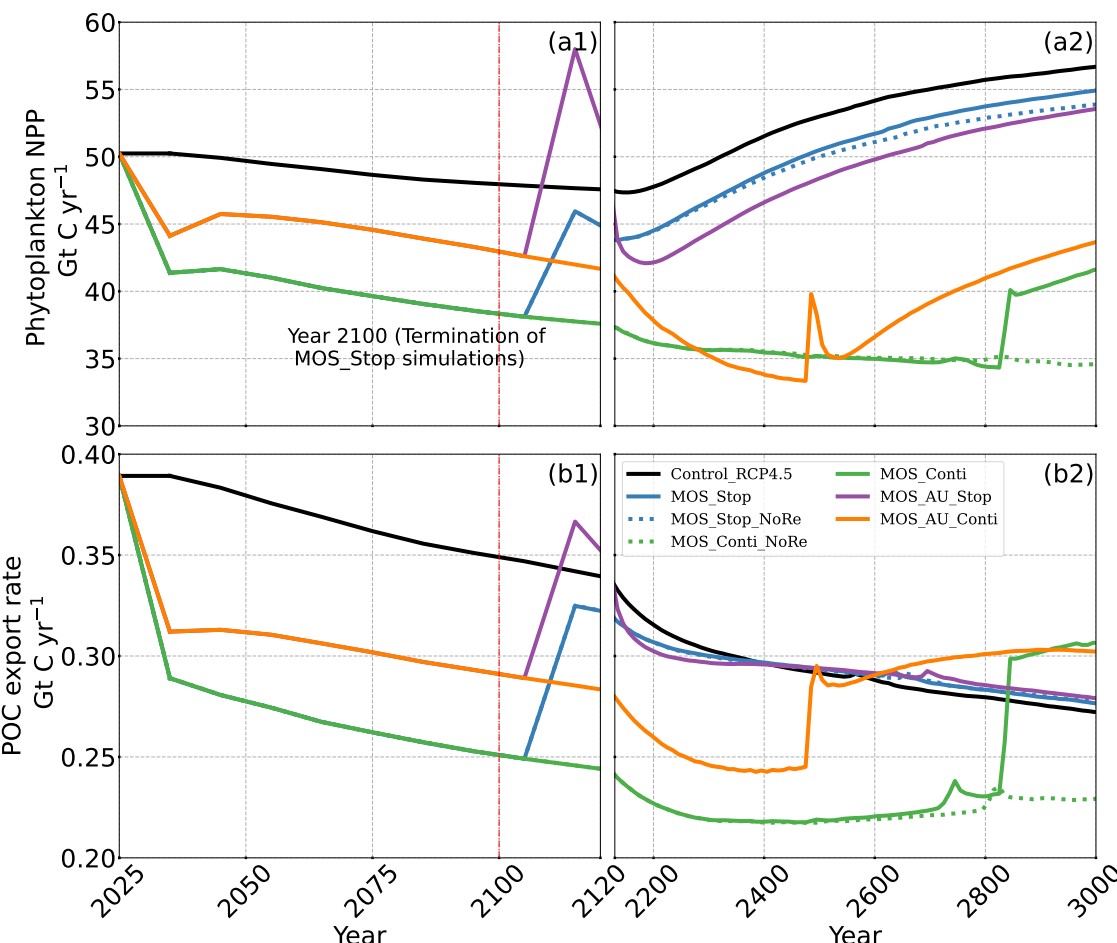

**Figure 9.** Temporal evolution of globally integrated PNPP (a1, a2); downward POC flux at 2km depth (b1, b2). The termination runs branch off of the continuous ones in year 2100 and are identical up to that point. Through the 21st century, MOS reduces PNPP and POC export due to canopy shading and competition for nutrients. Obvious rebounds followed by quick decline can be observed right after terminations of MOS.

### 4.3.5 Impacts on dissolved oxygen

The two major impacts of MOS on oceanic dissolved oxygen are: 1) increased deoxygenation at the sea floor by the remineralization of sunk macroalgal biomass (except for MOS_NoRe) and 2) increased dissolved oxygen at mid depths (e.g. 300m depth) caused by the reduction of the downward POC flux and the associated decline in oxygen consumption by POC remineralization.

In Control_RCP4.5, the global oceanic dissolved oxygen inventory decreases throughout the simulation. The two main driving mechanisms are the reduced solubility in the warming ocean and the decelerating overturning circulation. The long-

term decline of oxygen is especially obvious at depth ( 1200m), which is induced by increasing deep water residence times and the accumulation of respiratory oxygen deficit under global warming (Oschlies et al., 2019; Oschlies, 2021).

As a result of reduced respiration in the upper water column, the size of the oxygen minimum zone (OMZ) in Eastern Tropical Pacific (ETP) shrinks substantially, and the volume of waters with $O_2 < 80$ mmol/m$^3$ in the North Pacific even disappears. In the Southern Ocean dissolved oxygen increases as well (Fig.10.c). This is more pronounced when AU is applied
(Fig.10.e). The increase in dissolved oxygen is caused by decreased microbial remineralization of POC, a consequence of the reduced downward POC flux resulting from the inhibition of PNPP in the surface layer (Sect. 4.3.4). Some decrease in oxygen concentrations occurs in the western Pacific and the Indian Ocean (Fig.10.g), where the surface PNPP is enhanced by the surplus nutrients leaked out from MOS occupied area (see Sect.4.3.4 and Fig.7.c).

Fig.10.d&f and the $O_2$ yr2100 panel of Fig.6 illustrate how MOS changes dissolved oxygen in the deep ocean. Within the
normal MOS simulations, the decline of benthic dissolved oxygen mainly happens in the Southern Ocean by year 2100 with the appearance of a few new areas with oxygen concentrations less than 80 mmol m$^{-3}$ (Fig.10.d). However, when AU is also deployed, the increased macroalgal biomass sinking and remineralization creates even more benthic low-oxygen zones (Fig.10.f) in the ETP and North Pacific Ocean. These new locations correspond to MOS-occupied surface areas.

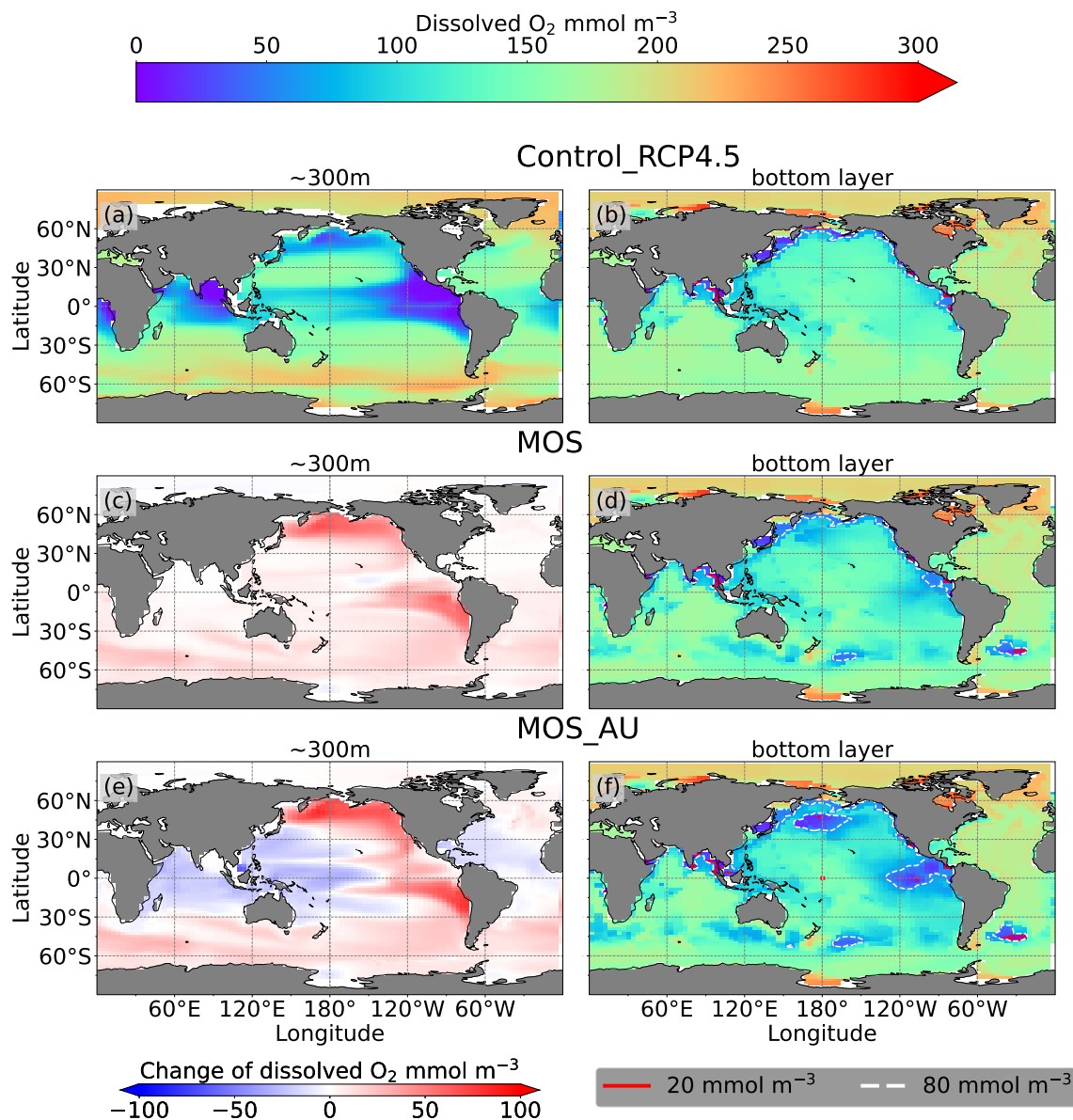

**Figure 10.** Dissolved $O_2$ concentration distribution in year 2100 at 300m depth (**a, c, e**) and the seafloor (**b, d, f**); lines delineate boundaries of OMZs anywhere within the water column with less than 80 mmol m$^{-3}$ oxygen (white dashed) and less than 20 mmol m$^{-3}$ (red solid)). At 300m depth, elevated dissolved oxygen levels in MOS simulations are caused by the decline in POC export (**c**), exceptions are the reduced oxygen concentrations in regions outside the MOS-AU deployment (**e**). On the seafloor, remineralization of sunk biomass creates several new low-oxygen areas (**d, f**).

## 4.4 Long-term effects of MOS

Here we will analyse the long-term effects of hypothetical massive MOS deployment beyond the Paris Agreement time frame on a millennial timescale.

Even after a simulated continuous millennial-scale deployment, the distribution of MOS in year 3000 is nearly identical to the one in year 2100 with only a minimal decrease in biomass (Fig.B1.b). When deployed beyond the year 2100 (MOS_Conti), MOS will continue to sequester carbon and reduce atmospheric $CO_2$ on millennial timescales or, in our set-up, until atmospheric

$CO_2$ falls back to the pre-industrial level of 280ppm. The MOS_Conti simulation ultimately sequesters 2533 PgC and decreases atmospheric $CO_2$ to 318.5ppm $CO_2$ by the year 3000, but never achieves the pre-industrial $CO_2$ level. Notably, atmospheric $CO_2$ stops decreasing by year 2780 and rebounds afterwards even though MOS continues to sequester carbon. This can be explained by a recurrent deep convection in the Southern Ocean around year 2800 that accelerates oceanic carbon leakage back to the atmosphere (Martin et al., 2013; Reith et al., 2016; Oschlies, 2021). Meanwhile, the leakage of MOS-captured

carbon eventually offsets the MOS carbon sequestration (Sect.4.6).

In the sensitivity simulations MOS_Conti_NoRe and MOS_AU_Conti, atmospheric $CO_2$ reaches 280ppm by the year 2820 and 2475, respectively. After reaching 280ppm, MOS is stopped and atmospheric $CO_2$ increases again as remineralized carbon leaks out of the ocean and the surface ocean adjusts to the no MOS situation. The largest increase in $CO_2$ is found in MOS_AU_Conti. Meanwhile, when MOS is deployed (uninterruptedly or till the $CO_2$ 280ppm trigger), the land carbon up-

take is constantly lower than the control level owing to the reduced $CO_2$ fertilization effect. Due to the permanent storage of MOS-C in sunk biomass, rebounds of atmospheric $CO_2$ are relatively gentle in MOS_Conti_NoRe (Fig.B11). Nevertheless, the atmospheric $CO_2$ levels in continuous MOS simulations are significantly lower (35% to 50% of Control_RCP4.5) by the end of year 3000.

The side effects of MOS also persist and often grow in magnitude with continuous deployment. Though PNPP is enhanced

around MOS areas by nutrient leakage (PNPP 'halo', see Sect.4.3.4), the global reduction of surface nutrients and local canopy shading by MOS leads to continuous but gentle lowering of global PNPP after the sharp decreases in the initial 20 years (Fig.9.a2). For instance, in MOS_Conti, PNPP drops by ∼60% by the end of year 3000 (Tab. 5). Correspondingly, in MOS_Conti POC export eventually declines by 50% relative to Control_RCP4.5 (Fig.9b2). In sensitivity run MOS_AU_Conti, the nutrient supply by AU, which initially maintains a higher phytoplankton biomass and NPP than in MOS without AU

(Sect.4.3), declines with time as source waters of the upwelling become reduced in nutrients. Therefore, PNPP as well as the POC export levels drop after year 2200 (Fig.9.a2&b2).

The redistribution of DIC and nutrients is intensified in the continuous simulations. As shown in Fig.6, when remineralization of MOS sunk biomass is turned on, the Pacific deep ocean and the Southern deep ocean show the highest DIC and $PO_4$ enrichment by year 3000. The accumulations depend on ocean circulation (e.g., thermohaline circulation) and the distribution

of MOS at the surface. In MOS_Conti_NoRe, the ocean DIC decreases globally relative to Control_RCP4.5. This results from the continuous DIC removal into biomass via MOS with no remineralization. Another cause of the declined global DIC is the

declining downward POC flux owing to the PNPP reduction caused by the declining nutrient levels in the surface layer (see Sect.4.3.4).

NO$_3$ enrichment in the deep ocean is considerably smaller than that of DIC and PO$_4$, because of enhanced denitrification in the developing benthic low-oxygen regions (Sect.4.3.5, Fig.B3). In the zero remineralization situations, deep ocean PO$_4$ and NO$_3$ concentrations decrease compared to the control levels, due to the reduced remineralization of POM resulting from the weakened downward flux of POM.

As shown in Fig.11 and the O$_2$ yr3000 panel of Fig.6, dissolved oxygen concentrations at mid depth (e.g. 300m) increased during millennial MOS deployment due to reduced PNPP and associated downward flux and remineralization of POM in the water column. In benthic waters, regions with very low dissolved oxygen are shown in Fig.11.f&j in the Pacific and Southern Ocean. In contrast, increased oxygen concentrations are found in MOS_Conti_NoRe (Fig.11.h), especially in the Atlantic, the Indian and the Southern Ocean. Besides the absence of oxygen consumption by macroalgal biomass remineralization, another reason for these oxygen increases lies in the reduction of POC downward flux described in Sect. 4.3.4.

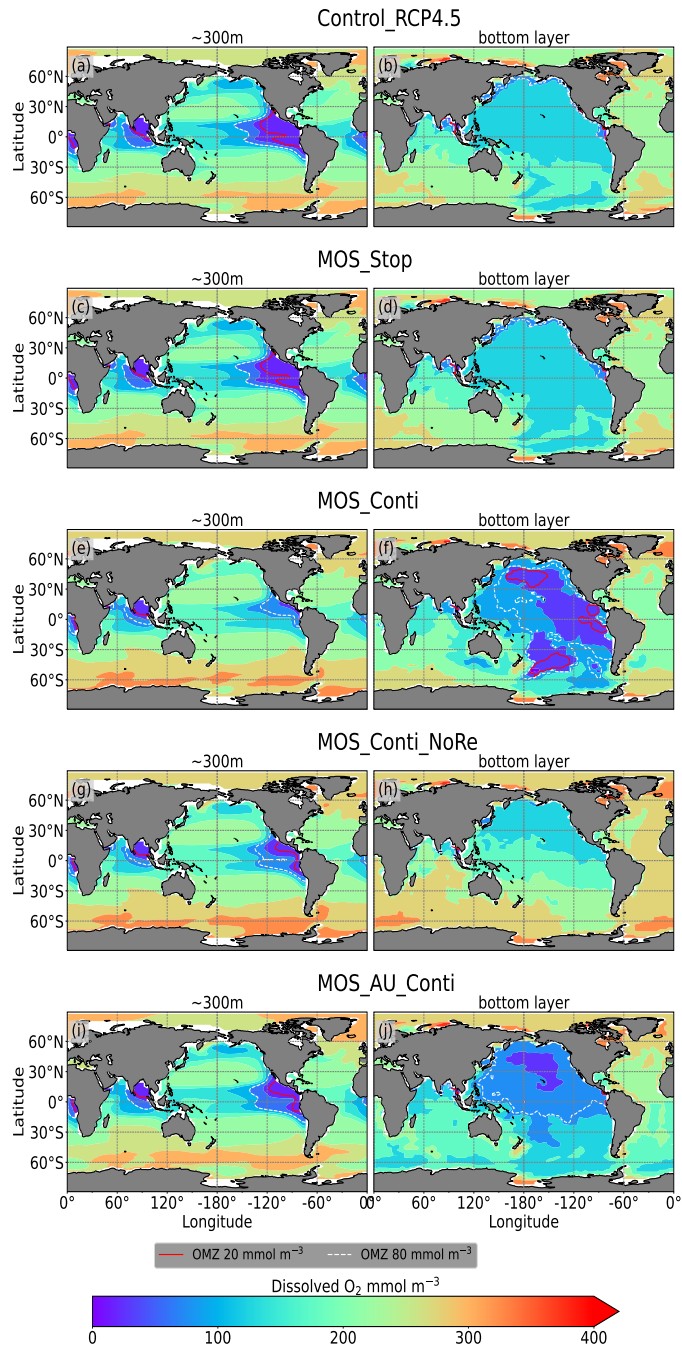

**Figure 11.** Dissolved $O_2$ concentrations at depth ~300m (left panel) and the ocean bottom (right panel) in year 3000: contour lines indicate boundaries of OMZs with less than 80 mmol $m^{-3}$ oxygen (white dashed) and less than 20 mmol $m^{-3}$ (red solid). Continuous MOS deployment further shrinks the OMZ at 300m depth (e,g,i) but expands them at bottom (f,j) except for the zero remineralization one in (h).

## 4.5 Termination effects

After termination of MOS_Stop in year 2100, the atmospheric $CO_2$ concentrations and SAT both rise, but generally remain lower than those of the Control_RCP4.5 simulation (Fig.4.b,d). More than half (**FR** ranges from 58.6% to 64.4%, Tab.5, calculated by Eq. 23) of MOS-captured carbon is still stored in the ocean by the end of year 3000. As shown in Fig.4.b,d, in the termination simulations MOS_Stop and MOS_AU_Stop $CO_2$ concentration and SAT gradually converge against the Control_RCP4.5 run as a result of DIC from remineralized macroalgal biomass being transported to the ocean surface and into the atmosphere (see subsequent Sect.4.6). By year 3000, the atmospheric $CO_2$ in MOS_Stop is only 28.5 PgC less than in Control_RCP4.5, while $\Delta$SAT slightly rebounds from -0.38°C to -0.23°C. In MOS_AU_Stop, the differences of $pCO_2$ and $\Delta$SAT are smaller than the normal MOS, as AU has augmented MOS carbon sequestration and 64.4% of it is retained in the ocean. As expected, the idealized non-remineralization condition (MOS_Stop_NoRe) is able to permanently store the sequestered carbon, thus the rebounds of $CO_2$ and $\Delta$SAT are less than the normal MOS_Stop.

In all MOS termination simulations, PNPP and POC export rebound abruptly following the cessation of MOS, but sharply drop over the subsequent decades (Fig.7.a1). The sharp increase in PNPP and POC export results from the sudden absence of macroalgae as a main competitor for nutrients and light. The subsequent decline in PNPP and POC export results from consumption of the surface nutrients and the lack of subsurface nutrients that has previously been exported directly to the seafloor with sinking of macroalgae biomass (Fig.B7). Afterwards, PNPP recovers gradually due to the slow returning of remineralized nutrients to the upper ocean. By the year 3000, PNPP in MOS_Stop recovers to 97% of the control level (Tab. 5), with the only differences attributable to the slightly different climate state. In the MOS_NoRe simulation, the PNPP recovery is slower due to the permanent nutrient removal from the upper water column. In the MOS_AU_Stop simulation PNPP rebounds to higher levels than the normal MOS, but drops to lower levels afterwards. The amplified oscillation of PNPP results from the simultaneous termination of AU and MOS: When MOS_AU_Stop is suddenly terminated, the canopy shading and nutrient competition by MOS are removed. Meanwhile, the surplus of nutrients from AU still remains. This boosts PNPP rapidly. However, once these nutrients are consumed, the natural nutrient supply to surface waters is insufficient to maintain the high PNPP.

After the termination of MOS, the rate of oceanic carbon uptake falls abruptly (Fig.B11). After a short peak caused by the abrupt rebound of PNPP, it remains slightly lower than the control level due to the declined PNPP rates and lower atmospheric $CO_2$ levels. Oppositely, the MOS-induced reduction in terrestrial carbon uptake starts to rebound after MOS cessation (Fig.B10) due to the rise of atmospheric $CO_2$, which tend to enhance the terrestrial photosynthesis.

When MOS deployment is stopped, the elevated (compared to Control_RCP4.5) dissolved oxygen concentrations at mid depth generally decline as the downward POC flux recovers ($O_2$ yr3000 panel in Fig.6). The lowered oxygen concentrations in the deep ocean are also reversible after cessation of MOS. For instance, by year 3000, the benthic dissolved oxygen of MOS_Stop (Fig.10.d) is similar to Control_RCP4.5 (Fig.10.b).

## 4.6 Leakage of MOS Sequestered Carbon

The leakage of MOS sequestered carbon (MOS-C) occurs mostly in the Southern Ocean (Fig.12, Fig.B5). The explanation lies in the dynamics of Southern Ocean upwelling, where Pacific Deep Water (PDW), Indian Deep Water (IDW) and Antarctic Deep Water (AADW), laden with DIC of remineralized MOS biomass, reach the subantarctic ocean surface (Talley, 2013; Weber and Bianchi, 2020; Anderson and Peters, 2016). Moreover, the recurrent deep convection (see Sect.4.4) in the Southern Ocean around year 2600 accelerates the carbon leakage, which can be observed in Fig.12 as an enhanced outgassing around year 2600.

The outgassing of MOS-C in discontinuous simulations (e.g. MOS_Stop) starts since year 2100, while the continuous ones starts since year 2300 (Fig.12). Thus, by the end of the 21st century, nearly the entire MOS-C in all MOS simulations is retained in the ocean. However, by the year 3000, even in the continuous MOS simulation, only about 75% of MOS-C remains in the ocean, while the accelerated vertical water transport by AU slightly reduces this portion to 73%. In run MOS_Stop 59% of MOS-C remains in the the ocean by year 3000, whereas the additional sunk macroalgal biomass in MOS_AU_Stop results in more MOS-C (64%) being retained (Tab.5). When sunk biomass is free from remineralization (NoRE runs), the contained carbon is permanently isolated from the atmosphere and stored in the ocean.

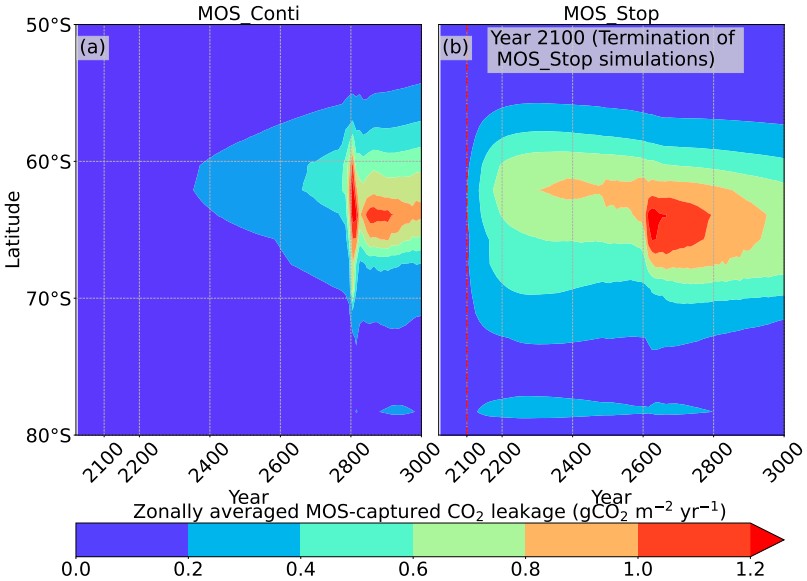

**Figure 12.** Zonally averaged MOS-captured carbon outgassing in the simulation (**a**) MOS_Conti and (**b**) MOS_Stop. When conveyed back to the surface, the DIC from MOS remineralization participates the air-sea exchange (Sect. 2.2.5). MOS-C outgassing starts in year 2100 when MOS is terminated (**a**), or after year 2300 when continuously deployed. The outgassing mainly happens in the Southern Ocean. The outgassing is strengthened around year 2800 when a Southern Ocean deep convection event accelerates the upwelling of deep waters with high concentrations of remineralized DIC (Sect. 4.4.

## 5 Concluding discussions


In this study we have tested the potential of the 'macroalgae open-ocean mariculture and sinking (MOS)' as a carbon dioxide removal method. Although environmental conditions (e.g. nutrients, temperature, etc.) in the open oceans differ considerably from the coastal/near-shore regions where macroalgae aquaculture is currently applied in reality, our simulations suggest that in certain open ocean regions macroalgae may successfully grow and sequester carbon (if engineering constraints can be

overcome). Even for continuous deployment at a maximum scale currently deemed possible, MOS alone is not able to reduce the warming to the 2°C target by the end of the 21st Century under the RCP4.5 moderate mitigation scenario. This finding is consistent with conclusions from previous studies that no single carbon dioxide removal (CDR) method alone can ensure reaching the current climate goals (Keller et al., 2014; Anderson and Peters, 2016; Fuss et al., 2020; IPCC, 2018). Clearly emissions reductions must be the primary means of mitigation, a portfolio of CDR options can only compliment these other

efforts.

In order to estimate the maximum $CO_2$ removal potential, the possible synergy of MOS with artificial upwelling (AU) has been investigated. The employed AU concept aims at piping nutrient-rich deep water to the surface to enhance the growth of macroalgae in MOS. As expected, AU is found to have the potential to successfully enlarge the growing area of MOS and enhance the CDR capacity of MOS.

In the first 80 years of deployment, the maximum MOS carbon sequestration potential is 3.38 PgC/yr for regular MOS, but can be boosted up to 5.56 PgC/yr with assistance from AU. If deployment is discontinued from year 2100, about 58.6% to 70.2% (normal MOS and MOS_AU, respectively) of MOS sequestered carbon would be retained in the ocean by year 3000.

Several potential side effects have also been revealed and analyzed. One side effect is the reduction in phytoplankton NPP (PNPP) due to canopy shading and nutrient removal from the sea surface to the bottom by MOS. The declined PNPP in turn

offsets ∼37% of the MOS CDR. Intriguingly, some areas with enhanced PNPP (PNPP 'halo', Sect.4.3.4) are found surrounding the major areas occupied by MOS, fueled by the residual $NO_3$ that leaks from MOS areas.

Another strong side effect of large-scaled MOS is the impact on oxygen distributions. Dissolved oxygen concentrations increase in near-surface and intermediate waters, as MOS reduces the downward flux of plankton-derived organic matter by restraining the surface PNPP. On the other hand, the massive amount of sunk biomass from MOS at the ocean bottom and its

subsequent remineralization consumes oxygen and can create large benthic OMZs.

An uncertain factor is the fate of the sunk biomass. It will affect the benthic fauna by depositing large amounts of organic matter as well as expanding low oxygen regions on the sea floor upon oxygen consumption by remineralization. Therefore, we performed additional sensitivity simulations focusing on the macroalgal biomass remineralization rate. When macroalgal biomass does not undergo microbial remineralization, the captured $CO_2$ can be permanently stored without leakage. This in-

creases the CDR potential of MOS. The benthic OMZs created by remineralization of sunk biomass would also be avoided, while the shrinking of intermediate water OMZs persists. However, other side effects can not be neglected: in zero reminer-alization simulations, the constant removal of nutrients in the surface will impede the recovery of PNPP. This may eventually affect the marine surface ecology and ocean services such as food provision. These potential side effects are also noteworthy

for another macroalgae farming approach, i.e. harvesting the macroalgae biomass for bioenergy with carbon capture or storage (BECCS) or biochar (Sect. 3.2).

The impacts of MOS on oxygen distributions may also influence the oceanic sources of $N_2O$, an atmospheric GHG gas and a major ozone-depleting compound (Ravishankara et al., 2009). The increased/decreased oxygen levels in the mid/bottom layers impacts denitrification, and may weaken the $N_2O$ sources in the subsurface but increases those in the deep waters (Bange et al., 2019).

Moreover, attention should also be paid to the calcification by calcareous macroalgae (if cultivated) and/or associated epibionts that grow on macroalgae. Bach et al. (2021) have suggested that epibionts living on *Sargassum* offsets 16.5% of the POC fixed by *Sargassum* and therefore decrease its natural carbon sequestration potential if the biomass was intentionally sunk for CDR purposes. These calcification rates and the response to ocean acidification of macroalgae are also species-specific (Koch et al., 2013). These factors need to be investigated with further research if macroalgae are to be considered for ocean-based CDR methods.

The production and export of DOC is also an area of further studies of large-scaled farmed macroalgae for carbon sequestration. Macroalgae have been reported to release considerable amounts of DOC and to contribute to the global DOC export from coastal to open ocean waters. The estimated global DOC release of macroalgae habitats is 355 Tg C $yr^{-1}$(Krause-Jensen and Duarte, 2016). The averaged DOC release rate by macroalgae is of $23.2 \pm 12.6$ mmol C $m^{-2}$ $d^{-1}$ (equivalent to $8.5 \pm 4.6$mol C $m^{-2}$ $yr^{-1}$), but with a high range of 8.4 to 71.9 mmol C $m^{-2}$ $yr^{-1}$(Barrón et al., 2014). If we simply multiply this annual averaged DOC release rate with the MOS occupied area ($S_{MOS}$, Tab.5), the estimated annual DOC release by MOS would be $7.1 \pm 3.8$PgC (MOS) or $12.9 \pm 7.0$ PgC (MOS_AU). Although the refractory DOC released by macroalgae could potentially be an additional contribution of carbon sinking by MOS, the available information of the generation and composition of the macroalgae DOC is not enough to either parameterize a model of this process, and more research on the topic is needed (Krause-Jensen and Duarte, 2016; Barrón et al., 2014; Barrón and Duarte, 2015).

Another side effect not investigated here is the production and emission of halocarbons from macroalgae farms. Macroalgae species have been reported to generate halocarbons in polar, temperate and tropical coastal regions with a highest producing rate of 6000 pmol $CHBr_3$ $gFW^{-1}$ $h^{-1}$ (Leedham et al., 2013; Baker et al., 2001; Carpenter and Liss, 2000; Latumus, 1995). These volatile low molecular-weight halocarbon compounds (e.g. $CH_3I$, $CHBr_3$ and $CHCl_3$) are potent greenhouse gases (Meinshausen et al., 2011). They also influence stratospheric ozone destruction when transported by deep atmospheric convection into the stratosphere (Ziska et al., 2013; Tegtmeier et al., 2012, 2013), therefore enhancing radiative forcing (Ramaswamy et al., 1992; Daniel et al., 1995). Large-scale MOS cultivation might release a significant quantity of halocarbons. However, as MOS also reduces global phytoplankton NPP, it is likely that the production of halocarbons by phytoplankton decreases. Further studies are needed to investigate possible effects of halocarbon emissions from large-scaled macroalgae cultivation and how this is offset by a potential decrease in phytoplankton halocarbon production.

Meanwhile, here we only discuss the CDR potential of MOS_AU combination under an idealized situation, where the artificial upwelling (AU) upwells nutrients without changing the ambient water temperature on the surface. The aim of deploying akin AU to MOS is to assess the maximum potential of CDR, as AU can upwell nutrient-rich deeper water. However, it can

be expected that the NPP of MOS will be slower if the ambient water temperature is reduced by the upwelled cold water up-welled, which will further limit the CDR potential of MOS_AU. Moreover, several dominant side effects due to the upwelled cold water in ordinary AU have been revealed, such as the quick rebound or even surpassing of CO2 concentrations and surface temperatures after the termination of AU (Oschlies et al., 2010b). If the cold water upwelling was included, there would be extra analysis on the associated impacts on planetary radiation budget imbalance, marine biogeochemistry and global carbon pool (e.g., the enhanced terrestrial carbon sequestration due to the cold effects by AU), which are beyond the scope of this study. Thus, we used the hypothetical AU systems excluding thermal exchange to address the nutrients supplementing of AU to MOS and avoid the unnecessary complexities and side effects in the meantime. However, further studies are required if ordinary AU should be considered in association with macroalgae farming.

Besides the CDR effect of MOS, in case of large scale deployment, the macroalgae farms are likely to increase the albedo of the oceans's surface, especially when they occur near the sea surface (Fogarty et al., 2018; Bach et al., 2021). Meanwhile, we have not considered possible hydrodynamic impacts on ocean circulation, as the thick macroalgae layers and the farming infrastructures may influence the momentum and mixing of the ambient flow fields (Liu and Huguenard, 2020; Nepf, 2012; Thomas and McLelland, 2015).

The MOS model analysis presented here clearly has some limitations, that future studies might improve on. One of the most critical issues is to improve the realism of the model design by including more representative macroalgae species for various regions. Another aspect is the consideration of dynamic cellular stoichiometry of macroalgae. With a better simulation of the cellular quota, we could improve our understanding of the relation of nutrient and carbon fluxes between MOS and the environment. Explicit consideration of the variable morphology of the macroalgae, as well as of the impacts of currents on frond erosion (Broch and Slagstad, 2012) would also improve the representation of macroalgae loss rates in the model. Further optimization of deployment timing and location for MOS are achievable by evaluating data from field tests or implementations of macroalgae mariculture in the open oceans. Another aspect that needs improvement is the modeling of benthic macroal-gal biomass remineralization. Here we treated macroalgal biomass homogeneously as particulate organic matter. Though the degradation of macroalgal fragments under deep sea conditions (e.g. low temperature and unique microbial colonies) remains unclear, it might be different from POC in terms of remineralization rate and oxygen consumption. Tracking of macroalgae se-questered carbon will be required to record its fate and possible carbon leakage after sinking (e.g. by an eDNA method to trace macroalgae carbon in marine sediments by D'Auriac et al. (2021)). The economic perspectives of developing and deploying MOS also needs to be investigated. Furthermore, more research on how large-scale macroalgae mariculture will impact human activities (e.g., ocean shipping, fisheries) needs to be undertaken. Associated legal and political issues regarding the usage of international waters for MOS deployment should be considered as well.

Overall, this study adds to the rapidly expanding field of considering macroalgae cultivation for $CO_2$ removal. The evidence from this study suggests that macroalgae mariculture&sinking has a considerable CDR potential under these idealized con-ditions, but brings about substantial side effects on marine ecosystems, and marine biogeochemistry. Given this, the concept requires further research with less idealized experimental settings to determine if its CDR benefits outweigh the side effects (Dean et al., 2021).

## 6 Model codes and data availability

The model codes are available online at https://git.geomar.de/jiajun-wu/wu_esd_cdr_mos.

The data used to generate the contents, tables and figures is available online at https://data.geomar.de/downloads/20.500.12085/d88214cc-43aa-40d4-be40-f26aa346e8fa/.

## 7 Author contribution

J.W., D.P.K. and A.O. conceived and designed the experiments. J.W. implemented and performed the experiments and analyzed the data. J.W. wrote the manuscript with contributions from D.P.K. and A.O.

## 8 Competing interests

The authors declare that they have no conflict of interests.

## 9 Acknowledgements

D.P.K. and A.O. have been supported by the European Union's Horizon 2020 research and innovation program under grant agreement numbers 869357 (OceanNETs) and 820989 (COMFORT). J.W. has been supported by the China Scholarship Council. We thank Wolfgang Koeve, Nadine Mengis, Karin Kvale and Fabian Reith for helpful discussions.

## Appendix A: MOS validations

### A1    MOS yield calculation

For the convenience of calculation, we assume that when MOS occupies a surface grid cell the area is covered by parallel
cultivation ropes (lines) with an interval distance of $d$ (see table 1 & Fig.A1). The total length of cultivation lines ($L_{MOS}$, in
meters) of MOS in the grid cell is then:

$$L_{MOS} = S_{MOS} \div d \tag{A1}$$

where $S_{MOS}(m^{-2})$ is the area of ocean surface occupied by MOS. Accordingly, the conversion between macroalgal biomass
yield on ropes ($Y_{rope}$ in kg DW m$^{-1}$) or in fields ($Y_{field}$ in kg DW m$^{-2}$) can be calculated as:

$Y_{rope} = Y_{field} \times d \tag{A2}$

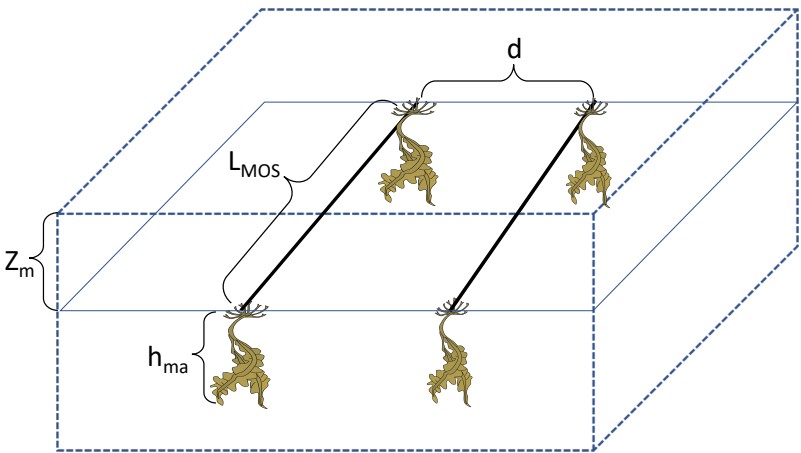

**Figure A1.** Sketch of key features of MOS.

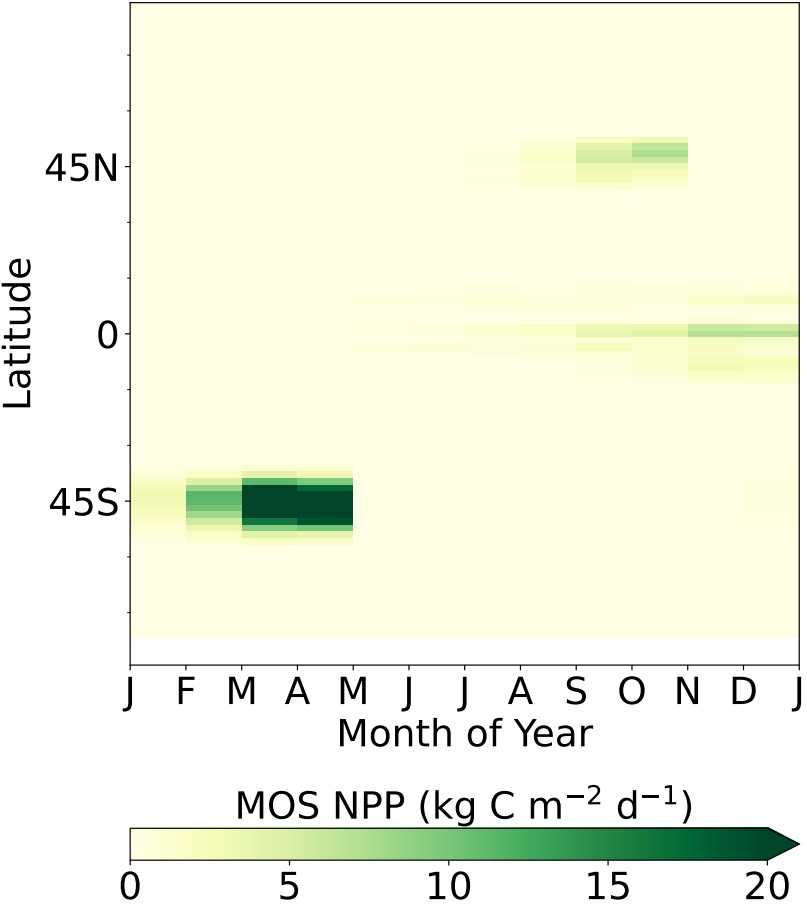

**Figure A2.** Hovemoeller plot of latitudally and vertically integrated MOS NPP. High NPP are found in the Southern Ocean. The change of MOS NPP follows the seasonal solar radiation in UVic ESCM.

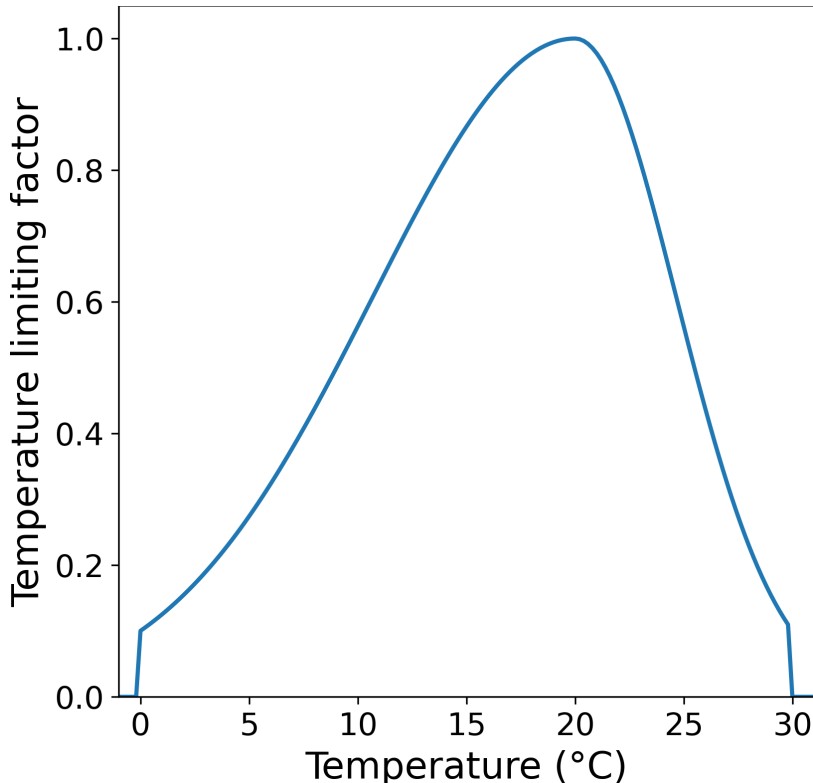

**Figure A3.** Temperature optimum curve of the macroalgae in MOS

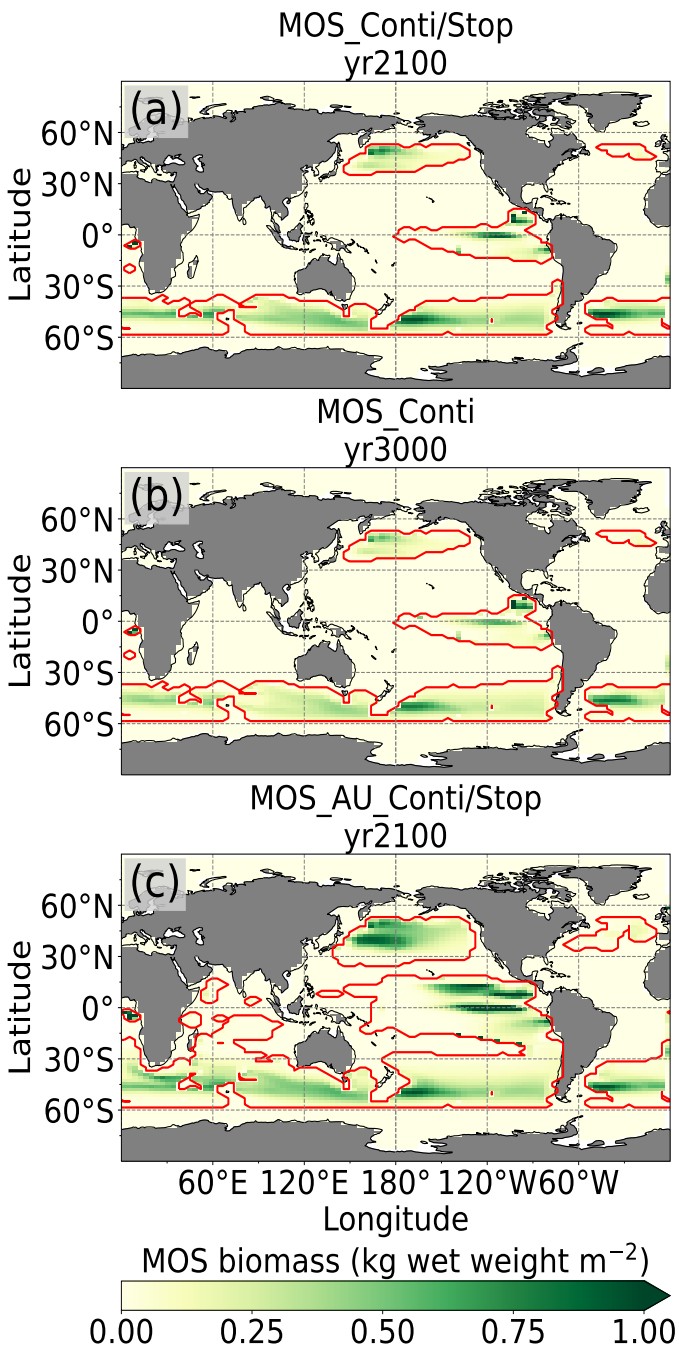

**Figure B1.** MOS biomass distributions. Red lines contour the maximum MOS occupied area during the previous years. The annual macroalgal biomass of MOS in this figure is an average over a 10 year period, which includes times of low and high biomass due to the sinking of biomass. Thus the biomass shown here is less than the biomass shown in Fig.2.

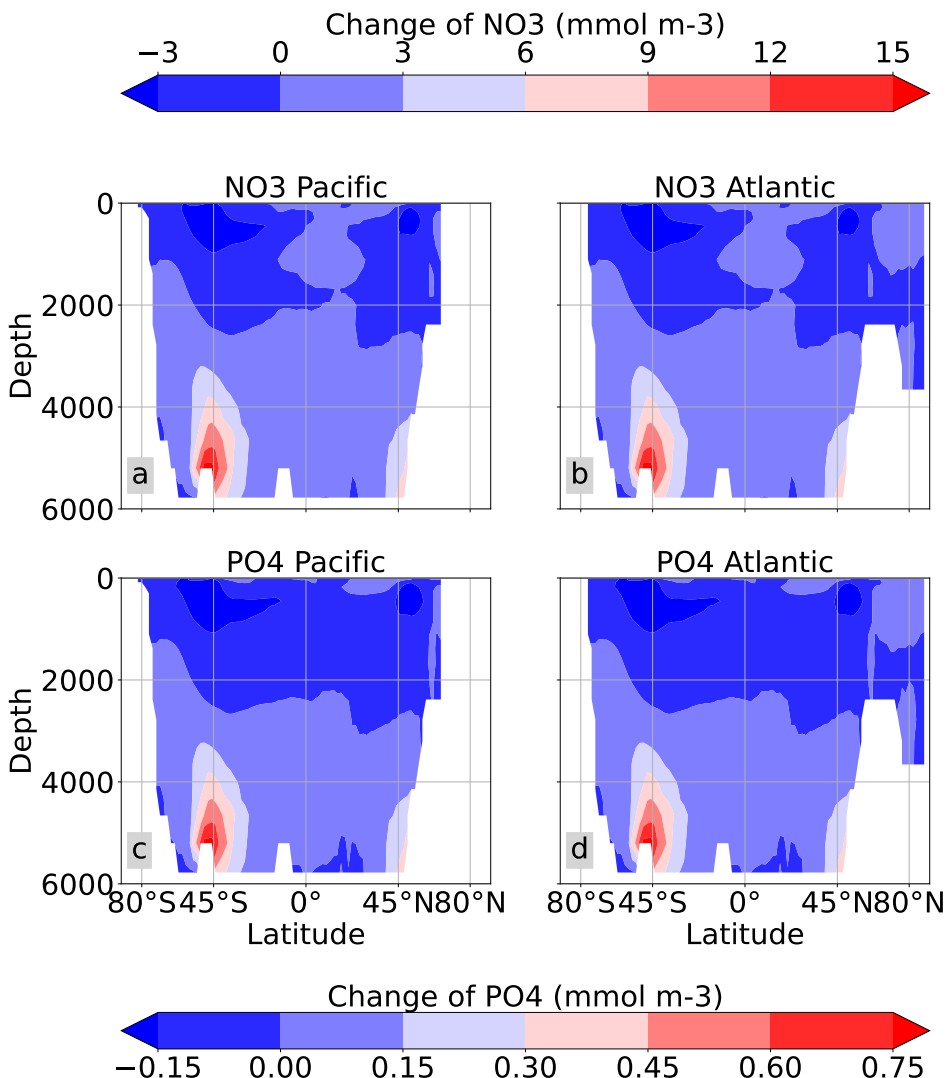

**Figure B2.** Nutrients horizontal distribution changes in the Pacific & Atlantic basin (MOS_Conti) relative to RCP4.5 at year 2100. The nutrients trapping by MOS can be observed on the upper layers, at e.g. the Southern Ocean and mid-high latitude in the northern hemisphere. The nutrients are enriched at the ocean bottom by the remineralization of MOS biomass, especially in the Southern Ocean deep waters.

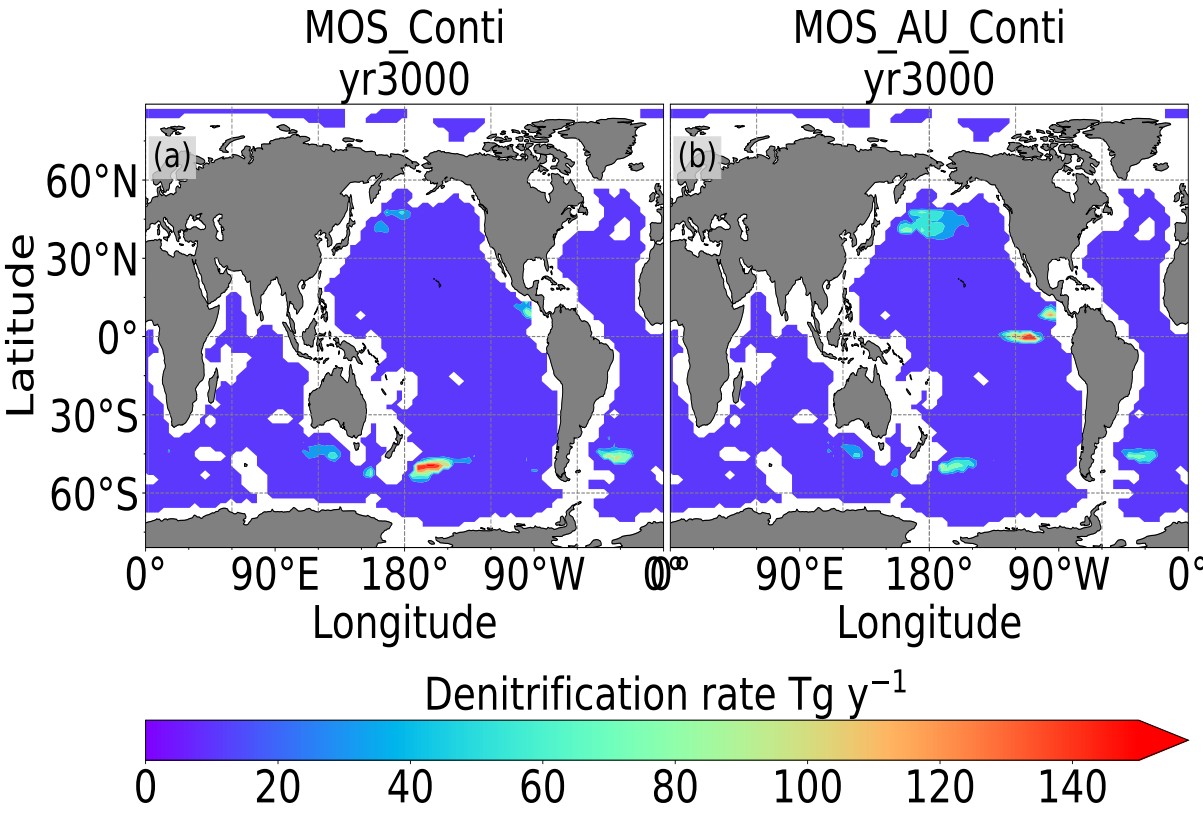

**Figure B3.** Denitrification rate at depth 3000-6000m in year 3000, where the oxygen level is lower than 5 $\mu$mol m$^{-3}$ caused by the remineralization of continuously sunk MOS biomass.

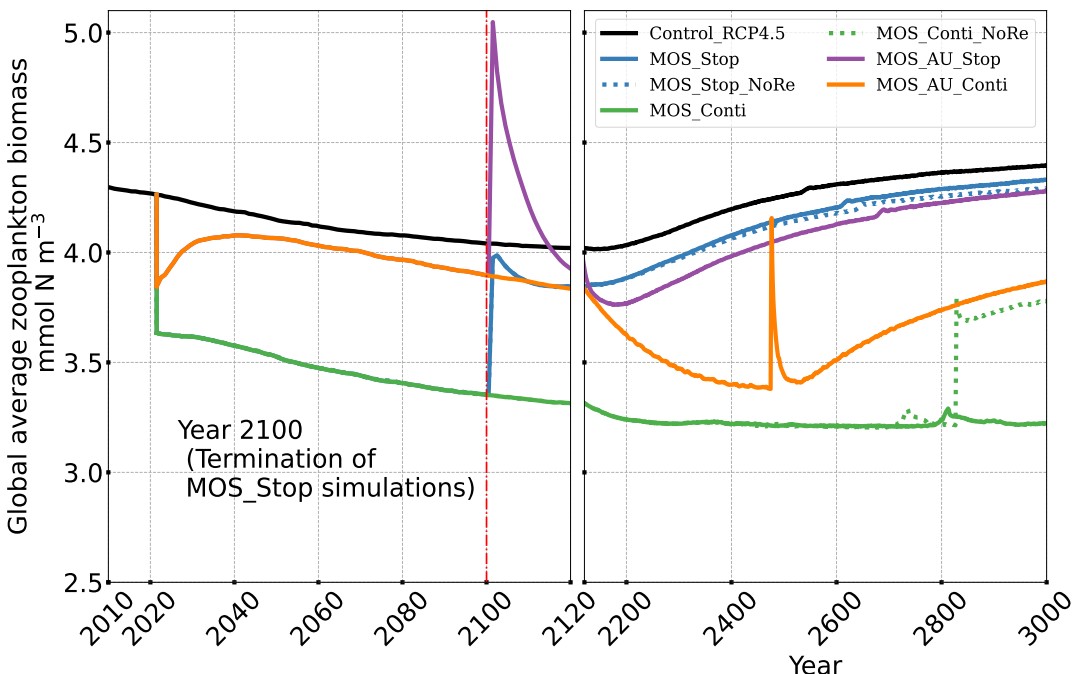

**Figure B4.** Plot of global averaged biomass of zooplankton

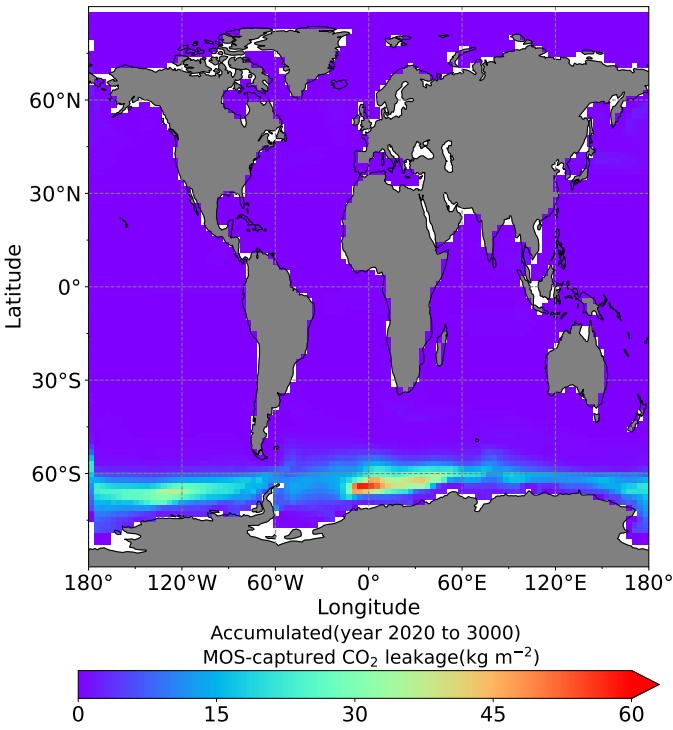

**Figure B5.** Cumulative (year 2020 to 3000) leakage of MOS-captured carbon in the simulation MOS_Stop.

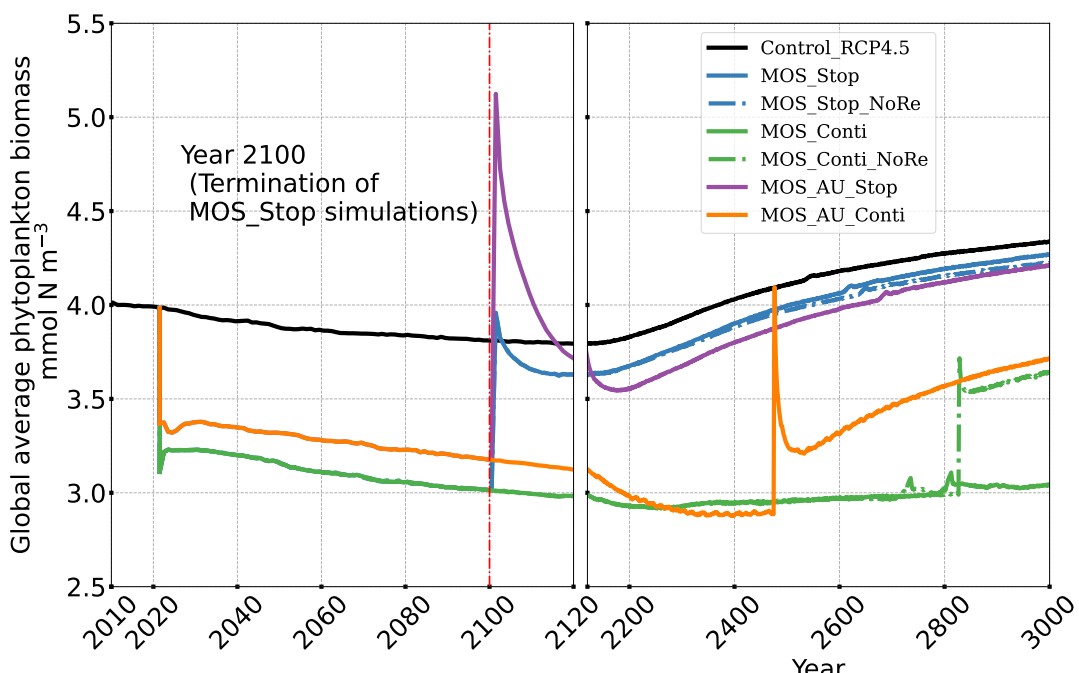

**Figure B6.** Plot of global averaged phytoplankton biomass

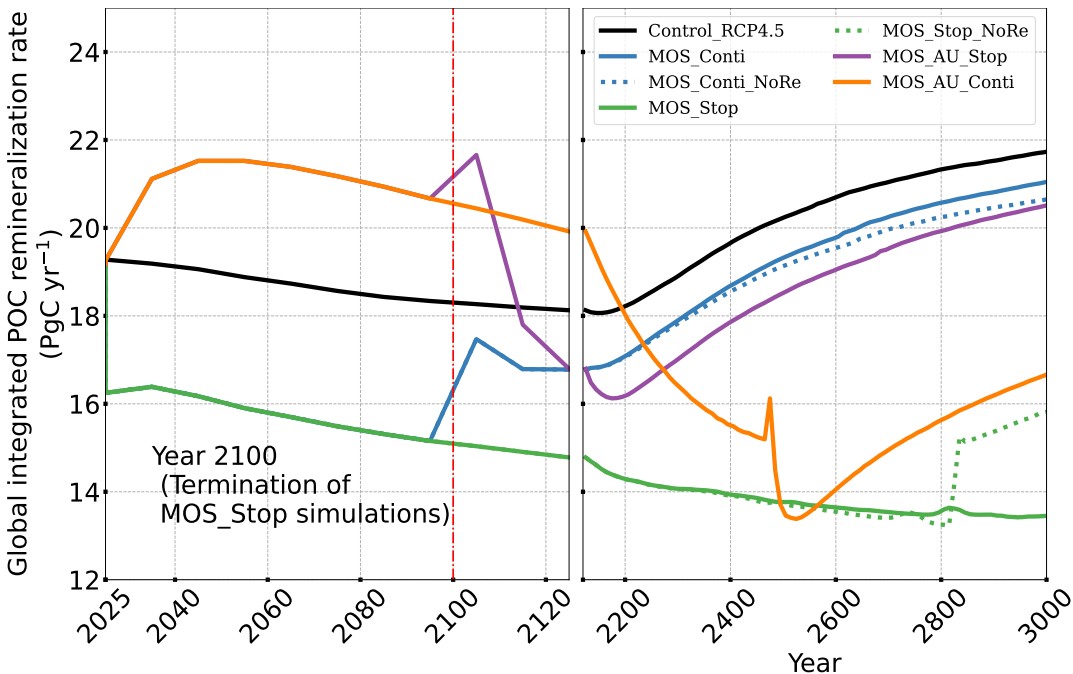

**Figure B7.** Plot of 0-2km averaged detritus remineralization rate

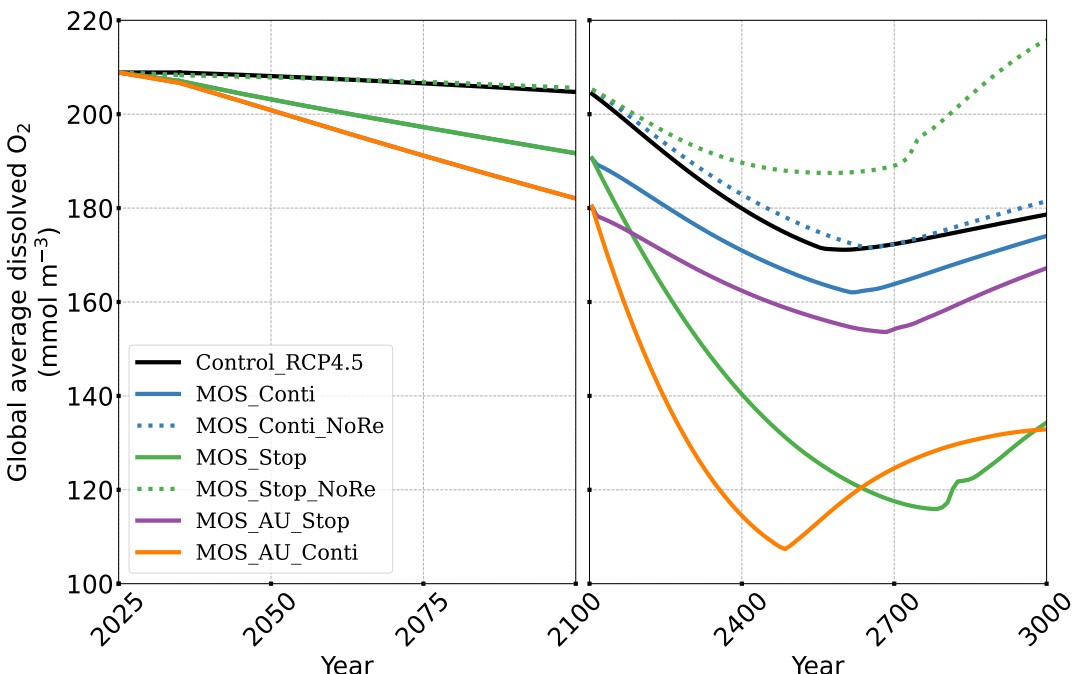

**Figure B8.** 3-6km averaged dissolved oxygen concentrations

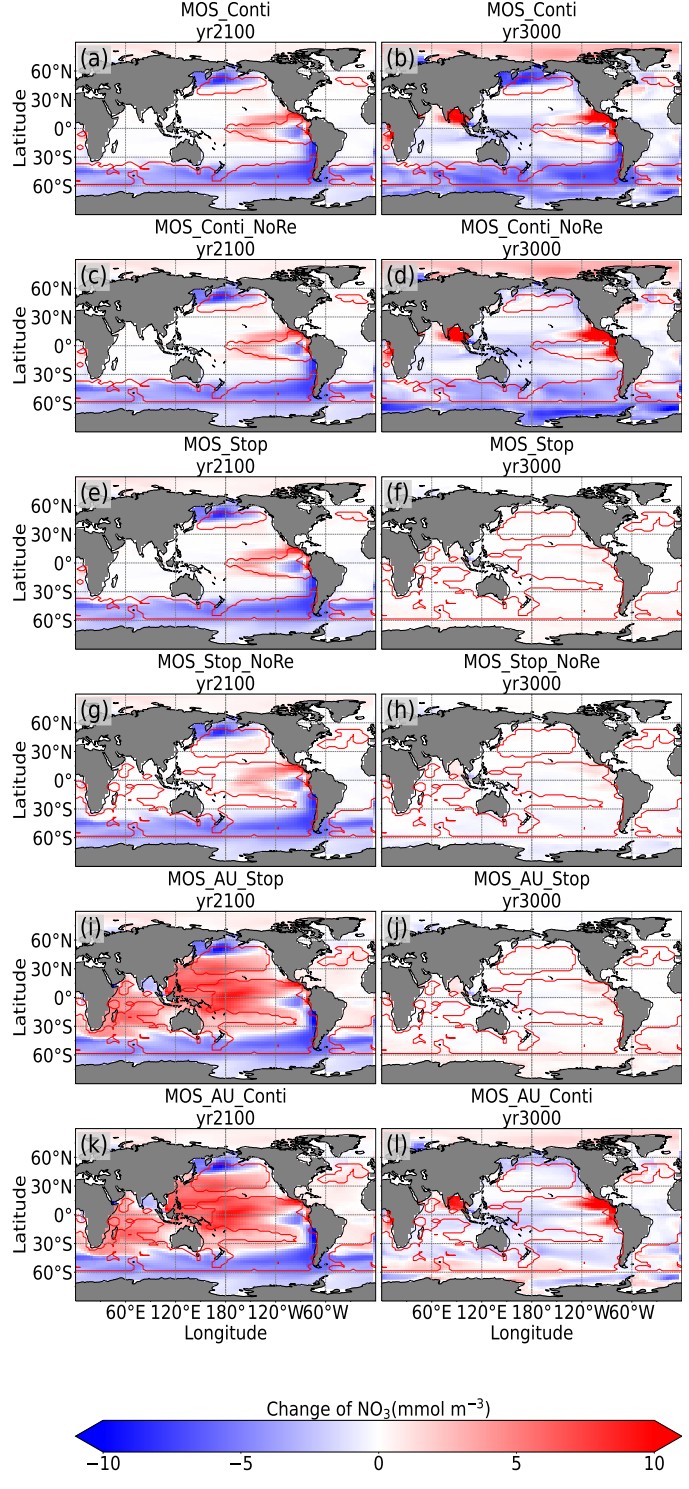

**Figure B9.** Redistributions of NO$_3$ avg. 0-200m depth relative to RCP4.5

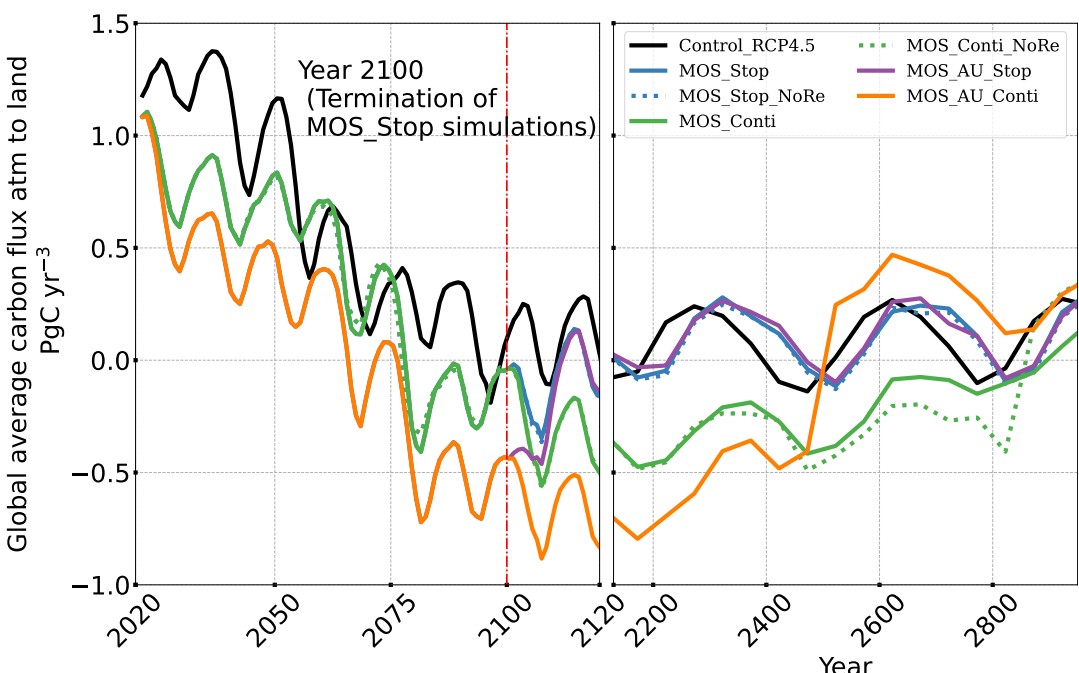

**Figure B10.** Global averaged carbon flux from atmosphere to land

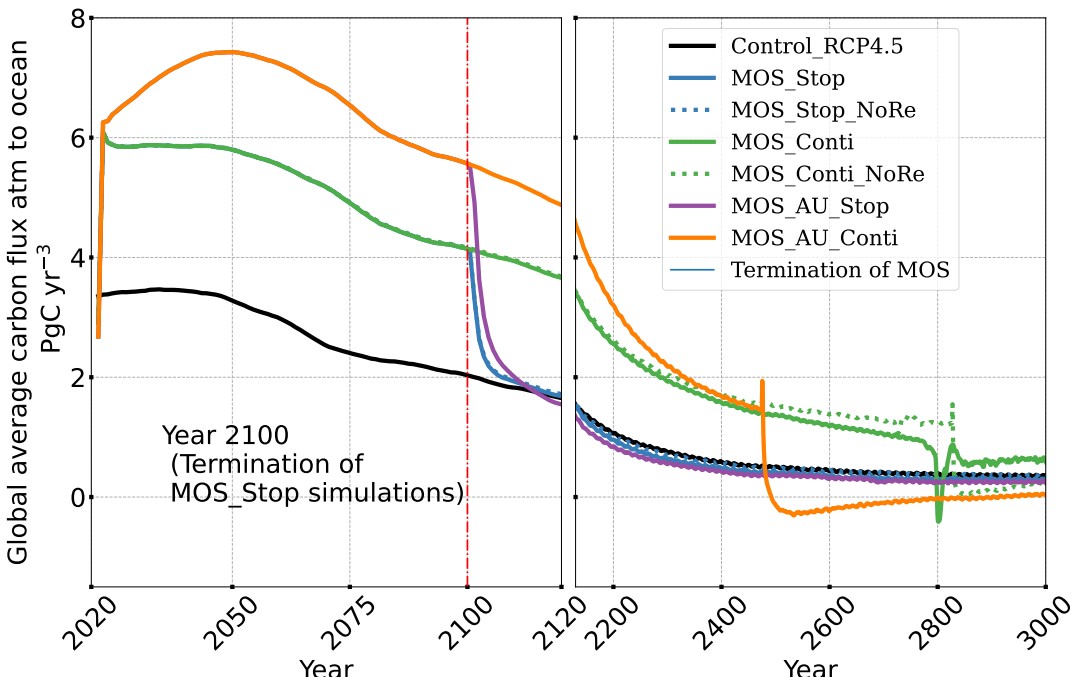

**Figure B11.** Global averaged carbon flux from atmosphere to ocean

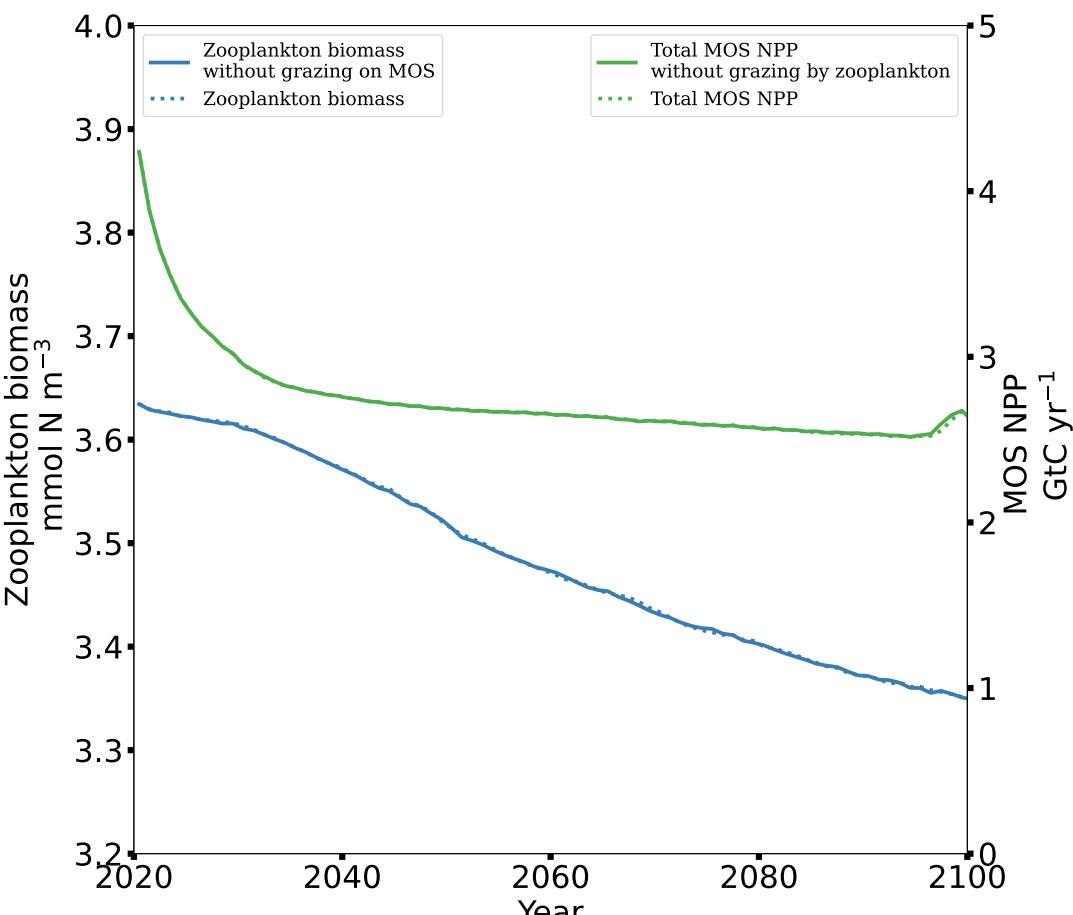

**Figure B12.** Global profiles of zooplankton biomass (left y-axis) & macroalgal NPP of MOS (right y-axis) in comparison with/without zooplankton grazing on macroalgae. The "zooplankton" communities do not have large effects, via grazing, on macroalgae NPP nor its own biomass.

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
