# Peer review of "Carbon Dioxide Removal via Macroalgae Open-ocean Mariculture and Sinking: An Earth System Modeling Study"

_Earth System Dynamics, 2021_

## Author Comment (AC6)

MOS_Conti/Stop
yr2100

(a)

MOS_Conti
yr3000

(b)

MOS_AU_Conti/Stop
yr2100

(c)

Longitude

MOS biomass (kg wet weight m$^{-2}$)

---

## Author Comment (AC7)

MOS_Conti/Stop
yr2100

(a)

MOS_Conti
yr3000

(b)

MOS_AU_Conti/Stop
yr2100

(c)

Longitude

MOS biomass (kg wet weight m$^{-2}$)

---

## Author Response (AR2)

**Response to Community Comment from K.Caldeira**

**Reviewer's comments:**

This seems like a nice paper addressing an important topic. Thank you, authors !!

It would improve the paper to perform additional simulations with atmospheric CO2 concentrations specified to be the same as the Control_RCP4.5 simulation or held at per-industrial concentrations.

If we were doing a study on the efficacy of geologic storage in a leaky reservoir, we would not need to model the terretrial biosphere to see the terrestrial biosphere response to the leaked carbon. It would be enough to assess the geologic storage by knowing the leak rate.

For example, let's see I remove a ton of CO2 from the atmosphere using kelp. If we allow atmospheric CO2 to respond, their will be outgassing of CO2 out of the rest of the ocean. This same thing would happen if we used direct air capture on land, but we would not say that the efficacy of direct air capture on land is diminished by ocean outgassing. We would say that the ocean responds to an atmospheric CO2 removal oppositie in sign as it would respond to an atmospheric addition.

Another way of thinking about a constant (or specified) atmospheric CO2 removal is it is answering the question: how much more (or less) co2 could be emitted to the atmosphere to maintain the same concentrations. With direct air capture, we say one ton is removed because we could emit an additional ton with no net effect on the atmosphere.

This issue is discussed in Chapter 6 of the IPCC SRCCS (https://archive.ipcc.ch/report/srccs/) Box 6.3 on page 293. "Fraction retained" is the metric of interest.
* * *
**Authors responses:**

Dear Ken Caldeira,

Thank you very much for your very useful and constructive comments on our manuscript. We appreciate your concern about correctly quantifying the amount of carbon removed, and have carried out a few additional model runs with reduced emissions to compare against the various possible metrics for calculating carbon removal via Macroalgae Open-ocean mariculture and Sinking (MOS). There are some differences compared to geological storage of $CO_2$ because sinking of macroalgae to the ocean's abyss will not only remove carbon but also nutrients from the surface ocean in contact with the atmosphere, with consequences for the marine biology and carbon cycle.

[Figure]

**Fig. 1**. The solid lines refer to the reduction in atmospheric $CO_2$ by normal MOS and MOS_AU. The **dotted** and **dashed** lines refer to the reduction in atmospheric $CO_2$ achieved by emission reduction **1.0** and **0.8** times the total CDR of MOS and MOS_AU, where CDR is measured as the amount of macroalgae carbon sunk to the sea floor. Red and cyan colors represent MOS and MOS_AU.

To estimate the size of $CO_2$ emissions offset by MOS, we performed a series of simulations without MOS, but with reduced $CO_2$ emissions relative to the RCP4.5 scenario by an amount proportional to the total CDR by MOS and MOS_AU experiments from the year 2020 to 2100.

In the original MOS simulation, the reduction of atmospheric $CO_2$ (142.6 Pg C) accounts for ca. 53% of the total carbon sequestration by MOS (270 PgC), which the smaller airborne reduction being to a large part due to $CO_2$ back fluxes between terrestrial, oceanic and atmospheric reservoirs - in the same way as the airborne fraction of $CO_2$ emissions is only about half. As discussed in Sec. 4.3.2, the offset of MOS CDR compared to an emission reduction by the same magnitude is generally caused by the reduced phytoplankton NPP (PNPP) due to nutrient competition and canopy shading effect of MOS.

**Table 1.** Amount of $CO_2$ emissions avoided in the emission reduction runs by year 2100 and reduction in atmospheric $CO_2$ in emission reduction runs without MOS and in MOS simulations.

| | $CO_2$ Emission reduction vs. RCP4.5 (PgC) | Atm. $CO_2$ reduction by yr2100 vs. RCP4.5 (PgC) |
|---|---|---|
| **MOS** | / | 142.6 |
| **Cut 1.0xMOS_CDR** | 270 | 171.5 |
| **Cut 0.8xMOS_CDR** | 216 | 137.7 |
| **MOS_AU** | / | 230.7 |
| **Cut 1.0xMOS_AU_CDR** | 446.8 | 279.1 |
| **Cut 0.8xMOS_AU_CDR** | 357.4 | 225.8 |

In a model run without MOS but with $CO_2$ emissions reduced the annual equivalents of the MOS-induced carbon exports, yielding a total amount of 270 PgC by year 2100 (**Cut 1.0xMOS_CDR,** Table. 1), the 270 PgC emissions removal yields a reduction of atmospheric $CO_2$ by 171.5 GtC by year 2100 (solid red line in Fig. 1). This is 20% more than the atmospheric CO2 reduction of 142.6 PgC realized in the original MOS simulation where the MOS-induced shading and removal of nutrients from the surface layers reduces the biological carbon pump and the associated carbon storage in the ocean.

With emission cuts 20% less than the MOS-induced carbon export (**Cut 0.8xMOS_CDR**), atmospheric $CO_2$ concentrations simulated by the MOS-free emission-cut runs agree closely with those of the respective MOS experiments (dashed lines in Fig. 1). That is, each ton of $CO_2$ sequestered in the ocean by MOS is, in our model and on a 100 year timescale, equivalent to an emission cut of about 0.8 tons of $CO_2$.

Similar trends are obtained for the MOS_AU tests (Tab. 1 and Fig. 1).

We have added these information in the revised manuscript at the last paragraph of Sect. 4.3.2, which reads:

"The current model results provide additional evidence that the CDR potential of MOS is partly offset by its negative impacts on the pelagic biological production and the biological carbon pump. In an additional model run (not shown) without MOS but with $CO_2$ emissions reduced by the annual equivalents of the MOS-induced carbon exports, yielding a total amount of 270 PgC by year 2100, the 270 PgC emissions removal yields a reduction of atmospheric $CO_2$ by 171.5 GtC by year 2100. This reduction is 20% more than the atmospheric $CO_2$ reduction of 142.6 PgC realized in the original MOS simulation where the MOS-induced shading and removal of nutrients from the surface layers reduces the biological carbon pump and the associated carbon storage in the ocean. When $CO_2$ emissions are instead cut by an amount corresponding to 80% of the MOS-induced carbon export, atmospheric CO2 concentrations simulated by the MOS-free emission-cut runs agree closely with those of the respective MOS experiments. That is, each ton of $CO_2$ sequestered in the ocean by MOS is, in our model and on a 100 year timescale, equivalent to an emission cut of about 0.8 tons of $CO_2$."

On another topic: A key issue is what are the C:N:P ratios in the macro-algae export relative to the background organic carbon export. For the macroalgae C:N is 20 and C:P is 111, so that is C:N:P to be 111:5.6:1.

I don't think you report the C:N and C:P ratios for the planton export. I think Eby et al 2012 that you cite uses 106:16:1.

I am going to guess that the main reason that the macroalgae does something is because you can export 20 molC for each molN with the macro-algae but only 6.6 molC for each molN with phytoplankton. Is this understanding correct?

If so, why don't phosphate constraints govern? The 111 vs 106 differs by less than 5%. So why does using macroalgae give you only a 5%.

I guess my main issue with this paper on a superficial quick read is that I do not understand why the authors got the result that they got.

1. What if the Redfield ratios of the macroalgae were the same as the Redfield ratios of the phytoplankton? Would the macroalgae then be effective?

2. What if the remineralization depth of the macroalgae were the same as that of the phytoplankton? Would the macroalgae then be effective?

In short, and I admit a superficial reading, I don't understand why you obtained the results you obtained.

A good thing to ask yourself in any modeling study is: What assumption would have to be false to lead to a qualitatively different conclusion? This is often a good way to state the factors that are critical to your conclusions and what features of your model are additional ornamentation not central to your result.

Citation: https://doi.org/10.5194/esd-2021-104-CC1

**Response:** Sorry for the confusion, we will use this opportunity to clarify how the model works and why we obtained the results that we did. First, the model assumes that the detritus (or export) from macroalgae erosion will be directly converted back to nutrients and DIC (please see the explanation after Eq. 15 in Sect.2.2.1)Second, macroalgae detritus is not exported in the traditional way in the model with sinking and remineralization along the way, but instead the model assumes that the macroalgae biomass is instantly sunk to the seafloor (i.e., organic C, N and P are instantly transferred from the macroalgae farm at the surface directly to the seafloor with no remineralization along the way). Remineralization of macroalgae biomass can then occur, but only at the seafloor. The assumed macroalgae C:N:P ratio (see lines 100-101 for references), which is much higher than the ratio typically found in phytoplankton, is one of the reasons why sinking macroalgae has been proposed as a CDR method. The other reason is that directly sinking the biomass would avoid remineralization on the way down and thus, much carbon would be efficiently exported to the deep sea.

To answer your 1st question, if the standard Redfield ratio were used for macroalgae instead of the one that we used, then no, the simulated CDR would not be as effective. However, by directly moving biomass from the ocean surface to the seafloor, remineralization along the way would be avoided and the approach would still be somewhat more effective at transferring carbon to the deep ocean than normal biological pump processes.

The answer to the 2nd question is also similar. If detritus from macroalgae were required to sink naturally to the seafloor, then much of the C would be lost along the way, as currently happens to phytoplankton biomass via the biological pump. Due to the macroalgaes' higher C:N:P ratio, more C would end up in the deep ocean, but how much is an open question (see discussion on this in Krause-Jensen, D., & Duarte, C. M. (2016). Substantial role of macroalgae in marine carbon sequestration. *Nature Geoscience*, 9(10), 737–742. https://doi.org/10.1038/ngeo2790).

We have added some text to the introduction to address the issues that you raise, by pointing out that the C:N:P ratio of macroalgae is higher than that of phytoplankton in the UVic ESCM model. In the introduction we have pointed out (line 65-67) that by sinking macroalgae biomass directly to the seafloor we avoid the loss of much organic C along the way. These texts are:

"...... The assumed constant C:N:P ratio is 400:20:1, which is higher than the stoichiometric ratio of the general phytoplankton in the UVic ESCM (C:N:P=106:16:1, the Redfield ratio). The immediate transfer to depth can be thought of as a short circuiting of the biological pump by bringing marine biomass directly to the seafloor without having it remineralized along the way. ......"

**Response to Community Comment from Marius Wiggert**

**Reviewer's comments:**

Dear Jiajun,

Thanks a lot for sharing this very interesting paper with the community. I found it very insightful and calibrated how the intervention is modelled and how you explain the results.

I'm a bit confused by the outcomes from the seaweed growth model that you show in Figure 2 and 3.

The NPP peak of Area 3 in the southern hemispheric summer (Jan/Feb/March) makes sense to me. But I don't understand why NPP of Area 1 would peak in the northern hemispheric atuum/winter (November/December). Shouldn't it peak in nothern hemisphere summer Juli/August when sunlight is abundant? Or is the growth nutrient limited at that time?

Similarly, I don't understand the NPP curve for Area 2 in August/September: In the tropics there's ample sunlight all year round so the only expalantion for the NPP peak would be nutrient limitation in other times of the year, which I haven't seen in any of the other papers looking at nutrient distributions throughout the year.

Is this potentially an artifact of the macro-algae species you chose? I would appreciate some more references/details on this. Thank you,

Marius Wiggert

Citation: https://doi.org/10.5194/esd-2021-104-CC2

**Authors responses:**

Dear Marius Wiggert,

Thank you very much for your comments on our manuscript.

We apologize for an error in the legend in Figure 3 (erroneously interchanging Area I and II) and have corrected this in the new version below and also updated in the upcoming revised version of our manuscript.

According to **Sec. 3.1** and **Tab. 2**, the seeding date for MOS is **1st of May in Area I** and **1st of January in Area II**, while the sinking dates are **31st October and 31st December** correspondingly. As a result, the macroalgae NPP in **Area I** peaks around September with the accumulation of macroalgae biomass in that area.

In **Area II**, due to the nutrient limitation and nutrient competition with ambient phytoplankton, macroalgae biomass grows slower than in the other two areas, leading to a later peak of NPP around October.

[Figure]

**Figure 3**. Vertically integrated macroalgae NPP simulated by experiment MOS (solid lines) for year 2024 and representative Areas I (dark blue circle), II (yellow pentagon) and III (cyan rectangle) highlighted by rectangles with corresponding colors in Fig.2. The NPP of macroalgae of experiment MOS_AU (dashed) shows an enhancement of NPP as expected.

Thank you for pointing out this lack of detail in the description of the seasonality shown in Figure 3. We will include the text in the manuscript in the 2nd paragraph of Sect. 4.1.2.

Thank you again for your attention to our manuscript!

**Response to Referee Comments from Dr. Bach**

Wu et al. investigate seaweed farming for atmospheric CO2 removal (abbreviated as 'MOS' in their study) in an Earth System model. They do so by plugging in a macroalgae model into UVic, making the biological carbon pump ultra-efficient, mainly by no remineralization losses during sinking of seaweed carbon and C/N ratios of 20 for

macroalgae primary production (instead of presumably 6.6 phytoplankton primary production). Their study is timely, important, and interesting. I have, however, several major comments, which I think need consideration and hopefully improve the manuscript.

Dear Dr. Bach,

We appreciate your time and efforts to provide feedback on our manuscript and are grateful for the insightful comments on and valuable improvements to our manuscript "Carbon Dioxide Removal via Macroalgae Open-ocean Mariculture and Sinking: An Earth System Modeling Study". We will incorporate  your suggestions in the revised manuscript. Please see below for a point-by-point response to your comments and concerns.

**Major comments:**

· Zooplankton grazing on macroalgae:

The authors implement grazing on macroalgae by the generic zooplankton group in the model. I was curious because I have never heard of zooplankton grazing on macroalgae. The authors cite a modeling study as evidence (Baird et al. 2003) but this study says: "In the model, large zooplankton graze on large phytoplankton and microphytobenthos, while small zooplankton graze on small phytoplankton." This study does not simulate macroalgae grazing by zooplankton according to this sentence and therefore it cannot be taken as evidence for this mechanism.

Thank you for pointing out this. In the revised MS, we have removed the reference of Baird et al., 2003. See the response below for further explanation of our parameterization.

Later in the text the authors reference another modeling study (Trancoso et al., 2005) but this study does not provide evidence either but instead refers to a 700+ page textbook to justify their zooplankton grazing preference for macroalgae of 0.0008. Based on this, it is quite nebulous where the justification for zooplankton grazing on macroalgae comes from. Introducing this trophic link and the associated matter flux may not be appropriate. But I have too limited understanding of the model to understand if this parameterization has noticeable influence on any relevant outcome. The authors should justify/clarify this issue.

Thank you for the opportunity to clarify the parameterization used in the model. The confusion about having pelagic zooplankton graze on macroalgae likely comes about because we used the original terminology of such models, which designates all higher trophic levels as "zooplankton". We admit that there is little evidence that pelagic zooplankton graze on macroalgae. However, in our model "zooplankton" represent all higher trophic levels, including known macroalgae grazers such as amphipods (Jacobucci et al., 2008), gastropods (Chikaraishi et al., 2007;  Krumhansl et al., 2011), sea urchins (e.g. Yatsuya et al., 2020) and fishes (e.g. Peteiro et al., 2012). Thus, we included this food web pathway of the model to assess the sensitivities of macroalgae to potential grazers in the ocean, assuming that with large macroalgae farms the pelagic larva of some grazing organisms like fish or urchins, would settle within the farms. Text will be added to the manuscript to clarify this point and better explain what "zooplankton" actually represents in the model.

However, it is worth noting that even with the assigned grazing preference ($\psi = 1 \times 10^{-4}$), the "zooplankton" communities do not have large grazing impacts on macroalgae biomass. In the revised manuscript we will illustrate this via a sensitivity experiment where zooplankton does not graze on macroalgae but follows the original grazing preferences in UVic ESCM.

· The importance of CN ratios:

From my understanding, the key parameter determining the CDR potential of MOS in their model is the CN ratio of macroalgae (assumed to be 20), or more precisely the delta_CN relative to phytoplankton. I assume the UVic phytoplankton CN is 6.6 (this should be mentioned in the text) so the delta_CN is 13.4. Surprisingly, there is no sensitivity analysis on this important parameter which should largely determine the CDR (e.g. 270 PgC until 2100) as given in the abstract. There are two issues with this:

When I utilize our own calculation from Bach et al. (2021), which calculated the CDR dependency of MOS as a function of phytoplankton and macroalgae CN, then I calculate with your numbers (20 and 6.6) a reduction of the CDR potential of 33%. This is very similar to your 37%, which is encouraging. However, the important point is that our calculation underlined the massive dependency of the MOS CDR potential on the assumed phytoplankton/macroalgae CN combinations (I attached the figure below). For example, just mildly increasing phytoplankton CN from 6.6 to 8 (8 being more realistic for (sub)tropics; Martiny et al., 2013), then CDR reduction increases from 33% to 40%. I

would assume that similar sensitivities occur in their model, so that it is at least important to emphasize how influential the CN assumptions are.

**Response:** Thank you for pointing out this important point concerning the potential influences of the stoichiometric CN ratio of macroalgae on the CDR capability of MOS. According to your suggestions, we have performed additional sensitivity simulations (MOS_Conti ± 120% of the original macroalgae molar C:N ratio) to address this issue. The result indicates that the CDR capacity of MOS is sensitive to the molar C:N ratio of macroalgal biomass. Compared to MOS_Conti, the carbon captured by MOS (MOS-C) in MOS_Conti_CN_High increases by 22% (by the year 2100) and 19% (by the year 3000) when the macroalgae C:N ratio is increases by 20% to 240:10. When the macroalgae C:N is 20% lower than the original value, MOS-C decreases by 13% (by the year 2100) and 18% (by the year 3000). Our results agree with the range of CDR potential reduction by nutrient reallocation (7-50%) reported in Bach et.al., 2021.

We have added these results and analysis in the revised manuscript.

The authors provide the CDR potential and provide e.g. 270 PgC until 2100 for the scenario without artificial upwelling. I believe this number is based on macroalgae-specific carbon sequestration. If this is the case then this number is misleading because the growth of macroalgae resulted in a substantial decline in phytoplankton carbon sequestration. The MOS CDR potential should be provided as CDR_MOS – CDR_phytoplankton to account for the substantial reduction of an already existing carbon sink.

**Response:** Sorry if there is any confusion over how much CDR is done and how we define it. In table 5 we actually give several numbers including how much C is sequestered in the sunk macroalgae biomass (the 270 Pg C you mention above) and how much atmospheric $CO_2$ has been reduced, which for the above example is only a reduction of 142.6 Pg C. The difference between what is sequestered by the macroalgae and how much atmospheric $CO_2$ decreases is due to several C cycle responses and the direct effect of MOS on the ocean. These effects include the decline in phytoplankton CDR as a result of shading and nutrient removal by MOS, and also the C-cycle responses of the land and ocean to lower atmospheric $pCO_2$ when MOS is deployed (see Keller, D. P., Lenton, A., Littleton, E. W., Oschlies, A., Scott, V., & Vaughan, N. E. (2018). The Effects of Carbon Dioxide Removal on the Carbon Cycle. *Current Climate Change Reports*, *4*(3), 250–265. https://doi.org/10.1007/s40641-018-0104-3). We will

explain the differences between MOS-induced carbon export, air-sea carbon fluxes and reduction of atmospheric $CO_2$ in more detail in Section 4.3.2 of the revised version of the manuscript. In response to the comments by Ken Caldeira, we also added runs without MOS but with $CO_2$ emissions reductions and found that one ton of $CO_2$ sequestered in the ocean by MOS is, in our model and on a 100 year timescale, equivalent to an emission cut of about 0.8 tons of $CO_2$. This will now also be included in the revised manuscript.

· Artificial upwelling:

The authors test if nutrients acquired from artificial upwelling could boost MOS. Artificial upwelling is implemented by upwards movement of nutrients but the upwards movement of cold temperature is avoided. I understand the rationale for this (I guess you don't want to make this an artificial upwelling paper and therefore 'engineer' a perhaps rather unrealistic counter-current heat exchange into the pipes). However, for the not so familiar reader this paints a very rosy picture. As you have shown in the 'sorcerer's apprentice paper' (Oschlies et al., 2010), upward movement of temperature enhances heat absorption by the oceans so that Earth absorbs more heat than without artificial upwelling (and will be warmer than without artificial upwelling once it is stopped). For me this is such an extreme side-effect that I would think artificial upwelling has very little chance for large-scale implementation. The problem I am having is that the neglect of AU side-effects in the paper could make this addition to MOS sound more attractive than it may be. I think this needs to be thoroughly explained.

Thank you for noting this important discrepancy. In this study, we avoided the pumping of cold water by artificial upwelling (AU) by using a hypothetical AU system that keeps temperatures at ambient levels (e.g. via heat exchangers). As you mentioned above, it has been revealed that the upwelled cold waters by ordinary AU may cause significant side effects, such as the quick rebound, or even surpassing, of $CO_2$ concentrations and surface temperatures after the termination of AU (Oschlies et al., 2010). Meanwhile, if the cold water upwelling was included in this study, there would be extra analysis required on the associated impacts on planetary radiation budget imbalance, marine biogeochemistry and global carbon pool (e.g., the enhanced terrestrial carbon sequestration due to the cooling effects by AU). All these points are interesting to discuss, however, they are beyond the scope of this study.  Our main object in this study is to analyze the maximum CDR potential of MOS, and the AU systems are aimed at

providing extra nutrients that supply the macroalgae of MOS. Therefore, we decided to avoid the simulations of MOS with ordinary AU in the manuscript. This reasoning will now be stated with more detail and also a more cautionary note regarding the highly idealized scenario.

Here we show a comparison of MOS with ordinary AU (MOS_AU_Conti_with_heat) with the MOS_AU_Conti and RCP 4.5 in the following **Figure 1**. As shown in the left panel, the decline of $CO_2$ concentration in MOS_AU_Conti_with_heat is slightly slower than MOS_AU_Conti. This can be explained by the slackened macroalgae NPP due to the ambient upwelled cold water. As expected, the upwelling of cold water by ordinary AU leads to a surface temperature drop during the first decades after deployment. However, the drop in surface averaged temperature (SAT) cannot be maintained due to higher atmospheric $CO_2$ and slowly warming subsurface waters that become source waters of the artificial upwelling. After a few hundred years, SAT is even higher in MOS_AU_Conti_with_heat than the MOS_AU_Conti with elimination of heat exchange.

These results indicate that the ordinary AU systems will limit the potential of MOS in CDR and global warming mitigation, due to the upwelling of cold water to the surface. However, it is quite difficult to isolate the contribution of upwelled cold water and the CDR by MOS to the decline of SAT.

Although the MOS experiments with ordinary AU will not be included in the manuscript, we have added some discussion in the chapter 5 **Concluding discussions** to address the discrepancy, which states:

 "Here we only discuss the CDR potential of MOS_AU combination under an idealized situation, where the artificial upwelling (AU) upwells nutrients without transporting heat and thereby changing the ambient water temperature. The aim of deploying this AU with MOS is to assess the maximum potential of CDR under AU-induced fertilization. However, it can be expected that the NPP of MOS will be slower if the ambient water temperature is reduced by the upwelled cold water upwelled, which will further limit the CDR potential of MOS_AU. Moreover, several dominant side effects due to the upwelled cold water in ordinary AU have been revealed, such as the quick rebound or even surpassing of $CO_2$ concentrations and surface temperatures after the termination of AU (Oschlies et.al., 2010b). A more detailed analysis of these physical effects would be required when ordinary AU were to be deployed in association with macroalgae farming."

[Figure]

**Figure 1.** Simulations of annual global mean atmospheric $CO_2$ concentrations (left panel) and surface averaged temperature relative to the pre-industrial (right panel). The green lines (MOS_AU_Conti_with_heat) represent the continuous MOS_AU with ordinary AU systems which upwell cold water.

· Environmental context:

The authors 'bombarded' the oceans with MOS. Accordingly, they yield very high amounts of CDR. This is an important finding and the necessary first step but I think the discussion on side-effects does not reflect the massive change superimposed by modelled MOS appropriately (with the exception of de-oxygenation, which is covered very well). Looking at Fig. 7, it looks as if phytoplankton net primary production (PNPP) is largely replaced by macroalgae NPP (MNPP) in places where MOS was implemented (e.g. Eq. Pacific or Southern Ocean). The authors provide a global mean for the reduction of PNPP (37%) but this distracts from the problem that this reduction may be locally much higher. It seems like the CDR potential calculated here comes at the cost of an almost complete replacement of phytoplankton food webs in some regions (e.g. Eq. Pacific or Southern Ocean). For the productive Eq. Pacific this sounds like replacing the rain forest with palm oil plantations. Such a requirement for a deep transformation of entire marine ecosystems needs to be mentioned alongside the numbers that suggest a high CDR potential, just to illustrate what it really takes to achieve those numbers. (I am mainly stumbling over the second last sentence of the abstract which sounds almost encouraging but is it really encouraging considering the environmental transformation?).

One suggestion to illustrate that would be a world map that shows %NPP of macroalgae in 2100 (i.e. MNPP/(MNPP+PNPP)*100).

**Response:** Thank you for pointing out these very valid concerns, which we share. We appreciate your suggestion of presenting regional shifts rather than only global relative changes of phytoplankton NPP and have produced the map of %NPP of macroalgae as suggested by the reviewer (see below). In the revised version we will add more text in Section 4.3.4 to better illustrate the impacts on NPP and the implications that this would have for marine food webs, which read:

"Fig.8 illustrates the shift of oceanic NPP from PNPP to MOS_NPP. In regions where MOS is deployed, 70% of the total NPP is macroalgae NPP. The macroalgal NPP is thus nearly twice as high as PNPP. This may lead to additional ecological and biogeochemical issues. One of them is the decline of zooplankton in this study. We performed an additional simulation, in which the zooplankton grazing on MOS is turned off, and the grazing preferences follow the original settings in Keller et al., 2012. As shown in Fig.B12, the grazing by zooplankton on MOS has no significant effect on neither the zooplankton biomass nor the MOS_NPP. As the zooplankton grazing preference for macroalgae is lower than for phytoplankton, the zooplankton community is still mainly fed by phytoplankton. Therefore, the decline in zooplankton biomass (Fig.B4) follows the declining phytoplankton biomass trend (Fig.B6)."

[Figure]

**Figure 8.** Proportion of MOS NPP (MOS_Stop/Conti) in the global oceanic NPP by year 2100 (MOS_NPP/(MOS_NPP + PNPP) x 100). Note that the NPP values are converted to carbon using each own C:N ratio. The MOS NPP generally amounts to more than 70% of total oceanic NPP where MOS is deployed, indicating an obvious NPP shift from phytoplankton PNPP to MOS NPP.

Other comments:

· Line 54: Sherman Ref missing.

**Response:** Actually this is included in the reference list, which reads:

"Sherman, M. T., Blaylock, R., Lucas, K., Capron, M. E., Stewart, J. R., DiMarco, S. F., Thyng, K., Hetland, R., Kim, M., Sullivan, C.,Moscicki, Z., Tsukrov, I., Swift, M. R., Chambers, M. D., James, S. C., Brooks, M., Von Herzen, B., Jones, A., and Piper, D.: SeaweedPaddock: Initial Modeling and Design for a Sargassum Ranch, in: Ocean. 2018 MTS/IEEE Charleston, Ocean 2018, pp. 1–6, IEEE, https://doi.org/10.1109/OCEANS.2018.8604848, 2019".

Or did you mean some other paper by Sherman that we are not aware of?

· Line 60: The authors stress that there are now well-funded research efforts going into the development (engineering) of seaweed platforms making MOS 'no longer fictional'. This text suggests, that there haven't been such efforts before, which is not the case. The US DOE (I think) has put significant money into this idea already in the 80ies (see review by Ritschard, 1992). So it wasn't fictional back then but because the ocean destroyed the infra-structure it seemed to have returned to fictional after this early reality-check revealed how hard it really is. This text should be amended to include these early trials (also indicating how old the idea really is).

**Response:** Thank you for noting this and sorry for missing these earlier references. Text will be added to the 4th paragraph of the Introduction pointing out this early work and citing Ritschard, 1992, which reads:

"In the 1970's, the concept of ocean farms using macroalgae as a marine carbon sink and for bioenergy production was studied with an actual small test farm established off the coast of southern California. These research activities were abandoned due to the damage of the test farm by winter storms and several technical and economic reasons (Ritschard, 1992)."

· Line 77: I stumbled over the word 'comprehensively'. While I totally see the massive value of testing MOS in an ESM, I would say an assessment is only comprehensive when it includes 1) Models 2) experiments 3) assessments using natural analogues. Maybe the word is just a bit too much as ESMs still miss much of the real-world complexity and many relevant processes (e.g. DOC cycling).

**Response:** Yes, we fully agree with the reviewer that this phrasing was not correct, even though we meant it in the narrow sense of evaluating atmospheric $CO_2$ drawdown by accounting for global C cycle responses. We will replace 'comprehensively' by 'in a global carbon cycle context'.

· Line 99: This is understandable, but has major implications for the overall conclusion concerning CDR potential (see major comment**).**

**Response:** As also stated in our response to the major comment above, we have performed a sensitivity analysis using different C:N ratios of macroalgae. Results of these experiments will be presented in the revised version to better illustrate the sensitivity of our results to this assumption. We will also expand the discussion of how the next steps in modeling of MOS need to account for variations in essential parameters of macroalgae dynamics.

· Line 107: Including iron? Much of the area considered for MOS in this study would be Fe-limited. Can we simply assume that coastal seaweeds with very low surface/volume ratio would have any chance in Fe-competition against mostly small (i.e.high S/V ratio) and specialized open ocean phytoplankton? Or do the authors assume Fe comes from the platform? I get that iron may perhaps be too premature to be included in this study (as there seems to be little data) but iron is not mentioned once in the manuscript. It may be worthwhile to at least mention this limitation somewhere.

**Response:** Thank you for pointing out this issue. Indeed, iron limitation of macroalgae is not considered in the current study. We agree that most of the ocean regions occupied by MOS are iron-limited for phytoplankton, but we do not know the limitation effect on macroalgae. While we can expect some iron release from the platform, we will acknowledge the lack of information and model parameterisations of iron limitation in macroalgae (both in the methods section and discussion).

· Line 122: Is the species-specific optimum as in equation 7 for a specific species (e.g. Laminaria) or is that a generic equation?

**Response:** The species-specific optimum temperature is an optimum temperature for the specific macroalgae species. Considering both the reported optimum temperature of genus *Saccharina* (reported range 13-30) as well as the observed global distribution of macroalgae, we chose the value of 20°C for a generic optimum temperature. This will be

explained in more detail in the revised manuscript. We will also update the reference for optimum temperature in Tab. 1.

· Line 125: I had problems understanding eq. 9, perhaps because Tmin and Tmax were not explicitly introduced. Assuming they mean min and max T at which a species can grow (?) that would mean that there are only three possible cases for temperature (Topt, Tmin, Tmax)?

**Response**: Sorry for not including definitions of what Tmin and max are. Tmin/Tmax is the lower/upper temperature limit above/below which growth of macroalgae ceases. We will add a sentence before this set of equations that explicitly define them.

· Line 153: The immediate remineralization may be a critical assumption, if the fraction of eroded biomass is high. Can you indicate how high the erosion typically is?

**Response:** The parameterization for immediate remineralization was implemented to minimize the computational expense of the model, as explicitly including particulate detritus from eroded biomass would require an additional state variable. Some phytoplankton biomass is treated in a similar manner in a "fast recycling" loop designed to simulate some DOM and microbial loop dynamics (see Keller et al., 2012). We agree that if eroded biomass were high and the material were to rapidly sink or be transported out of the grid cell, then there would be biogeochemical dynamics that the model does not adequately capture. However, the rate of erosion is 0.01% per day, i.e. a few percent per year (see Table 1) and its detailed parameterization is therefore expected to have only a small impact on the dynamics.

To help the reader understand why we parameterized the model this way we will include some text stating that "this parameterization of erosion, a small biomass loss of 0.01% per day, pragmatically set as instantaneous remineralization rather than introducing another finite remineralization and finite sinking parameterization with difficult-to-constrain parameters.".

· Line 212: I did not understand this sentence. How was $pCO_2$ calculated separately for MOS_DIC? Is it added to the DIC pool and then the delta to the DIC pool?

**Response:** Sorry for the confusing expression. The air exchange of MOS_DIC shares the identical physical properties as the original DIC. We have deleted this sentence in the revised manuscript.

· Table 1: Reference numbers? Where can these be checked??? Hard to review because it would be great to know where all these numbers come from.

**Response:** Sorry for the confusion. These were in an early version, but were inadvertently left out of the final draft. The references have now been added directly to the table.

**Table 1.** Model parameters

| Symbol | Parameter | Unit | Value | Reference |
|---|---|---|---|---|
| $a_{ma}$ | Macroalgae carbon specific shading area | $m^2 kgC^{-1}$ | 11.1 | Trancoso et al. (2005) |
| d | Distance between the cultivating ropes | m | 10 | Van Der Molen et al. (2017) |
| $R_{erosion}$ | Individual erosion rate | $\% \, d^{-1}$ | 0.01 | Zhang et al. (2016) |
| $I_{opt}$ | Optimum light intensity for macroalgae growth | $W \, m^{-2}$ | 180 | Zhang et al. (2016) |
| $NO_3$ | Nitrate Concentration | $\mu mol \, l^{-1}$ | model calculation | Keller et al. (2012) |
| $PO_4$ | Phosphate Concentration | $\mu mol \, l^{-1}$ | model calculation | Keller et al. (2012) |
| $K_N$ | Half-saturation constant for nitrogen uptake | $\mu mol \, l^{-1}$ | 2 | Zhang et al. (2016) |
| $K_P$ | Half-saturation constant for phosphorus uptake | $\mu mol \, l^{-1}$ | 0.1 | Zhang et al. (2016) |
| $k_w$ | Coefficient of light attenuation through water | $m^{-1}$ | 0.04 | Keller et al. (2012) |
| $k_c$ | Coefficient of light attenuation through phytoplankton | $m^{-1}(mmol \, m^{-3})$ | 0.047 | Keller et al. (2012) |
| $M_{ma}$ | Thickness of MOS macroalgae canopy | m | 10 | Trevathan-Tackett et al. (2015) |
| $MR_{C:N}$ | Molar C:N ratio of macroalgal biomass | - | 20 | Atkinson and Smith (1983) |
| $MR_{P:N}$ | Molar P:N ratio of macroalgal biomass | - | 0.05 | Atkinson and Smith (1983) |
| $MR_{C:P}$ | Molar C:P ratio of macroalgal biomass | - | 200 | calculated |
| $MR_{DW:WW}$ | Ratio of DW to WW of macroalgal biomass | - | 0.1 (values reported:0.05~0.2) | Aldridge and Trimmer (2009); Conover et al. (2016); Van Der Molen et al. (2017) |
| $MR_{C:DW}$ | Carbon content of dried macroalgal biomass | % | 30 | Chung et al. (2011) |
| $MR_{N:DW}$ | Nitrogen content of dried macroalgal biomass | - | 0.16 | Duarte et al. (2003) |
| $R_{max20}$ | Maximum respiration rate at 20°C | $\% \, d^{-1}$ | 1.5 | Martins and Marques (2002); Zhang et al. (2016) |
| r | Empirical coefficient for macroalgae respiration | $d^{-1}$ | 1.047 | Martins and Marques (2002) |
| Seed | Initial macroalgal biomass per kilometer cultivating line | $kgC \, km^{-1}$ | 2.5 | Van Der Molen et al. (2017) |
| $T_b$ | E-folding temperature of biological rates | °C | 15.56 | Schmittner et al. (2008) |
| $T_{opt}$ | Optimum temperature for growth | °C | 20(values reported: 13-30) | Zhang et al. (2016); Martins and Marques (2002) |
| $T_{max}$ | Upper temperature limit above which growth ceases | °C | 35 | Breeman (1988) |
| $T_{min}$ | Bottom temperature limit below which growth ceases | °C | 0 | Martins and Marques (2002) |
| $u_{max}$ | Maximum growth rate | $d^{-1}$ | 0.2 | Zhang et al. (2016) |
| w | Areal mean artificial upwelling rate | $cm \, d^{-1}$ | 1,10 | Oschlies et al. (2010b) |
| $Y_{max}$ | Maximum yield of macroalgal biomass on MOS | $t \, DW \, km^{-2} \, yr^{-1}$ | 3300 | |
| $\psi_{ma}$ | Zooplankton grazing preference on macroalgae | - | $1 \times 10^{-4}$ | Trancoso et al. (2005) |
| $\mu_{ma0}$ | Remineralization rate of sunk macroalgal biomass at 0 °C | $\% \, d^{-1}$ | 7 | Partanen et al. (2016) |

· Line 285: May be nice to have the temperature optimum curve for your seaweed in a figure.

**Response:** Thank you for the suggestion. We have added the following temperature optimum curve to the macroalgae model as Figure A3.

[Figure]

**Figure A3.** Temperature optimum curve of the macroalgae in MOS

· Line 312: These numbers are highly dependent on the CN ratio according to the equations provided above. This links back to the issue raised in the major comment above.

**Response:** Yes, we agree, see our response to your major comment. We will also add a sentence here, referring to the new sensitivity experiments with different C:N ratios and noting how our results depend on the fixed C:N:P ratio and that modeling macroalgae in this way leads to some uncertainty that may have been compensated for by tuning other parameters.

· Section 4.3.3. In addition to vertical changes in nutrient distribution, it would have possibly been equally interesting to talk about horizontal changes in nutrient distribution and associated changes in productivity in the surface ocean. This nutrient robbing issue is in my opinion one of the biggest problems in for MOS but not investigated here.

· It seems like there is some information on this matter provided in B8 but the figure shows local changes, not quite what is needed for this I think.

**Response:** We agree that horizontal changes in the nutrient distribution and associated productivity changes are important. We will add a few additional sentences on this to section 4.3.3 and include supplemental zonally averaged figures of nutrient changes with depth for the Atlantic and Pacific basins to illustrate these dynamics. The new text will read, "In addition to the localized depletion of nutrients by MOS, the MOS-induced Southern Ocean uptake and transport of N and P to the deep ocean acts as a type of "nutrient trapping". These dynamics thereby reduce nutrients and productivity in mid to low latitudes because less N and P are available to be transported northward out of the Southern Ocean. A similar dynamic has been seen in modeling studies of ocean iron fertilization (Oschlies et al., 2010; Keller et al., 2014)."

· Line 401: Should be B3

In the manuscript the sentence "Therefore, the declines in zooplankton biomass (Fig.B3) agree with the declining phytoplankton biomass trend (Fig.B5)." appears to be correct as the figures in the Appendix are:

Figure B3. Plot of global averaged biomass of zooplankton.

Figure B5. Plot of global averaged phytoplankton biomass.

Thus we think the references to appendix figures are correct here.

· Line 435: I wonder if the permanence of CO2 storage would be even longer if sediment carbonate dissolution due to high respiratory DIC at the bottom was occurring. Have you considered a CaCO3 sediment dissolution? May be worth mentioning.

**Response:** Thank you for pointing this out. We agree that this is something to be explored in a further study that includes a CaCO3 sediment model. We will add a line to the discussion section on this topic stating that, "Processes such as $CaCO_3$ dissolution at the seafloor, which could be affected by MOS and may impact its overall efficiency (in particular increasing durability of the carbon sequestration), also need to be included in future studies."

· Line 540: Throughout the manuscript I was wondering why the sensitivity study without remineralization of biomass at the bottom was made? Is there any reason to believe this could be the case?

**Response:** While we agree that it is unlikely that there would be no remineralization at all, having a zero remineralization case study allows us to do two things: First, we can use this sensitivity study as an extreme case of infinitely slow remineralization to help estimate the range of possible fates of remineralized organic matter. Second, this sensitivity study could represent a different macroalgae farming approach - that of harvesting the biomass to create bioenergy (on land, with carbon capture or storage) or biochar, with the assumption that all harvested biomass was permanently removed from the ocean. While this is a very idealized case, it serves the useful purpose of providing information on how marine biogeochemistry is impacted by the permanent removal of fixed C, N, and P. We have added the following sentences to the revised manuscript to better describe the idea behind this sensitivity study.

In the 5th paragraph of Sect. 3.2:

".... This sensitivity study can also simulate an extreme case of infinitely slow remineralization, which can help with estimating the range of possible fates of remineralized organic matter. Meanwhile, this sensitivity study could represent a different macroalgae farming approach - that of harvesting the biomass to create bioenergy with carbon capture or storage (BECCS) or biochar (e.g., Kerrison et al., 2015,Laurens et al., 2020,Roberts et al., 2015), with the assumption that all harvested biomass was permanently removed from the ocean. While this is a very idealized case, it serves the useful purpose of providing information on how marine biogeochemistry is impacted by the permanent removal of fixed C, N, and P."

In the ending of 6th paragraph of Sect. 5:

"These potential side effects are also noteworthy for another macroalgae farming approach, i.e. harvesting the macroalgae biomass for bioenergy with carbon capture or storage (BECCS) or biochar (Sect. 3.2)."

· Line 555: What is the difference between MOS and 'ocean afforestation'?

**Response:** Ocean afforestation is a term coined by N'Yeurt et al., 2012 and referred to growing macroalgae which could then be harvested to produce biofuels, with the carbon

captured and stored upon their combustion to achieve CDR. As this term added some confusion we will remove it from the sentence as it is not necessary.

· Line 562: I find this argument not well thought through. We have argued in Bach et al., 2021 that one needs to consider the production of halocarbons (or DMSP or any other climate relevant substance) relative to phytoplankton and the re-allocated nutrient consumption. If seaweeds produce less halocarbons per nutrient resource than phytoplankton than halocarbon production may decline (even though spatial aggregation /re-allocation may occur).

**Response:** Our argument was based on the assumption that per unit area, farmed macroalgae production of halocarbons is likely much higher than that of the pelagic phytoplankton communities that they would replace. Although of course, as we noted, macroalgae halocarbon production varies by species. However, from a global context you are correct, that if phytoplankton production of halocarbons decreased outside of the farmed area due to fewer nutrients being available, then what really matters is halocarbon production by either phytoplankton or macroalgae per nutrient resource. We will add text to this section to correct this oversight. The new text will read, "Large-scale MOS cultivation might release a significant quantity of halocarbons. However, as MOS also reduces global phytoplankton NPP, it is likely that the production of halocarbons by phytoplankton decreases. Further studies are needed to investigate possible net effects of halocarbon emissions from large-scale macroalgae cultivation and how this is offset by a potential decrease in phytoplankton halocarbon production."

· Appendix C: The RCP8.5 run is not really considered in the main text discussion. I see the value of it but may not be needed to have it in there if not discussed appropriately.

**Response:** Thanks for the comment. We agree and have removed this from the appendix.

**Response to Comments from Anonymous Referee**

**Authors present results of assessing the efficiency and impacts of macroalgal ocean sequestration (MOS) within an Earth System Model. The model assumes that MOS infrastructures appear throughout deep ocean sites where long-term sequestration is possible and macroalgal biomass will grow where there are adequate nutrients to sustain an annual crop. They then look at several MOS scenarios to address the long-term impacts on ocean ecosystems and the carbon cycle.**

**The paper is interesting and potentially important as many are looking to seaweeds as a CO2 removal (CDR) strategy. However, there are several issues that make it not quite ready for publication at this time in my opinion. I have issues with the overall premise of how the authors envision MOS, there are missing and odd elements of the farmed macroalgae model and the presentation is not adequate and the manuscript needs both a reorganization and some editing to make it more easily readable. I will detail these overall concerns and follow with specific comments by line number.**

**The basic MOS scenario created by the authors have created is in my opinion an unrealistic possibility which degrades the relevance of the model results presented. As I understand it, they are trying to assess is whether MOS (farmed everywhere it can) can by itself keep global temperatures within the Paris accord targets while still allowing moderate emission scenario (following RCP4.5). This seems to me to be an odd thing to test as I cannot imagine that actually happening. The recent NASEM report on ocean CDR suggests that portfolio of several CDR approaches is more likely solution to the negative emissions quandary. Further the presumption that MOS infrastructure can be deployed everywhere that macroalgae can grow requires a number of logistical hurdles to be overcome. Together, I am having a hard time understanding the actual relevance of these scenarios understanding the efficacy and impacts of MOS as a CDR strategy. Within that basic frame, the individual cases for stopping MOS, no decay of biomass and adding artificial upwelling cases, all make sense. But the overall premise does not, at least in my opinion.**

**Response:** Thank you for your thoughts on the manuscript, we are sorry that the intentions of our study were not more clearly communicated in the manuscript and we will revise it so that there is no confusion. The overall goal of the study is not to see if MOS

alone can limit warming enough to reach the Paris Agreement goals, this is as you say only potentially possible with a portfolio of CDR approaches and as much reduction in emissions as possible (emissions reductions are the primary means of meeting the Paris Agreement goals, CDR is no substitute). The study is also not supposed to suggest that macroalgae farming could be done everywhere. Instead the study is designed to explore the maximum natural physical/biogeochemical potential of open ocean macroalgae farming with sinking as a means of long-term CDR. This approach is valuable in a number ways. First, we can put an upper limit on what is physically / biogeochemically possible. Second, some side effects may only become evident at the very large scales found in our study. Understanding at this level is useful for deciding what CDR approaches to prioritize with further research. Finally, by initially seeding the macroalgae everywhere in a dynamic model, with simulated climate change, we can help determine locations where open ocean macroalgae farming is viable. This goes beyond previous studies (e.g. Lehahn et al., 2016; Froehlich et al., 2019), which used static models of environmental conditions to determine where macroalgae might initially be farmed in the open ocean. To make the aims of the study clearer, we will revise the text as follows.

Abstract: (1st sentence) "In this study we investigate the maximum physical / biogeochemical potential of macroalgae open-ocean…".

Introduction: (5th paragraph) ".... The aim of this study is to investigate 1) the maximum physical / biogeochemical CDR potential of MOS, 2) the side effects of such large scale deployment, and 3) to understand where offshore macroalgae farming would be viable if done at a large scale. This information is needed to help prioritize further research into CDR, to understand if there are potential MOS side effects that become evident only at large scale, and to provide information on the viability of large-scale offshore macroalgae farming in different regions over time by accounting for the implications of nutrient utilization and climate change."

**I have some quibbles with the modeling that I think requires some discussion. Carlos Duarte and his colleagues have focused on the importance of recalcitrant dissolved organic carbon (DOC) that is released as the farmed macrophytes grow to long-term carbon sequestration. This mechanism is not included in the model nor is it discussed why it is not included. Given the long lifetimes (1000's years) of the recalcitrant DOC pool in the ocean, even a small fraction of recalcitrant DOC released during growth could be important.**

**Response:** The production of recalcitrant DOC is something that we have thought about, however, refractory DOC dynamics are difficult to include in a global model and beyond the scope of this study. This is because as far as we are aware no global models have successfully simulated refractory DOC cycling in a dynamic manner that resolves all source and sink terms. Most attempts have used data constrained models (offline tracer-modeling techniques to constrain DOC)(Letscher et al., 2015) or are still at the theoretical stage of development (Mentges et al., 2019; Zakem et al., 2021) and not coupled to a 3-D physical ocean model, let alone an Earth system model. Global 3-D models that do include DOM cycling, usually only resolve labile or semi-labile pools of DOM with a limited number of DOM source and sink terms that are poorly constrained and thus, unable to realistically simulate DOM spatial distributions or concentrations (Anderson et al., 2015). This is why the marine biogeochemical model that we use as the basis for the work does not explicitly resolve DOC and is instead parameterized to implicitly include these dynamics in a manner that allows the model to simulate other key marine biogeochemical variables in a reasonable manner (Keller et al., 2012). Furthermore, there is also not enough information on the production and bioavailability of DOC from different macroalgae species to add a new parameterization to a model of DOM cycling, if one was available. Few studies exist and the uncertainties about the release of DOC are large. For example, in the Barrón et al., (2014) study, which is often cited by C. Duarte and colleagues, the release of DOC by macroalgae from a few species is reported to be $23.2 \pm 12.6$ mmol C m$^{-2}$ d$^{-1}$ with no information on bioavailability. This does not mean that we think that DOC release by macroalgae is unimportant, we fully agree that it could be, however, we are unable to confidently include such dynamics in our model.

To address this issue we will include a statement in the methods section noting that we are unable to include DOC release by macroalgae due to a lack of information. This sentence will state:

"The parameterization of DOC release by macroalgae could not be included because of the lack of enough information. Few studies exist and the uncertainties about the release of DOC are large. For example, Barron et al., 2014 reported a release of DOC by macroalgae from a few species of $23.2 \pm 12.6$ mmol C m-2 d-1 with no information on bioavailability. Meanwhile, refractory DOC dynamics are difficult to include in a global Earth system model and beyond the scope of this study (Anderson et al., 2015; Mentges et al., 2019; Zakem et al., 2021). Thus, the DOC release from macroalgae is not included in this study.".

We will also add text to the discussion section to highlight the need for more research on this topic. This sentence will state:

"Macroalgae has been reported to release a considerable amount of DOC to the global DOC export from coastal to open ocean waters. The estimated total DOC release of macroalgae habitats is 730 PgC yr-1 (Duarte and Cebrian, 1996). The averaged DOC release rate by macroalgae ranges is 23.2 ± 12.6 mmol C m$^{-2}$ d$^{-1}$ (eq. 8.5 ± 4.6mol C m$^{-2}$ yr$^{-1}$), but with a high range of 8.4 ± 1.6 to 71.9 ± 33.1 mmol C m$^{-2}$ d$^{-1}$ (Barron et al., 2014). If we simply multiply this annual averaged DOC release rate with the MOS occupied area (S_MOS, Tab.5), the estimated annual DOC export by MOS would be 7.1 ± 3.8 x10$^3$ PgC (MOS) or 12.9 ± 5.8 x10$^3$ PgC (MOS_AU). Although the refractory DOC released by macroalgae could potentially be an additional contribution of carbon sinking by MOS, the available information of the generation and composition of the macroalgae DOC is not enough to either parameterize a model of this process, and more research on the topic is needed (Krause-Jensen et al., 2016, Barron et al., 2014, Barron et al., 2015)."

**On another issue, I do not see the rationale to have zooplankton graze on farmed macroalgal biomass directly (the Trancoso et al. paper provides no observational evidence supporting this). It makes no sense to me that the same modeled organisms that would ingest phytoplankton would also affect the farmed biomass.**

**Response:** Thank you for the opportunity to clarify why this was parameterized in the model, as both you and one other reviewer picked up on this point. In this study, we simulated the grazing on macroalgae in the marine nutrients–phytoplankton–zooplankton– detritus (NPZD) module of the UVic ESCM by having zooplankton graze on macroalgae. The confusion about having pelagic zooplankton graze on macroalgae likely comes about because we used the original terminology of such models, which designates all higher trophic levels as "zooplankton". We admit that there is little evidence that pelagic zooplankton graze on macroalgae. However, in our model "zooplankton" represent all higher trophic levels, thus, our parameterization is meant to include known macroalgae grazers such as amphipods (Jacobucci et al., 2008), gastropods (Chikaraishi et al., 2007;  Krumhansl et al., 2011), sea urchins (e.g. Yatsuya et al., 2020) and fishes (e.g. Peteiro et al., 2012). Thus, we included this food web pathway to assess the sensitivies of macroalgae to all potential grazers in the ocean, assuming that with large macroalgae farms the pelagic larva of some grazing organisms like fish or urchins, would settle within the farms. Text will be

added to the manuscript to clarify this point and what "zooplankton" actually represents in the model.

However, it is worth noting that even with the assigned grazing preference ($\psi = 1\times10^{-4}$), the "zooplankton" communities do not have large effects, via grazing, on macroalgae NPP nor the biomass of the zooplankton community. Please see the figure below (Fig. B12 in the revised MS).

[Figure]

**Fig. B12**: Global profiles of average zooplankton biomass (left y-axis) & macroalgal NPP of MOS (right y-axis) in comparison with/without zooplankton grazing on macroalgae. The macroalgal NPP drops to zero as MOS is terminated at year 2100. The "zooplankton" communities do not have large effects, via grazing, on macroalgae NPP nor its own biomass.

Last, I am not convinced that biomass once harvested would be transported to depth without any losses. This assumption needs to stated or losses along the sinking path accounted for.

**Response:** Thank you for the opportunity to clarify our assumption, which is a key part of the idea that we test, of sinking biomass to the seafloor to avoid remineralization and maximize the sequestration of carbon. As we test an idea, we have had to assume that once harvested the biomass could be engineered and sunk in some manner. We cannot define exactly how this would be done as some engineering work would need to be conducted to find the most efficient way to sink biomass. However, one could imagine potentially baling the biomass and weighing it down to sink.

To clarify that this is our assumption we will add text to Sect. 2.2.2. after the 1st sentence of the paragraph stating that, "This assumes that the harvested biomass could be engineered to sink to the seafloor in a rapid and efficient manner with no remineralization along the way."

**My last major issue is the writing – both organization and in its execution. There is no single statement of the high-level modeling goals, assumptions and scenarios to be used and the rationale supporting their validity. That information is spread out from pages 4 to 14 (and beyond), making the paper very hard to read and review. This information clearly needs to be in one place – right after the introduction. I am sure that a serious relook at the organization of the paper would really help its overall presentation. With regard to execution, you spend too much time referencing what you are doing that is similar to other works, but do not say in the text what you're actually doing. This is especially annoying in the model introduction. For example, in lines 96-97 you refer to models that yours is based on and then say how yours differs from these, but don't say upfront what your generic adaptation of their will actually do. Also, lines 131-134 were particularly obtuse, but there are many other examples throughout. The model introduction section needs to be understandable without making the reader refer to a stack of other papers. This is a correctable writing issue that made it hard to review the text but requires serious attention.**

**Response**: Thank you for the suggestion to improve the structure of the manuscript and to provide some more key information on the aims and details of the study. We will revise the organization of the manuscript and add more information to correct these deficiencies. First, as mentioned in the response to your first general comment we have added sentences stating the overall aims of the study to the 5th paragraph of Sect.1: Introduction.

We have also added text throughout the manuscript to highlight how our study addresses these aims:

In the 1st paragraph of Sect. 2.2.1, we added a sentence that states, "...Instead, the C:N:P ratio of the macroalgae biomass is assumed as a constant (Tab.1), which is based on seasonally averaged measurements of the biomass composition of these genus (Zhang et al., 2016; Martins and Marques, 2002)."

We have modified the description of the limiting factor of solar radiation intensity functions (Eq.11 & Eq.12) in Sect.2.2.1.

To better describe the model so that the reader doesn't have to reference other articles we have made the following changes in Section 2.2:

1. We added separate two paragraphs at the beginning of Sect. 2.2: "In this study, the modelling of macroalgae is done with a macroalgae growth model coupled in the UVic ESCM. In the macroalgae model, the net growth rate is affected by several limiting factors, including nutrients, temperature, and solar radiation intensity. The cellular C:N:P ratio of macroalgae is fixed. The loss of macroalgal biomass includes erosion and grazing by zooplankton. The deployment of MOS is done with an algorithm considering spatial and temporal conditions.

   The macroalgae model is also connected to the global marine biogeochemical processes, including the inorganic carbon and nutrient pools. On the surface layers, it impacts on phytoplankton via nutrients competition and canopy shading. The zooplankton communities are also designed to graze on macroalgae. In the bottom layers, the remineralization of sunk macroalgal biomass will consume the dissolved oxygen, which in turn limits the rate of remineralization."

2. Sect. 2.2.1: " The macroalgae model is an idealized generic model of genus *Laminaria* and *Saccharina*, mainly based on Martins and Marques (2002) and Zhang et al., 2016. The rate of biomass change is governed by Eq. 1 as the imbalance of **NGR** (net growth rate, $d^{-1}$) and **LR** (loss rate, fraction of daily biomass loss due to mortality, erosion and grazing by zooplankton, $d^{-1}$).

   The aim of the macroalgae model is to investigate the carbon sequestration capacity of MOS as well as the potential impacts on marine biogeochemistry. Modelled macroalgae is seeded 5 meters underwater, considering the light

requirement and reduction of damaging risks (Eq. 11). The deployment of macroalgae considers ambient nutrients availability and avoidance of winter periods (Sect. 3.1).

3. At the end of Sect. 2.2.1: For simplicity and to limit the number of state variables, we made the following modifications to the macroalgae model:

1. We did not include a dynamic C:N:P ratio or a representation of luxury nutrient uptake and storage (Broch and Slagstad, 2012; Hadley et al., 2015). Instead, the C:N:P ratio of the macroalgae biomass was set as a constant (Tab.1), which is based on seasonally averaged measurements of the biomass composition of these genus (Zhang et al., 2016; Martins and Marques, 2002).

2. The macroalgae life cycle processes (e.g. alternations of generations) are also not considered in our model (Brush and Nixon, 2010; Trancoso et al., 2005; Duarte and Ferreira, 1997). We thus assumed that the plantlet (e.g. sporophytes for Saccharina) will be reseeded annually on the MOS infrastructure. The assumed deployment strategy, i.e., timing of seeding and sinking of MOS is latitude-dependent according to the seasonality of solar irradiance (see Sect.3.1). Whenever conditions are unfavorable for macroalgae and no growth occurs during an annual cycle, no re-seeding of macroalgae will occur in these regions.

**Detailed comments follow.**

**Lines 131-146 were hard to follow without having the macroalgae cultivation assumptions stated earlier and a separate section outlining modeling goals and assumptions is needed.**

We have added the 2nd paragraph in Sect. 2.2.1, which reads:

"The aim of the macroalgae model is to investigate the carbon sequestration capacity of MOS as well as the potential impacts on marine biogeochemistry. Modelled macroalgae is seeded 5 meters underwater, considering the light requirement and reduction of damaging risks (Eq. 11). The deployment of macroalgae considers ambient nutrients availability and avoidance of winter periods (Sect. 3.1)."

Meanwhile, the general goal of this study is described in the Introduction, which reads:

"The aim of this study is to investigate 1) the maximum physical / biogeochemical CDR potential of MOS; 2) the side effects of such large scale deployment, and 3) to understand where offshore macroalgae farming would be viable if done at a large scale. This information is needed to help prioritize further research into CDR, to understand if there are potential MOS side effects that become evident only at large scale, and to provide information on the viability of large-scale offshore macroalgae farming in different regions over time by accounting for the implications of nutrient utilization and climate change."

**Lines 146-155 –    Please tell the reader why is R_erosion a constant without having to refer to the reference.**

Thank you for pointing out this issue. We have now revised these sentences as : "where the erosion of biomass (ER) is controlled by the individual erosion rate $R_{erosion}$. As the frond morphology of macroalgae is not modelled here, we set the $R_{erosion}$ as a constant independent of physical impacts (Trancoso et al., 2005; Zhang et al., 2016)."

**Lines 204-214 – Definition of FR is out of order and needs to be moved down after the sunk biomass is defined.**

Thank you for the suggestion. We agreed and have moved the Eq.23 of FR downward. Now it reads as:

"The DIC from remineralization of sunk biomass will eventually be conveyed back to the ocean surface and may leak back to the atmosphere. Eq.23 calculates the ocean-retained fraction (FR, %) of MOS-captured carbon (MOS-C),

where the $C_{captured}$ is carbon in cumulative sunk biomass, $C_{Sunk\ Biomass}$ is the carbon in sunk macroalgal biomass that still remains on the seafloor.

$$FR = \frac{C_{retained}}{C_{captured}} = \frac{(DIC_{remineralized} + C_{sunkbiomass})}{C_{captured}} \quad (23)"$$

**Table 1 – What is the column denoted "Reference" referring to?  The references in this paper are not numbered.**

Sorry for the confusion. These were in an early version, but were inadvertently left out of the final draft. The references have now been added directly to the table.

**Table 1.** Model parameters

| Symbol | Parameter | Unit | Value | Reference |
|---|---|---|---|---|
| $a_{ma}$ | Macroalgae carbon specific shading area | $m^2 kgC^{-1}$ | 11.1 | Trancoso et al. (2005) |
| $d$ | Distance between the cultivating ropes | m | 10 | Van Der Molen et al. (2017) |
| $R_{erosion}$ | Individual erosion rate | $\% \, d^{-1}$ | 0.01 | Zhang et al. (2016) |
| $I_{opt}$ | Optimum light intensity for macroalgae growth | $W \, m^{-2}$ | 180 | Zhang et al. (2016) |
| $NO_3$ | Nitrate Concentration | $\mu mol \, l^{-1}$ | model calculation | Keller et al. (2012) |
| $PO_4$ | Phosphate Concentration | $\mu mol \, l^{-1}$ | model calculation | Keller et al. (2012) |
| $K_N$ | Half-saturation constant for nitrogen uptake | $\mu mol \, l^{-1}$ | 2 | Zhang et al. (2016) |
| $K_P$ | Half-saturation constant for phosphorus uptake | $\mu mol \, l^{-1}$ | 0.1 | Zhang et al. (2016) |
| $k_w$ | Coefficient of light attenuation through water | $m^{-1}$ | 0.04 | Keller et al. (2012) |
| $k_c$ | Coefficient of light attenuation through phytoplankton | $m^{-1}(mmol \, m^{-3})$ | 0.047 | Keller et al. (2012) |
| $M_{ma}$ | Thickness of MOS macroalgae canopy | m | 10 | Trevathan-Tackett et al. (2015) |
| $MR_{C:N}$ | Molar C:N ratio of macroalgal biomass | - | 20 | Atkinson and Smith (1983) |
| $MR_{P:N}$ | Molar P:N ratio of macroalgal biomass | - | 0.05 | Atkinson and Smith (1983) |
| $MR_{C:P}$ | Molar C:P ratio of macroalgal biomass | - | 200 | calculated |
| $MR_{DW:WW}$ | Ratio of DW to WW of macroalgal biomass | - | 0.1 (values reported:0.05~0.2) | Aldridge and Trimmer (2009); Conover et al. (2016); Van Der Molen et al. (2017) |
| $MR_{C:DW}$ | Carbon content of dried macroalgal biomass | % | 30 | Chung et al. (2011) |
| $MR_{N:DW}$ | Nitrogen content of dried macroalgal biomass | - | 0.16 | Duarte et al. (2003) |
| $R_{max20}$ | Maximum respiration rate at 20°C | $\% \, d^{-1}$ | 1.5 | Martins and Marques (2002); Zhang et al. (2016) |
| $r$ | Empirical coefficient for macroalgae respiration | $d^{-1}$ | 1.047 | Martins and Marques (2002) |
| Seed | Initial macroalgal biomass per kilometer cultivating line | $kgC \, km^{-1}$ | 2.5 | Van Der Molen et al. (2017) |
| $T_b$ | E-folding temperature of biological rates | °C | 15.56 | Schmittner et al. (2008) |
| $T_{opt}$ | Optimum temperature for growth | °C | 20(values reported: 13-30) | Zhang et al. (2016); Martins and Marques (2002) |
| $T_{max}$ | Upper temperature limit above which growth ceases | °C | 35 | Breeman (1988) |
| $T_{min}$ | Bottom temperature limit below which growth ceases | °C | 0 | Martins and Marques (2002) |
| $u_{max}$ | Maximum growth rate | $d^{-1}$ | 0.2 | Zhang et al. (2016) |
| $w$ | Areal mean artificial upwelling rate | $cm \, d^{-1}$ | 1,10 | Oschlies et al. (2010b) |
| $Y_{max}$ | Maximum yield of macroalgal biomass on MOS | $t \, DW \, km^{-2} \, yr^{-1}$ | 3300 | |
| $\psi_{ma}$ | Zooplankton grazing preference on macroalgae | - | $1 \times 10^{-4}$ | Trancoso et al. (2005) |
| $\mu_{ma0}$ | Remineralization rate of sunk macroalgal biomass at 0 °C | $\% \, d^{-1}$ | 7 | Partanen et al. (2016) |

**Line 236 – The units for Seed and KN do not match in Table 1. ???**

Sorry for the confusion. To make the concept of Seed more comprehensive, we provide the unit of *Seed* in Table 1 as kgC km$^{-1}$, referring to 'Initial macroalgal biomass per kilometer cultivating line'. However, in the UVic ESCM, macroalgae biomass is calculated as concentration of N in the ocean model (Seed$_{conc}$, mmol N m$^{-3}$). It can be converted according to the mass conversion functions in Sect. 2.2.4 and Appendix A1 as:

Seed$_{conc}$ = Seed/(12 x MR$_{C:N}$ x d x Depth$_{1st\_layer}$)

Where the $Depth_{1st\_layer}$ is 50m, referring to the depth of the top layer of UVic ESCM. Accordingly, the Seed of 2.5 kgC km$^{-1}$ is equal to $Seed_{conc}$ of 0.02 mmol N m$^{-3}$.

We have modified the item of *Seed* in Table 1. Now it reads as

| Symbol | Parameter | Unit | Value | References |
|--------|-----------|------|-------|-----------|
| Seed | Initial macroalgal biomass per kilometer cultivating line

concentration of N | kgC km$^{-1}$

mmol N m$^{-3}$ | 2.5

0.02 | Van Der Molen et al.(2017)

calculated |

We have modified this sentence to:

" The ambient surface NO3 concentration is greater than *Seed* plus $K_N$ (Tab. 1); this ensures sufficient nutrients for initial growth as *Seed* is directly transferred from dissolved $NO_3$ and $K_N$ is the half saturation constant for $NO_3$ uptake. Note that in this calculation, *Seed* is 0.02 mmol N m$^{-3}$".

**Lines 244 and Fig 3 – It took me a long time to figure out what belts were referring to.**

Thank you for the comments about the belt and selected area and sorry for the confusion.

We have revised this paragraph as:

"During the MOS simulations, seasonality of temperature as well as solar radiation are essential limiting factors of the primary productivity of MOS in various latitudinal regions. In order to avoid the unnecessary loss of macroalgal biomass during winter periods when solar radiation is insufficient and the ambient water temperature is low, we partitioned the global ocean surface into three belts (N, M and S) and pragmatically applied farming strategies according to Tab. 2. The period between the seeding and sinking of macroalgae is set as six months from May to October in belt N and from November to the next April in belt S. In belt M, the macroalgae is seeded at the beginning of the year and sinking occurs after 12 months. The geographical locations of the three belts are shown in Fig. 2. "

We have updated Figure 2 with braces referring to various belts (N, M, S):

[Figure]

**Figure 2.** Annual vertically integrated macroalgae biomass of MOS (a) and MOS_AU (b) in year 2024. Red solid lines outline the MOS occupied area at year 2024 in both, while red dashed lines outline the initial MOS seeding area at year 2020 in (a). The simulated MOS area generally covers the $NO_3$-rich ocean surface (a) and can be expanded with nutrients supplemented by AU (b), larger than the estimated adequate area for macroalgae in previous studies(Lehahn et al., 2016, Froehlich et al., 2019). Results for areas I (blued circle), II (yellowish pentagon) and III (cyan rectangle) are discussed in the text and displayed in Fig 3. Braces indicate the belts of N, M and S with various seeding strategies of macroalgae (Tab. 2), which are designed to avoid winter periods.

**Table 4 – Do not understand the difference between "selected area" and "belt" and why there are units there in km^2.**

The selected area shown in Figure 2 refer to 3 representative MOS areas in their belts. These selected areas are fully occupied by MOS, allowing us to better validate the simulated macroalgae NPP and biomass yield. In comparison, the belts are designed for the seeding strategy of MOS in order to avoid unnecessary loss of macroalgal biomass during the various winter periods in each belt, when solar radiation is insufficient and the ambient water temperature is low.

We used $km^2$ for the selected areas and belts as $km^2$ is a common unit for area due to their magnitude (from 1,000 to $10^9$ $km^2$).

**Line 256 – This topic sentence is not useful. Please state what are in the references.**

Sorry for the confusion. We have added more information to this paragraph. Now it reads as:

"The simulated MOS-AU system is based on Oschlies et al., 2010 and Keller et al., 2014. We placed modelled 'pipes' that pump water from the 1,000m deep lower end of the pipe to the ocean surface in areas where MOS is deployed. The simulated upwelling works by transferring water adiabatically from the grid box at the lower end of the pipe to the surface grid box at a rate of 1 cm day$^{-1}$. These pipes will function continuously until the termination of MOS (in year 2100 or 3000). However, because these earlier studies have revealed a dominant effect associated with low temperatures of the upwelled colder waters, we here concentrate on the nutrient aspect and simulate a hypothetical MOS-AU system that keeps temperatures at ambient levels (e.g. via heat exchangers)."

**Lines 306-318 – The "validation" should be in a table comparing the NPP, yields, etc. from farms with the model results.**

Thank you for the suggestion. We have added a column in Table 4 with the observational data mentioned in Sect. 4.1.2. The table is now:

Table 4. Properties of globally implemented MOS. Selected areas are from data of year 2024, whereas Belt areas are values averaged from 2020 to 2024. Areal NPP rates and Biomass Yield refer to the respective MOS area. The observational data comes from the references in the footnotes and is only comparable to MOS.

| Property | Unit | Observations | Exp. | Selected area($10^3$ km$^2$) | | | Belt($10^9$ km$^2$) | | | |
| --- | --- | --- | --- | --- | --- | --- | --- | --- | --- | --- |
| | | | | Area I | Area II | Area III | N | M | S | Global |
| MOS occupied area(S$_{MOS}$) | km$^2$ | - | MOS | 218.8 | 320.3 | 204.2 | 9.1 | 15.7 | 44.8 | 69.6 |
| | | - | MOS_AU | | | | 17.4 | 44.3 | 64.6 | 126.3 |
| NPP | gC m$^{-2}$ yr$^{-1}$ | 91-750[1] | MOS | 159.2 | 176.9 | 199.3 | 50.8 | 52.0 | 67.5 | 61.8 |
| | | | MOS_AU | 202.2 | 231.1 | 217.7 | 45.5 | 32.2 | 56.9 | 46.7 |
| Biomass Yield | t DW km$^{-2}$ yr$^{-1}$ | 40-456[2] | MOS | 648.2 | 492.4 | 579.7 | 173.2 | 160.3 | 206 | 191.4 |
| | | Area I: 300-7,280[3] | MOS_AU | 715.3 | 615.4 | 597.0 | 142.2 | 85.4 | 169 | 136 |
| Total CO$_2$ captured in biomass | Pg CO$_2$ yr$^{-1}$ | - | MOS | 0.14 | 0.16 | 0.12 | 1.6 | 2.5 | 9.2 | 13.3 |
| | | - | MOS_AU | 0.15 | 0.20 | 0.12 | 2.5 | 3.8 | 10.9 | 17.2 |

[1]Krause-Jensen and Duarte (2016); [2]Buck and Buchholz (2004); Peteiro et al. (2014); [3]Zhang et al. (2016); Yokoyama et al. (2007)

**Lines 357 – Define PNPP before you use it.**

Thank you for pointing out this issue. The definition of PNPP is given in the abstract, however, it is too far from the main text. We have modified this sentence according to your suggestion, which now reads as: "One reason is that the reduced oceanic carbon uptake

by declined PNPP (phytoplankton net primary production, Sect. 4.3.4) offsets ~37% of the MOS-induced carbon sequestration."

**Lines 375-376 – Not sure if that obvious statement is needed.**

Agreed, we have removed the statement

**Line 377 – Is denitrification in the model?  It is not stated in the model description.**
Please see Sect. 2.2.2, in which we briefly describe the denitrification in the model. It states as "When the dissolved oxygen is insufficient (<5 mmol m$^{-3}$), aerobic remineralization will be replaced by oxygen-equivalent, but slower, denitrification via reduction of $NO_3$ (Keller et al., 2012). Note that remineralization will cease when $NO_3$ is completely consumed.".

**Correction of the colorbar and caption of Figure B1**

We have updated the colorbar and caption of Figure B1 after the correction of a numerical error in the biomass fresh weight conversion. These corrections in no way change any of the other results in the manuscript, and the biomass numbers are still within the valid range.

[Figure]

Figure B1: MOS biomass distributions. Red lines contour the maximum MOS occupied area during the previous years. The annual macroalgal biomass of MOS in this figure is an average over a 10 year period, which includes times of low and high biomass due to the sinking of biomass. Thus the biomass shown here is less than the biomass shown in Fig. 2a.

**Responses to Suggestions for Revision by Dr. Lennart Bach (Report #1)**

Dear Editor/Authors: Thanks for carefully working on the comments. My remaining comments related to the tracked-changes version of the manuscript.

Dear Dr. Bach,

We appreciate again your very helpful and constructive comments on our manuscript "Carbon Dioxide Removal via Macroalgae Open-ocean Mariculture and Sinking: An Earth System Modeling Study". The manuscript has been revised to incorporate your suggestions. Please see a point-by-point response to your comments and concerns below. The line numbers are related to the latest revised version of the manuscript without tracked changes.

Line 7: The numbers presented here are still misleading in my opinion. The number given here is the build-up of biomass, not the CDR potential (which in my opinion is C_oc). The CDR potential should consider the net gain, or "additionality" of the approach.

**Response:** Thank you for noting this important issue. In order to avoid ambiguity, in the text we have included both the carbon captured in MOS biomass as well as the enhancement of oceanic carbon reservoirs. The sentences now read as:

"The simulations are done under RCP4.5, a moderate emission pathway. When deployed globally between years 2020 and 2100, the carbon captured and exported by MOS is 270 PgC, which is further boosted by AU to 447 PgC. Because of feedbacks in the Earth system, the oceanic carbon inventory only increases by 171.8 PgC (283.9 PgC with AU) in the idealized simulations."

Line 18: The last two sentences are still inconsistent. When it has theoretically a considerable potential then it must be made clear that it not only "can have" substantial side-effects but that the scale considered here for "considerable potential" means more or less a complete change in marine food webs in large parts of the oceans. I think this is not articulated clear enough.

**Response:** We agree that the potential side effects of MOS large-scale deployment should be more prominently stated. We have revised the sentences from line 19 to 21, which now reads:

"Our results suggest that MOS has, theoretically, a considerable CDR potential as an ocean-based CDR method. However, our simulations also suggest that such large-scale deployment of MOS would have substantial side effects on marine ecosystems and biogeochemistry up to a reorganisation of food webs over large parts of the ocean."

Line 45: I would avoid the word efficient when it comes to PP because on a per biomass level they are way less efficient than phytoplankton. The word works better in the context of CN.

**Response:** Thank you for mentioning these details. The modified text in line 46 to 47 reads as:

"Macroalgae species (also known as `seaweed' or `kelp') are highly efficient carbon fixers with a high C:N ratio (Atkinson and Smith, 1983; Fernand et al., 2017) and observed net primary production (NPP) rates of 91-522 gC m$^{-2}$ yr$^{-1}$."

Line 76: This new sentence should come after "…centuries to millennia (Fig. 1)" as it breaks the flow and disconnects the "short-circuiting" sentence from what it refers to. Or maybe even to line 116.

**Response:** We thank the reviewer for this good suggestion and have revised the text accordingly. Line 79-81 now reads as:

" …centuries to millennia (Fig.1). The macroalgae used here is an idealized genus. The assumed constant C:N:P ratio is 400:20:1, which is higher than the stoichiometric ratio of the general phytoplankton in the UVic ESCM (C:N:P=106:16:1, the Redfield ratio)."

Line 153: In this case please also say that seaweed CDR would be a variety of ocean iron fertilization.

**Response:** We agree that if deployed with an iron supplement, the additional iron would also fertilize the ambient phytoplankton, which would make MOS a variety of ocean iron fertilization. We have additionally clarified in text from line 140 to 142, which now reads as:

"Besides, as iron is a micronutrient needed in low quantities, the MOS platform could be designed with an iron supply for the macroalgae, in which case MOS could be considered to include a targeted variant of the ocean iron fertilization concept."

Line 365: maybe refer to table 3 here.

**Response:** Thank you for the suggestion, line 366 now reads as:

"..., especially in the equatorial Eastern Pacific and the Southern Ocean (Tab.3)."

Line 507: Which of the simulations?

**Response:** The simulations refer to MOS_Conti and MOS_Stop. The revised text in line 482 now reads as:

"In the MOS simulations (MOS_Conti/Stop), globally…"

Line 507: why is everything given as a cumulative until 2100 or 3000 but PNPP not in Table 5?

**Response:** In Table 5 only CO2 emissions and MOS-C are given as cumulative values. Inventories of the carbon reservoirs, ΔSAT and ΔNPP are provided as annual-mean snapshots for years 2100 and 3000 to provide a picture of the system at work at these moments in time.

Line 686: 730 Pg C / year??? This is more than the entire DOC pool in the oceans. Surely there must be an error here? Seems high even when considering that >>90 percent will be remineralised in weeks.

Line 700: 7100 Pg C??? All DOC of which almost all would be reminerlised? These numbers are confusing as they suggest a huge number which is not really relevant if it is basically all turn-around carbon.

**Response**: Thank you for noting these and sorry for the incorrect numbers and calculations. We have rectified them and revised the respective text from line 667 to 674, which now reads:

"The estimated total DOC release from macroalgae habitats is 355 Tg C yr−1 (Krause-Jensen and Duarte, 2016). The averaged DOC release rate (plus minus standard deviation) by macroalgae is 23.2 ± 12.6 mmol C m−2 d−1 (equivalent to 8.5 ±4.6mol C m−2 yr−1), but with a high range of 8.4 to 71.9 mmol C m−2 yr−1 (Barrón et al., 2014). If we simply multiply this annually averaged DOC release rate with the MOS occupied area (SMOS , Tab.5), the estimated annual DOC release by MOS would be 7.1 ± 3.8 PgC (MOS) or 12.9 ± 7.0 PgC (MOS_AU). Although the refractory DOC released by macroalgae could potentially be an additional contribution of carbon sinking by MOS, the available information about the generation and composition of the macroalgae DOC is not enough to parameterize a model of this process, and more research on the topic is needed (Krause-Jensen and Duarte, 2016; Barrón et al., 2014; Barrón and Duarte, 2015)"

Line 748: I suggest "...has considerable potential under these idealized conditions but..."

**Response**: Thanks for the comment. We agree and have revised the text in line 719, which now reads:

"The evidence from this study suggests that macroalgae mariculture & sinking has a considerable CDR potential under these idealized conditions, but brings about substantial side effects on marine ecosystems, and marine biogeochemistry."

**Responses to Editor decision: Publish subject to minor revisions**

The authors have addressed the first round of referee comments mostly satisfactorily. One referee has submitted further minor responses, please can the authors address them.

In addition, the point made by Referee 2 on the assumption to transport to depth without losses should be highlighted in the abstract as well as the main text, as this is a fundamental assumption.

If these can be actioned, the manuscript can be accepted.

**Response:**
Dear Richard,

Thank you for your helpful comments in the editor report of our manuscript.

We have submitted our response to the minor revision request in Report #1 in our last email attachment. If there are further response from the reviewer 1 after the Report #1 which has been replied, please kindly remind us, and we would revise accordingly.

Concerning the point made by Referee 2 on the assumption to transport to depth without losses, we have already stated in the 1st paragraph of Sect 2.2.2. (line 204 to 206 of the revised manuscript), which reads: 'Biomass sinking is simulated by instantly transferring the macroalgal biomass from the surface grid cell to the deepest grid cell at the respective location at the end of each cultivating period. This assumes that the harvested biomass could be engineered to sink to the seafloor in a rapid and efficient manner with no remineralization along the way.'

Besides, now we have revised the 2nd sentence of the abstract, which now reads: 'Embedding a macroalgae model into an Earth system model, we simulate macroalgae mariculture in the open-ocean surface layer followed by fast sinking of the carbon-rich macroalgal biomass to the deep seafloor (depth > 3,000m), which assumes no remineralization of the harvested biomass during the quick sinking.' Thank you again for pointing out this issue.

We greatly appreciate your time dealing with the manuscript and assessing the review comments. We will submit a revised manuscript based on your and the reviewers' comments.